# Distributionally Robust Optimization with Bias and Variance Reduction

**Ronak Mehta**[*]  **Vincent Roulet**[†]  **Krishna Pillutla**[‡]  **Zaid Harchaoui**[*]

## Abstract

We consider the distributionally robust optimization (DRO) problem, wherein a learner optimizes the worst-case empirical risk achievable by reweighing the observed training examples. We present Prospect, a stochastic gradient-based algorithm that only requires tuning a single learning rate hyperparameter, and prove that it enjoys linear convergence for smooth regularized losses. This contrasts with previous algorithms that either require tuning multiple hyperparameters or potentially fail to converge due to biased gradient estimates or inadequate regularization. Empirically, we show that Prospect can converge 2-3x faster than baselines such as SGD and stochastic saddle-point methods on distribution shift and fairness benchmarks spanning tabular, vision, and language domains.

## 1 Introduction

The ingredients of empirical risk minimization (ERM) are generally considered to be: a model with parameters $w \in \mathbb{R}^d$ (e.g. a neural network), a training set $z_1, \ldots, z_n \in \mathcal{Z}$ of independent and identically distributed realizations of a random variable $Z \sim P$, a loss function $\ell : \mathbb{R}^d \times \mathcal{Z} \to \mathbb{R}$, and an optimization algorithm that solves

$$\min_{w \in \mathbb{R}^d} \mathbb{E}_{Z \sim P_n} \left[ \ell(w, Z) \right], \tag{1}$$

where $P_n$ is the empirical distribution of $\{z_i\}_{i=1}^n$. The fourth ingredient–often taken for granted–is the choice of *risk functional*, which aggregates the distribution of $\ell(w, Z)$ into a univariate summary to be minimized. The objective (1) (the expected loss under $P_n$) is an unbiased estimate of the expected loss under an underlying data-generating distribution $P$; however, a deployed model often observes data from distributions other than $P$. Motivated by this practical phenomenon, we consider instead an objective that explicitly captures sensitivity to such distribution shifts:

$$\min_{w \in \mathbb{R}^d} \max_{Q \in \mathcal{Q}} \; \mathbb{E}_{Z \sim Q} \left[ \ell(w, Z) \right] - \nu D(Q \| P_n), \tag{2}$$

in which $\mathcal{Q}$ is an *uncertainty set* of probability measures, $\nu \geq 0$ is a hyperparameter, and $D(Q \| P_n)$ represents the divergence of $Q$ from the original training distribution $P_n$ (e.g. the $\chi^2$ or Kullback Leibler divergence). The objective (2) emulates a game in which nature pays a price of $\nu$ per unit $D(Q \| P_n)$ to replace the expected loss under $P_n$ with the expected loss $\mathbb{E}_{Z \sim Q} \left[ \ell(w, Z) \right]$ associated with the shifted distribution $Q$. Since $\nu$ penalizes these shifts, we shall refer to it as the *shift cost*.

Objectives of the form (2), known as *distributionally robust optimization (DRO)* problems, have seen a wave of recent interest in machine learning theory and practice (Chen & Paschalidis, 2020). Historically used in quantitative finance, a popular such objective is the conditional value-at-risk (CVaR, a.k.a. superquantile/expected shortfall/average top-$k$ loss). In terms of methods, the CVaR has been used as a canonical DRO objective (Fan et al., 2017; Kawaguchi & Lu, 2020; Rahimian & Mehrotra, 2022), as well as in unsupervised learning (Maurer et al., 2021), reinforcement learning (Singh et al., 2020), and federated learning (Pillutla et al., 2023). In applications, it has also been employed for robust language modeling (Liu et al., 2021) and robotics (Sharma et al., 2020). The superquantile/CVaR falls into the broader category of *spectral risk measures (SRMs)*, a class of DRO objectives that includes the extremile and exponential spectral risk measure (ESRM) (Acerbi & Tasche, 2002; Cotter & Dowd, 2006; Daouia et al., 2019).

---

[*]University of Washington, [†]Google DeepMind, [‡]Google Research.

Motivated by 1) the success of the superquantile in applications and 2) the importance of stochastic optimization in machine learning, *the principal goal of this paper is to develop stochastic[1] algorithms for spectral risk minimization.*

**Contributions.** We propose Prospect, a stochastic algorithm for optimizing spectral risk measures with only one tunable hyperparameter: a constant learning rate. Theoretically, Prospect converges linearly for *any* positive shift cost on regularized convex losses. This contrasts with previous stochastic methods that fail to converge due to bias (Levy et al., 2020; Kawaguchi & Lu, 2020), may not converge for small shift costs (Mehta et al., 2023), or have multiple hyperparameters (Palaniappan & Bach, 2016). Experimentally, Prospect demonstrates equal or faster convergence than competitors on the training objective on nearly all objectives and datasets considered, and exhibits higher stability with respect to external metrics on fairness and distribution shift benchmarks.

**Related Work.** Examples of DRO formulations range throughout diverse contexts such as reinforcement (Liu et al., 2022b; Kallus et al., 2022; Liu et al., 2022c; Xu et al., 2023; Wang et al., 2023; Lotidis et al., 2023; Kallus et al., 2022; Ren & Majumdar, 2022; Clement & Kroer, 2021), continual (Wang et al., 2022), interactive (Yang et al., 2023; Mu et al., 2022; Inatsu et al., 2021; Sinha et al., 2020), Bayesian (Tay et al., 2022; Inatsu et al., 2022), and federated (Deng et al., 2020; Pillutla et al., 2023) learning, along with dimension reduction (Vu et al., 2022), computer vision (Samuel & Chechik, 2021; Sapkota et al., 2021), and structured prediction (Li et al., 2022; Fathony et al., 2018). Various forms of these objectives are parameterized by the uncertainty set $\mathcal{Q}$, including those based on $f$-divergences (Levy et al., 2020; Ben-Tal et al., 2013), the Wasserstein metric (Blanchet et al., 2019b; Kuhn et al., 2019), maximum mean discrepancy (Kirschner et al., 2020; Staib & Jegelka, 2019; Nemmour et al., 2021), or more general classes of metrics (Husain, 2020; Shapiro, 2017). We focus on SRM objectives, as motivated in detail in Sec. 2 and Appx. B.

These objectives also yield connections to other areas in modern machine learning. They are a special case of *subpopulation shift*, wherein the data-generating distribution is modeled as a mixture of subpopulations, and the distribution shift stems from changes in the mixture. In our case, the subpopulations are point masses at the observed data points. In the context of *algorithmic fairness*, the subpopulations may represent data conditioned on some protected attribute (e.g. race, gender, age range), and common notations of fairness such as *demographic/statistical parity* (Agarwal et al., 2018; 2019) impose (informally) that model performance with respect to each subpopulation should be roughly equal. As such, robustness to reweighting and algorithmic fairness are often aligned notions (Williamson & Menon, 2019), with recent research arguing that distributionally robust models are more fair (Hashimoto et al., 2018; Vu et al., 2022) and that fair models are more distributionally robust (Mukherjee et al., 2022). In supervised learning, the data distribution is modeled as $P = P_{X,Y}$ for a feature-label pair $(X, Y)$ and related settings of *covariate shift* (changes in $P_X$ and not $P_{Y|X}$) (Sugiyama et al., 2007) as well as *label shift* (changes in $P_Y$ and not $P_{X|Y}$) (Lipton et al., 2018) may also modeled with distributional robustness (Zhang et al., 2021) as illustrated in Fig. 1. In these settings, distributional robustness may be described as a property of learned representations that are transferable to multiple tasks (Słowik & Bottou, 2022).

In comparisons, we include stochastic algorithms that either are single-hyperparameter "out-of-the-box" methods such as stochastic gradient descent and stochastic regularized dual averaging (Xiao, 2009), or multi-hyperparameter methods that converge linearly on strongly convex SRM-based objectives, such as LSVRG (Mehta et al., 2023) and stochastic saddle-point SAGA (Palaniappan & Bach, 2016). Note that LSVRG may not converge for small shift costs. Other methods may achieve sublinear convergence rates, even with multiple hyperparameters (Yu et al., 2022; Ghosh et al., 2021; Carmon & Hausler, 2022; Li et al., 2019; Shen et al., 2022; Yazdandoost Hamedani & Jalilzadeh, 2023; Namkoong & Duchi, 2016). Non-convex settings have also been studied (Jin et al., 2021; Jiao et al., 2022; Sagawa et al., 2020; Luo et al., 2020), as well as statistical aspects of resulting minimizes of DRO objectives (Liu et al., 2022a; Blanchet et al., 2019a; Zeng & Lam, 2022; Maurer et al., 2021; Lee et al., 2020; Khim et al., 2020; Zhou & Liu, 2023; Zhou et al., 2021; Cranko et al., 2021; Słowik & Bottou, 2022). Our goal is to achieve unconditional linear convergence for smooth, strongly convex (regularized) losses with a single hyperparameter.

---

[1] We use *stochastic* interchangeably with *incremental*, meaning algorithms that make $O(1)$ calls per iteration to a fixed set of oracles $\{(\ell_i, \nabla \ell_i)\}_{i=1}^n$, and **not** *streaming* algorithms that sample fresh data at each iteration.

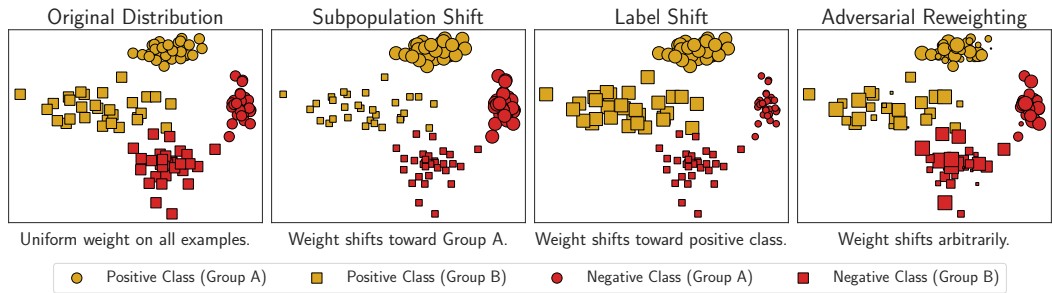

Figure 1: **Notions of Distribution Shift**. Illustration of various forms of distribution shift that are characterized by maintaining the same training data but changing the weight of each example.

## 2 MINIMIZING SPECTRAL RISK WITH BIAS AND VARIANCE REDUCTION

We first describe the key technical challenges in constructing a stochastic optimizer for spectral risk measures and how Prospect tackles them. Because we consider a fixed training set of size $n$ and use $f$-divergences for $D$ (reviewed in Appx. B), we may parameterize probability measures $Q$ by a set of weights $q \in \mathcal{P} \subseteq \{p \in \mathbb{R}^n : p_i \geq 0, \sum_{i=1}^n p_i = 1\}$ indicating the probability mass allotted to training examples $(z_1, \ldots, z_n)$. Similarly, the empirical distribution $P_n$ is represented by the uniform weights $\mathbf{1}_n/n = (1/n, \ldots, 1/n)$. We can then rewrite (2) in the convenient form

$$\min_{w \in \mathbb{R}^d} \mathcal{R}_\mathcal{P}(\ell(w)) \text{ for } \mathcal{R}_\mathcal{P}(l) := \max_{q \in \mathcal{P}} \left\{ \sum_{i=1}^n q_i l_i - \nu D(q \| \mathbf{1}_n/n) \right\}, \tag{3}$$

where $\ell(w) = (\ell_1(w), \ldots, \ell_n(w)) \in \mathbb{R}^n$ and $\ell_i(w) = \ell(w, z_i)$ is the loss on training example $i$. In order to build a convergent stochastic algorithm, we will construct an estimate $v_i$ for the gradient of (3) based on a single data index $i$, such that $v_i \to \nabla \mathcal{R}_\mathcal{P}(\ell(w))$ as the iteration counter approaches infinity. Precisely, we require that for $i \sim \text{Unif}[n]$,

$$\mathbb{E}\|\nabla \mathcal{R}_\mathcal{P}(\ell(w)) - v_i\|_2^2 = \underbrace{\|\nabla \mathcal{R}_\mathcal{P}(\ell(w)) - \mathbb{E}[v_i]\|_2^2}_{\text{bias}} + \underbrace{\mathbb{E}\|\mathbb{E}[v_i] - v_i\|_2^2}_{\text{variance}} \tag{4}$$

decreases to zero asymptotically. In the remainder of this section, we first identify concretely our target estimand (i.e. $\nabla \mathcal{R}_\mathcal{P}(\ell(w))$ for the spectral risk uncertainty set), construct an estimate, and then describe individual procedures to ensure that the bias and variance terms in (4) vanish.

**Spectral Risk Measures and their Gradients.** The conventional $p$-superquantile/CVaR (Rockafellar & Royset, 2013) of a loss vector $(\ell_1(w), \ldots, \ell_n(w))$ is defined as $\sum_{i=1}^n \sigma_i \ell_{(i)}(w)$, where $\ell_{(1)}(w) \leq \ldots \leq \ell_{(n)}(w)$ are the ordered losses and $\sigma_i = 1/k$ for the largest $k = np$ values of $i$ and $\sigma_i = 0$ otherwise (see Equation (21)). Other spectral risk measures (SRMs) are generated by constructing any vector $\sigma = (\sigma_1, \ldots, \sigma_n)$ of non-negative weights satisfying $\sigma_1 \leq \cdots \leq \sigma_n$ and $\sum_{i=1}^n \sigma_i = 1$, called the *spectrum*. Traditionally used in economics and finance, SRMs strike a user-defined balance between measuring average and tail loss. Recognizing that $\sum_{i=1}^n \sigma_i \ell_{(i)}(w)$ is the maximum of $\sum_{i=1}^n q_i \ell_i(w)$ for $q$ a permutation of $(\sigma_1, \ldots, \sigma_n)$, or equivalently, a convex combination of such permutations, we define the uncertainty set $\mathcal{P} = \mathcal{P}(\sigma) = \text{ConvexHull} \{\text{permutations of } \sigma\}$ to achieve the regularized formulation in (3). See Appx. B for further details on SRMs including the extremile (Daouia et al., 2019) in Equation (22) and ESRM (Cotter & Dowd, 2006) in Equation (23). SRMs are motivated not only by their historical use but also the efficiency with which we can compute their exact gradient, as we now describe.

Define $\mathcal{R}_\sigma := \mathcal{R}_{\mathcal{P}(\sigma)}$. When $\nu > 0$ and the map $q \mapsto D(q\|\mathbf{1}_n/n)$ is strongly convex over $\mathcal{P}(\sigma)$, we have that (Lem. 5, Appx. B) $\mathcal{R}_\sigma$ is in fact differentiable with gradient given by

$$\nabla \mathcal{R}_\sigma(l) = \underset{q \in \mathcal{P}(\sigma)}{\arg\max} \left\{ q^\top l - \nu D(q\|\mathbf{1}_n/n) \right\} \in \mathbb{R}^n. \tag{5}$$

This means the full-batch gradient $w \mapsto \nabla \mathcal{R}_\sigma(\ell(w)) \in \mathbb{R}^d$ can be computed by solving the inner maximization to retrieve $l \mapsto \nabla \mathcal{R}_\sigma(l) \in \mathbb{R}^n$, calling the oracles to retrieve $w \mapsto \nabla \ell(w) \in \mathbb{R}^{n \times d}$,

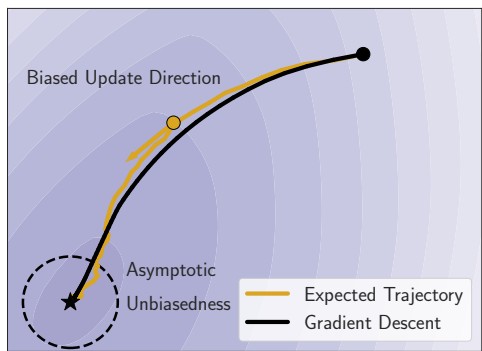 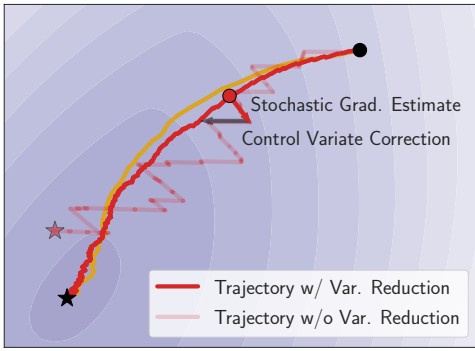

Figure 2: **Prediction Error Reduction**. Optimization trajectories on the DR objective. Darker shades indicate lower objective value. **Left (Bias):** Average trajectory of Prospect over 20 seeds compared to full-batch gradient descent. **Right (Variance):** Single trajectory of Prospect with/without adding a control variate.

and multiplying them by the chain rule. To solve for the maximizer, we prove by standard convex duality arguments (Prop. 3, Appx. B) that when $D = D_f$ is an $f$-divergence, the maximum over $q$ can be expressed as a minimization problem that reduces to isotonic regression problem involving $f^*$, the convex conjugate of $f$. Isotonic regression can be solved *exactly* by the Pool Adjacent Violators algorithm (Best et al., 2000), which runs in $O(n)$ time when the losses are sorted; see Appx. C.

**Bias Reduction via Loss Estimation.** With a formula for the gradient in hand, we proceed to estimation. Denote by $q^l := \nabla \mathcal{R}_\sigma(l)$ from (5), and observe that by the chain rule, $\nabla \mathcal{R}_\sigma(\ell(w)) = \sum_{i=1}^n q_i^{\ell(w)} \nabla \ell(w)$. In words, we compute the "most adversarial" reweighting $q^{\ell(w)} \in \mathcal{P}(\sigma)$ for a given set of losses $\ell(w)$, and then take a convex combination of the gradients $\nabla \ell_1(w), \ldots, \nabla \ell_n(w)$ weighted by the probability mass function $q^{\ell(w)}$. While the gradient is computable, however, accessing $\ell(w)$ and $\nabla \ell(w)$ requires $n$ calls to the function/gradient oracles $\{\ell_i, \nabla \ell_i\}_{i=1}^n$, which can be prohibitive. While using a plug-in estimate with a mini-batch of size $m < n$ is a natural choice in ERM (making the first term in (4) zero), this will be biased for our objective due to the maximization over $q$. For example, for $m = 1$, we have that $\mathcal{R}_\sigma(\ell_i(w)) = \ell_i(w)$ and $\nabla_w \mathcal{R}_\sigma(\ell_i(w)) = \nabla_w \ell_i(w)$, which are unbiased estimates of the ERM objective and gradient, respectively (not the SRM objective). However, note that if the optimal weights $q^{\ell(w)}$ were known, then for $i$ sampled uniformly on $[n]$, that $n q_i^{\ell(w)} \nabla \ell_i(w)$ is an unbiased estimate for $\sum_{i=1}^n q_i^{\ell(w)} \nabla \ell(w) = \nabla \mathcal{R}_\sigma(\ell(w))$. While computing $q^{\ell(w)}$ again requires computing $\ell(w)$, the key ingredient of bias reduction in Prospect is maintaining a table $l \in \mathbb{R}^n$ of losses such that $l \approx \ell(w)$ for the current iterate $w \in \mathbb{R}^d$, and using $q^l$ as a running estimate of $q^{\ell(w)}$. This is justified as when $q \mapsto D(q \| \mathbf{1}_n)$ is strongly convex, we have by Lem. 6 the map $l \mapsto q^l$ is Lipschitz continuous in $l$ with respect to $\|\cdot\|_2$. Thus,

$$l \approx \ell(w) \implies q^l \approx q^{\ell(w)} \implies \mathbb{E}_{i \sim \text{Unif}[n]} \left[ n q_i^l \nabla \ell_i(w) \right] \approx \nabla \mathcal{R}_\sigma(\ell(w)).$$

We prove in Sec. 3 that $l - \ell(w) \to 0$ as the iterate counter goes to infinity for our particular choice of $l$, yielding an asymptotically unbiased gradient estimate as illustrated in Fig. 2 (left).

**Variance Reduction via Control Variates.** The final ingredient of our stochastic gradient estimate is a variance reduction scheme. Given any estimator $\hat{a}$ of $a \in \mathbb{R}^d$, a *control variate* is another estimator $\hat{b}$ over the same probability space with a known expectation $\mathbb{E}[\hat{b}] = b \in \mathbb{R}^d$, such that $\mathbb{E}[(\hat{a} - a)^\top (\hat{b} - b)] > 0$. We can exploit this positive correlation to construct an estimator with strictly smaller variance than $\hat{a}$. Indeed, for $\gamma > 0$, we have that

$$\frac{\mathbb{E}\|\hat{a} - \gamma(\hat{b} - b) - a\|_2^2}{\mathbb{E}\|\hat{a} - a\|_2^2} = 1 - 2\gamma \frac{\mathbb{E}[(\hat{a} - a)^\top (\hat{b} - b)]}{\mathbb{E}\|\hat{a} - a\|_2^2} + o(\gamma) < 1 \text{ for small } \gamma, \tag{6}$$

demonstrating the improvement of $\hat{a} - \gamma(\hat{b} - b)$ over $\hat{a}$. Note that $\hat{a}$ need not be unbiased. In our case, we have $\hat{a} = n q_i^l \nabla \ell_i(w)$, where $l$ is the table of losses approximating $\ell(w)$. We also keep approximations $g \in \mathbb{R}^{n \times d}$ of $\nabla \ell(w)$ and $\rho$ of $q^{\ell(w)}$, and define $\hat{b} = n \rho_i g_i$ and $b = \mathbb{E}_{i \sim \text{Unif}[n]} [n \rho_i g_i] = \sum_{j=1}^n \rho_j g_j$. In the unrealistic case in which $\hat{b} = \hat{a}$, the optimal multi-

---

**Algorithm 1** Prospect

---

    **Inputs:** Initial $w_0$, spectrum $\sigma$, number of iterations $T$, regularization $\mu > 0$, shift cost $\nu > 0$.
    **Hyperparameter:** Stepsize $\eta > 0$.
1: **Initialize** $l \leftarrow \ell(w_0)$ and $g_i \leftarrow \nabla\ell_i(w_0) + \mu w_0$ for $i = 1, \ldots, n$.
2: Set $q \leftarrow \arg\max_{\bar{q} \in \mathcal{P}(\sigma)} \bar{q}^\top l - \nu D(q \| \mathbf{1}_n/n)$ and $\rho \leftarrow q$.
3: Set $\bar{g} \leftarrow \sum_{i=1}^n \rho_i g_i \in \mathbb{R}^d$.
4: Set $w \leftarrow w_0$.
5: **for** $T$ iterations **do**
6:      Sample $i, j \sim \mathrm{Unif}[n]$ independently.
7:      $v \leftarrow nq_i(\nabla\ell_i(w) + \mu w) - n\rho_i g_{i_t} + \bar{g}$.
8:      $w \leftarrow w - \eta v$.                                           ▷ **Iterate Update**
9:      $l_j \leftarrow \ell_j(w)$.
10:     $q \leftarrow \arg\max_{\bar{q} \in \mathcal{P}(\sigma)} \bar{q}^\top l - \nu D(\bar{q} \| \mathbf{1}_n/n)$.            ▷ **Bias Reducing Update**
11:     $\bar{g} \leftarrow \bar{g} - \rho_i g_i + q_i (\nabla\ell_i(w) + \mu w)$.
12:     $g_i \leftarrow \nabla\ell_i(w) + \mu w$.
13:     $\rho_i \leftarrow q_i$.                                              ▷ **Variance Reducing Update**
    **Output:** Final point $w$.

---

plier is $\gamma = 1$, trivially achieving zero variance. Similar to $l$, we prove in Sec. 3 that $g - \nabla\ell(w) \to 0$ and $\rho - q^{\ell(w)} \to 0$, so we have in the notation of (6) that $\hat{b} - \hat{a} \to 0$. Thus, by using $\gamma = 1$, our final stochastic gradient estimate is

$$\hat{a} - \gamma(\hat{b} - b) = nq_i^l \nabla\ell_i(w) - n\rho_i g_i + \sum_{j=1}^n \rho_j g_j, \tag{7}$$

with asymptotic variance reduction factor (with respect to $i \sim \mathrm{Unif}[n]$):

$$\frac{\mathrm{Var}\left[nq_i^l \nabla\ell_i(w) - n\rho_i g_i + g^\top\rho\right]}{\mathrm{Var}\left[nq_i^l \nabla\ell_i(w)\right]} \to \frac{2\mathbb{E}_{i\sim\mathrm{Unif}[n]}[(nq_i^l \nabla\ell_i(w) - \nabla\ell(w)^\top q)^\top (n\rho_i g_i - g^\top q)]}{\mathbb{E}_{i\sim\mathrm{Unif}[n]}\|nq_i^l \nabla\ell_i(w) - \nabla\ell(w)^\top q\|_2^2}.$$

This results in asymptotically vanishing variance *without* decreasing the learning rate, as illustrated in Fig. 2 (right). This scheme generalizes (and is inspired by) the one employed in the SAGA optimizer (Defazio et al., 2014) for ERM, in which $\rho = \mathbf{1}_n/n$. Finally, while ignored in this section for ease of presentation, each $g_i$ will actually store the gradients of the regularized losses $\ell_i + \mu \|\cdot\|_2^2$.

## 3 THE PROSPECT ALGORITHM

By combining the bias reduction and variance reduction schemes from the previous section, we build an algorithm that achieves overall *prediction error reduction*. Thus, we now present the **P**rediction Error-**R**educed **O**ptimizer for **Spect**ral Risk Measures (Prospect) algorithm to solve

$$\min_{w\in\mathbb{R}^d} \left[ F_\sigma(w) := \mathcal{R}_\sigma(\ell(w)) + \frac{\mu}{2} \|w\|_2^2 \right], \tag{8}$$

where $\mu > 0$ is a regularization constant. The full algorithm is given in Algorithm 1.

**Instantiating Bias and Variance Reduction.** Consider a current iterate $w \in \mathbb{R}^d$. As mentioned in Sec. 2, bias and variance reduction relies on the three approximations: the losses $l$ for $\ell(w) \in \mathbb{R}^n$, each gradient $g_i$ for $\nabla\ell_i(w) + \mu w \in \mathbb{R}^d$, and the weights $\rho$ for $q^{\ell(w)} \in \mathcal{P}$. Given initial point $w_0 \in \mathbb{R}^d$, we initialize $l = \ell(w_0)$, $g = \nabla\ell(w_0) + \mu\mathbf{1}_n w_0^\top$, and $\rho = q^{\ell(w_0)}$ (including $\bar{g} := g^\top\rho$).

At each iterate, we sample indices $i, j \sim \mathrm{Unif}[n]$ independently. The index $i$ is used to compute the stochastic gradient estimate (7), yielding the update direction $v$ in line 7 at the cost of a call to a $(\ell_i, \nabla\ell_i)$ oracle. Then, $l$ is updated by replacing $l_j$ with $\ell_j(w)$ costing another call to $(\ell_j, \nabla\ell_j)$, and we reset $q$ (the variable that stores $q^l$). Next, we use $i$ again to make the replacements of $g_i$ with $\nabla\ell_i(w) + \mu w$ and $\rho_i$ with $q_i = q_i^l$. In summary, each approximation is updated every iteration by changing one component based on the current iterate $w$. The indices $i, j$ are "decoupled" for theoretical convenience, but in practice using only $i$ works similarly, which we use in Sec. 4.

**Computational Aspects.** The weight update in Line 10 is solved exactly by (i) sorting the vector of losses in $O(n \log n)$, (ii) plugging the sorted loss table $l$ into the Pool Adjacent Violators (PAV) algorithm running in $O(n)$ time, as mentioned in Sec. 2. Because only one element of $l$ changes every iterate, we may simply bubble sort $l$ starting from the index that was changed. While in the worst case, this cost is $O(n)$, it is exactly $O(s)$ where $s$ is the number of swaps needed to resort $l$. We find in experiments that the sorted order of $l$ stabilizes quickly. The storage of the gradient table $g$ requires $O(nd)$ space in general, but it can be reduced to $O(n)$ for generalized linear models and nonlinear additive models. For losses of the form $\ell_i(w) = h(x_i^\top w, y_i)$, for a differentiable loss $h$ and scalar output $y_i$, we have $\nabla \ell_i(w) = x_i \, h'(x_i^\top w, y_i)$. We only need to store the scalar $h'(x_i^\top w, y_i)$, so Prospect requires $O(n + d)$ memory. In terms of computational complexity, Lines 8 and 13 require $O(d)$ operations and Line 10 requires at most $O(n)$ operations, so that in total the iteration complexity is $O(n + d)$. In comparison, a full batch gradient descent requires $O(nd)$ operations so Prospect decouples efficiently the cost of computing the losses, gradients, and weights.

**Convergence Analysis.** We assume throughout that each $\ell_i$ is convex, $G$-Lipschitz, and $L$-smooth. We also assume that the $D = D_f$ is an $f$-divergence with the generator $f$ being $\alpha_n$-strongly convex on the interval $[0, n]$ (e.g. $\alpha_n = 2n$ for the $\chi^2$-divergence and $\alpha_n = 1$ for the KL-divergence).

The convergence guarantees depend on the condition numbers $\kappa = 1 + L/\mu$ of the individual regularized losses, as well as a measure $\kappa_\sigma = n\sigma_n$ of the skewness of the spectrum. Note that both $\kappa$ and $\kappa_\sigma$ are necessarily larger than or equal to one. Define $w^\star := \arg\min_w F_\sigma(w)$, which exists and is unique due to the strong convexity of $F_\sigma$. The proof is given in Appx. D.6.

**Theorem 1.** *Prospect with a small enough step size is guaranteed to converge linearly for all $\nu > 0$. If, in addition, the shift cost is $\nu \geq \Omega(G^2/\mu\alpha_n)$, then the sequence of iterates $(w^{(t)})_{t \geq 1}$ generated by Prospect and learning rate $\eta = (12\mu(1 + \kappa)\kappa_\sigma)^{-1}$ converges linearly at a rate $\tau = 2\max\{n, 24\kappa_\sigma(\kappa + 1)\}$, i.e.,*

$$\mathbb{E}\|w^{(t)} - w^\star\|_2^2 \leq (1 + \sigma_n^{-1} + \sigma_n^{-2}) \exp(-t/\tau) \|w^{(0)} - w^\star\|_2^2.$$

The number of iterations $t$ required by Prospect to achieve $\mathbb{E}\|w^{(t)} - w^\star\|_2^2 \leq \varepsilon$ (provided that $\nu$ is large enough) is $t = O((n + \kappa\kappa_\sigma)\ln(1/\varepsilon))$. This exactly matches the rate of the LSVRG (Mehta et al., 2023), the only primal stochastic optimizer that converges linearly for spectral risk measures. However, unlike LSVRG, Prospect is guaranteed to converge linearly for any shift cost and has a single hyperparameter, the stepsize $\eta$. Similarly, compared to primal-dual stochastic saddle-point methods, our algorithm requires only one learning rate, streamlining its implementation.

**Prospect Variants for Non-Smooth Objectives.** We may wonder about the convergence behavior of Prospect when either the shift cost $\nu = 0$, or the underlying losses $\ell_i$ are non-smooth. While the smoothness of the objective is then lost, Prospect can still converge to the minimizer $w_0^\star$ as we prove below. The first setting is relevant as historically, SRMs such as the superquantile have been employed as coherent risk measures for loss distributions (Acerbi & Tasche, 2002) in the form of an $L$-estimator $\sum_{i=1}^n \sigma_i l_{(i)}$ (as seen in Sec. 2). If these losses are separated at the optimum, however, we may achieve linear convergence with Prospect even with $\nu = 0$, due to "hidden smoothness" of the objective, i.e. differentiability at points for which $\ell(w)$ has distinct components. Assume that each $\ell_i$ is convex and that $\mu > 0$.

**Proposition 2.** *Let $w_\nu^\star$ be the unique minimizer of (8) with shift cost $\nu \geq 0$. Assume that the values $\ell_1(w_0^\star), \ldots, \ell_n(w_0^\star)$ are all distinct. Then, there exists a constant $\nu_0 > 0$ such that $w_0^\star = w_\nu^\star$ exactly for all $\nu \leq \nu_0$. Thus, running Prospect with $\nu \in (0, \nu_0]$ converges to the minimizer $w_0^\star$.*

In particular, $\nu_0$ is chosen so that $2n\nu_0 (\sigma_{i+1} - \sigma_i) < \ell_{(i+1)}(w_0^\star) - \ell_{(i)}(w_0^\star)$ for each $i$, or as the multiplicative factor that relates gaps in the spectrum to gaps in the optimal losses (see Appx. B). When $\ell_i$ itself may be non-smooth, we generalize Prospect by applying it to the Moreau envelope of each loss $\ell_i$ and its gradient (Bauschke & Combettes, 2011; Rockafellar, 1976), allowing for losses such as those containing an $\ell_1$ penalty. Specifically, we consider oracles returning $\nabla \text{env}(\ell_i)(w)$ where $\text{env}(\ell_i)(w) := \inf_{v \in \mathbb{R}^d} \ell_i(v) + \|w - v\|_2^2$; the update steps can be expressed in terms of the proximal operators of the losses (Bauschke & Combettes, 2011). These oracles can easily be accessed either in closed form or by efficient subroutines in common machine learning settings (Defazio, 2016; Frerix et al., 2018; Roulet & Harchaoui, 2022). The resulting algorithm enjoys a linear convergence guarantee similar to Thm. 1 with a more liberal condition on the shift cost $\nu$ while providing competitive performance in practice (see Appx. E).

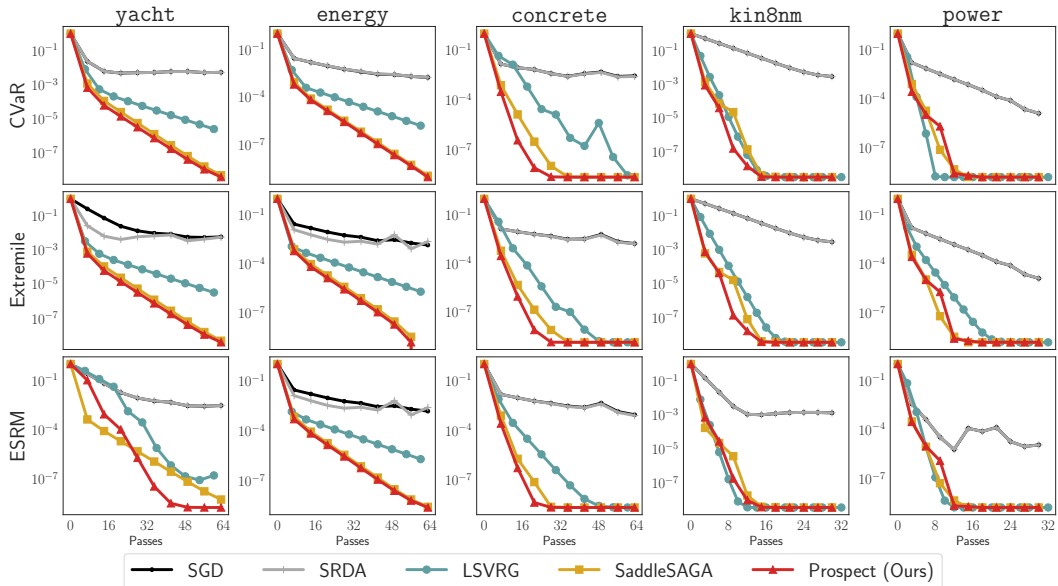

Figure 3: **Regression benchmarks**. The $y$-axis measures suboptimality as given by (9), while the $x$-axis measures the number of calls to the function value/gradient oracle divided by $n$ (i.e. passes through the training set). Rows indicate different SRM objectives while columns indicate datasets.

## 4    EXPERIMENTS

We compare Prospect against baselines in a variety of learning tasks. While we focus attention on its performance as an optimizer of its training objective, we also highlight metrics of interest on the test set in fairness and distribution shift benchmarks. The algorithm implementation and data preparation code is made publicly available online: https://github.com/ronakdm/prospect.

**Setting, Baselines, Evaluation.** We consider supervised learning tasks with input-label example $(x_i, y_i)$. Losses are of the form $\ell_i(w) := h(y_i, w^\top \phi(x_i))$, with a fixed feature embedding $\phi$, and $h$ measuring prediction loss. Uncertainty sets considered are the CVaR, extremile, and ESRM. We compare against four baselines: minibatch stochastic gradient descent (SGD), stochastic regularized dual averaging (SRDA) (Xiao, 2009), Saddle-SAGA (Palaniappan & Bach, 2016), and LSVRG (Mehta et al., 2023). For SGD and SRDA, we use a batch size of 64, and for LSVRG we use an epoch length of $n$. For Saddle-SAGA, we find that allowing different learning rates for the primal and dual variables improves theoretically and experimentally (Appx. F) and compare against an improved heuristic (setting the dual stepsize as $10n$ times smaller than the primal stepsize). We plot

$$\text{Suboptimality}(w) = \left(F_\sigma(w) - F_\sigma(w^\star)\right) / \left(F_\sigma(w^{(0)}) - F_\sigma(w^\star)\right), \tag{9}$$

where $w^\star$ is approximated by running LBFGS (Nocedal & Wright, 1999) on the objective until convergence. The $x$-axis displays the number of calls to any first-order oracle $w \mapsto (\ell_i(w), \nabla \ell_i(w))$ divided by $n$, i.e. the number of passes through the training set. We fix the shift cost $\nu = 1$ and regularization parameter $\mu = 1/n$. Further details of the setup such as hyperparameter tuning, and additional results are given in Appxs H and I respectively.

### 4.1    TABULAR LEAST-SQUARES REGRESSION

We consider five tabular regression benchmarks under square loss. The datasets used are yacht ($n = 244$) (Tsanas & Xifara, 2012), energy ($n = 614$) (Baressi Segota et al., 2020), concrete ($n = 824$) (Yeh, 2006), kin8nm ($n = 6553$) (Akujuobi & Zhang, 2017), and power ($n = 7654$) (Tüfekci, 2014). The training curves are shown in Fig. 3.

**Results.** Across datasets and objectives, we find that Prospect exhibits linear convergence at a rate no worse than SaddleSAGA and LSVRG but that is often much better. For example, Prospect

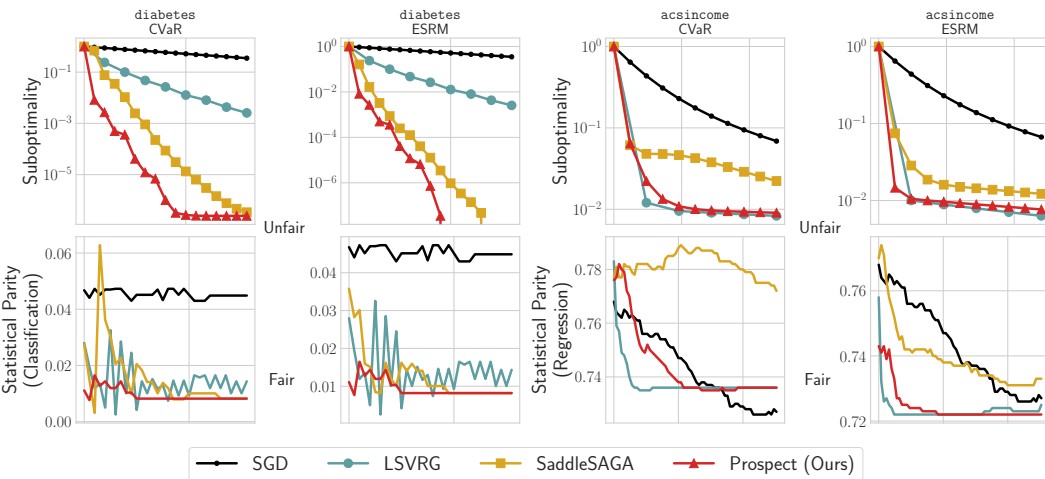

Figure 4: **Fairness benchmarks**. **Top:** Training curves for optimizers on the CVaR and extremile for `diabetes` (**left**) and CVaR and extremile for `acsincome` (**right**). **Bottom:** Statistical parity scores for the two classification objectives on `diabetes` (**left**) and regression objectives on `acsincome`. Smaller values indicate better performance for all metrics.

converges to precision $10^{-8}$ for the CVaR on `concrete` and the extremile on `power` within half the number of passes that LSVRG takes for the same suboptimality. Similarly, for the ESRM on `yacht`, SaddleSAGA requires 64 epochs to reach the same precision as Prospect at 40 epochs. The direct stochastic methods, SGD and SRDA, are biased and fail to converge for any learning rate.

## 4.2 Fair Classification and Regression

Inspired by Williamson & Menon (2019), we explore the relationship between distributional robustness and group fairness on 2 common tabular benchmarks. **Diabetes 130-Hospitals** (`diabetes`) is a classification task of predicting readmission for diabetes patients based on clinical data from US hospitals (Rizvi et al., 2014). **Adult Census** (`acsincome`) is a regression task of predicting income of US adults from data compiled by the American Community Survey (Ding et al., 2021).

**Evaluation.** We evaluate fairness with the *statistical parity score*, which compares predictive distributions of a model given different values of a particular protected attribute Agarwal et al. (2018; 2019). Letting $Z = (X, Y, A)$ denote a random (input, label, metadata attribute) triplet, a model $g$ is said to satisfy statistical parity (SP) if the conditional distribution of $g(X)$ over predictions given $A = a$ is equal for any value $a$. Intuitively, SP scores measure the maximum deviation between these distributions for any over $a$, so values close to zero indicate SP-fairness. In `diabetes`, we use gender as the protected attribute $A$, whereas in `acsincome` we use race as the protected attribute. Note that the protected attributes are not supplied to the models. Results are given in Fig. 4.

**Results.** Firstly, we note that Prospect converges rapidly on both datasets while LSVRG fails to converge on `diabetes` and SaddleSAGA fails to converge on `acsincome`. Secondly, LSVRG does not stabilize with respect to classification SP, showing a mean/std SP score of $1.38 \pm 0.25\%$ within the final ten passes on the `diabetes` CVaR, whereas Prospect gives $0.82 \pm 0.00\%$, i.e., a $40\%$ relative improvement with greater stability. While SaddleSAGA does stabilize in SP on `diabetes`, it fails to qualitatively decrease at all on the `acsincome`. Interestingly, while suboptimality and SP-fairness are correlated for Prospect, SGD (reaching only $10^{-1}$ suboptimality with respect to the CVaR objectives on `acsincome`) achieves a lower fairness score. Again, across both suboptimality and fairness, Prospect is either the best or close to the best.

## 4.3 Image and Text Classification under Distribution Shift

We consider two tasks from the WILDS distribution shift benchmark (Koh et al., 2021). The **Amazon Reviews** (`amazon`) task (Ni et al., 2019) consists of classifying text reviews of products to a

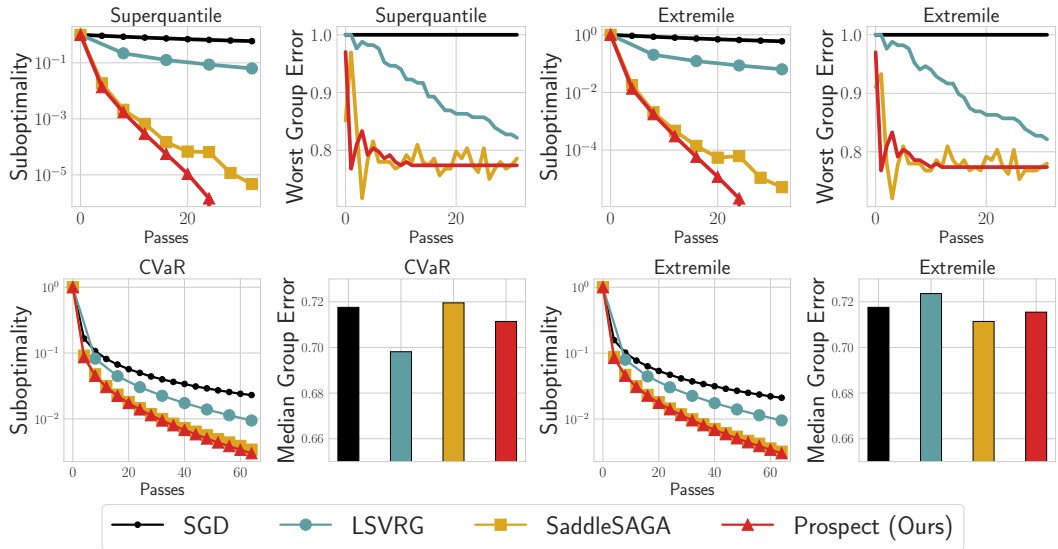

Figure 5: **Distribution shift benchmarks**. **Top row:** Training curves and worst group misclassification error on `amazon` test. **Bottom row:** Training curves and median group misclassification error on the `iwildcam` test set. Smaller values indicate better performance for all metrics.

rating of 1-5, with disjoint train and test reviewers. The **iWildCam** (`iwildcam`) image classification challenge (Beery et al., 2020) contains labeled images of animals, flora, and backgrounds from cameras placed in wilderness sites. Shifts are due to changes in camera angles, locations, lighting... We use $n = 10000$ and $n = 20000$ examples respectively. For both datasets, we train a *linear probe classifier*, i.e., a linear model over a frozen deep representation. For `amazon`, we use a pretrained BERT model (Devlin et al., 2019) fine-tuned on a held-out subset of the Amazon Reviews training set for 2 epochs. For `iwildcam`, we use a ResNet50 pretrained on ImageNet (without fine-tuning).

**Evaluation.** Apart from the training suboptimality, we evaluate the spectral risk objectives on their robustness to subpopulation shifts. We define each subpopulation group based on the true label. For `amazon`, we use the *worst group misclassification error* on the test set (Sagawa et al., 2020). For `iwildcam`, we use the *median group error* owing to its larger number of classes.

**Results.** For both `amazon` and `iwildcam`, Prospect and SaddleSAGA (with our heuristic) outperform LSVRG in training suboptimality. We hypothesize that this phenomenon is due to checkpoints of LSVRG getting stale over the $n$-length epochs for these datasets with large $n$ (leading to a slow reduction of bias). In contrast, Prospect and SaddleSAGA avoid this issue by dynamically updating the running estimates of the importance weights. For the worst group error for `amazon`, Prospect and SaddleSAGA outperform LSVRG. Prospect has a mean/std worst group error of $77.38 \pm 0.00\%$ over the last ten passes on the extremile, whereas SaddleSAGA has a slightly worse $77.53 \pm 1.57\%$. Interestingly, on `iwildcam`, LSVRG and Prospect give stronger generalization performance, nearly 1pp better, than SaddleSAGA in terms of median group misclassification rate. In summary, across tasks and objectives, Prospect demonstrates best or close to best performance.

## 5 DISCUSSION

We introduced Prospect, a distributionally robust optimization algorithm for minimizing spectral risk measures that has a linear convergence guarantee. Prospect demonstrates rapid linear convergence on benchmark examples and has the practical benefits of converging for any shift cost while only having a single hyperparameter. Promising avenues for future work include extensions to the nonconvex setting by considering the regular subdifferential, variations using other uncertainty sets, and further exploring connections to algorithmic fairness.

**Acknowledgements.** This work was supported by NSF DMS-2023166, CCF-2019844, DMS-2134012, NIH, and ODNI's IARPA program via 2022-22072200003. Part of this work was done while Z. Harchaoui was visiting the Simons Institute for the Theory of Computing.

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

# Appendix

The Appendix sections are organized as follows. We summarize notation in Appx. A and provide intuition and results regarding the primal/dual objective function in Appx. B. We describe in detail efficient implementations of the proposed algorithm in Appx. C. In Appx. D, we describe the convergence analyses of the main algorithm. In Appx. E and Appx. F, we describe an Moreau envelope-based variant of our method and an improved version of an existing saddle-point method, respectively. Appx. G contains technical results shared to multiple proofs. We then describe the experimental setup in detail in Appx. H and give additional results in Appx. I.

## Table of Contents

# A   SUMMARY OF NOTATION

We collect the notation used throughout the paper in Tab. 1.

| Symbol | Description |
|---|---|
| $\mu \geq 0$ | Standard regularization constant. |
| $\nu \geq 0$ | Shift cost. |
| $\alpha_n$ | Strong convexity constant for any $f$ generating an $f$-divergence. |
| $\bar{\nu}$ | Shorthand $\bar{\nu} = n\alpha_n\nu$ (used in the convergence proofs). |
| $\ell_1(w), \ldots, \ell_n(w)$ | Loss functions $\ell_i : \mathbb{R}^d \to \mathbb{R}$. |
| $\ell(w)$ | Vector of losses $\ell(w) = (\ell_1(w), \ldots, \ell_n(w))$ for $w \in \mathbb{R}^d$. |
| $r_i(w)$ | Regularized loss $r_i(w) = \ell_i(w) + \frac{\mu}{2}\|w\|_2^2$. |
| $r(w)$ | Vector of regularized losses $r(w) = (r_1(w), \ldots, r_n(w))$. |
| $\nabla\ell(w)$ | Jacobian matrix of $\ell : \mathbb{R}^d \to \mathbb{R}^n$ at $w$ (shape $= n \times d$). |
| $\sigma$ | The vector $\sigma = (\sigma_1, \ldots, \sigma_n) \in [0,1]^n$ where each $\sigma_1 \leq \ldots \leq \sigma_n$ and they sum to 1. |
| $\mathcal{P}(\sigma)$ | The set $\{\Pi\sigma \,:\, \Pi \in [0,1]^{n \times n}, \Pi\mathbf{1}_n = \mathbf{1}_n, \Pi^\top\mathbf{1}_n = \mathbf{1}_n\}$, known as the permutahedron. |
| $f$ | Convex function $f : [0, \infty) \to \mathbb{R} \cup \{+\infty\}$ generating an $f$-divergence. |
| $f^*$ | Convex conjugate $f^*(y) := \sup_{x \in \mathbb{R}} \{xy - f(x)\}$. |
| $\Omega_f$ or $\Omega$ | Shift penalty function $\Omega_f : \mathcal{P}(\sigma) \mapsto [0, \infty)$. We consider $f$-divergence penalties $\Omega_f(q) = D_f(q\|\mathbf{1}_n/n)$. |
| $F_\sigma$ | Main objective $F_\sigma(w) = \max_{q \in \mathcal{P}(\sigma)} \{q^\top\ell(w) - \nu D_f(q\|\mathbf{1}_n/n)\} + \frac{\mu}{2}\|w\|_2^2$. |
| $q^{\text{opt}}(l)$ or $q^l$ | Most unfavorable reweighting for a given vector $l$ of losses, i.e., $q^{\text{opt}}(l) = \arg\max_{q \in \mathcal{P}(\sigma)} q^\top l - \nu D(q\|\mathbf{1}_n/n)$. $q^l$ used only in main text for readability. |
| $w^\star$ | Optimal weights $\arg\min_{w \in \mathbb{R}^d} \max_{q \in \mathcal{P}(\sigma)} q^\top l - \nu D(q\|\mathbf{1}_n/n) + (\mu/2)\|w\|_2^2$. |
| $q^\star$ | Most unfavorable reweighting of $\ell(w^\star)$, i.e., $q^\star = q^{\text{opt}}(\ell(w^\star))$ |
| $G$ | Lipschitz constant of each $\ell_i$ w.r.t. $\|\cdot\|_2$. |
| $L$ | Lipschitz constant of each $\nabla\ell_i$ w.r.t. $\|\cdot\|_2$. |
| $M$ | $M = L + \mu$, the Lipschitz constant of each $\nabla r_i$ w.r.t. $\|\cdot\|_2$. |
| $\mathbb{E}_t[\cdot]$ | Shorthand for $\mathbb{E}\left[\cdot\,\middle|\,w^{(t)}\right]$, i.e., expectation conditioned on $w^{(t)}$. |

Table 1: Notation used throughout the paper.

# B   PROPERTIES OF THE PRIMAL AND DUAL OBJECTIVES

Recall that we are interested in the optimization problem

$$\min_{w \in \mathbb{R}^d} \left[ F_\sigma(w) := \max_{q \in \mathcal{P}(\sigma)} q^\top\ell(w) - \nu D_f(q\|\mathbf{1}_n/n) + \frac{\mu}{2}\|w\|_2^2 \right], \tag{10}$$

where $D_f(q\|\mathbf{1}_n/n)$ denotes an $f$-divergence between the distribution associated to the reweighting $q$ and the discrete uniform weights $\mathbf{1}_n/n = (1/n, \ldots, 1/n)$ and $\mathcal{P}(\sigma)$ is the spectral risk measure uncertainty set.

The first goal for this section will be to derive properties of the function $F_\sigma(w)$, or the *primal objective*, as well as the inner maximization problem, which we refer to as the *dual objective*. Both will be useful in motivating and analyzing Prospect (used for the primal minimization) and various subroutines used to compute the maximally unfavorable reweighting (i.e., the maximizer over $q$ in

the inner maximization). These properties do not depend on the structure of the $\mathcal{P}(\sigma)$ itself, only that it is a closed, convex set. The second goal of the section is to provide additional background on the choice of $\mathcal{P}(\sigma)$ from a statistical modeling perspective. The uncertainty set $\mathcal{P}(\sigma)$ is then described further from a computational perspective in Appx. C.

**Review of $f$-Divergences.** Let $q$ and $p$ be any two probability mass functions defined on atoms $\{1, \ldots, n\}$. Consider a convex function $f : [0, \infty) \mapsto \mathbb{R} \cup \{+\infty\}$ such that $f(1) = 0$, $f(x)$ is finite for $x > 0$, and $\lim_{x \to 0^+} f(x) = 0$. The $f$-*divergence* from $q$ to $p$ generated by this function $f$ is

$$D_f(q\|p) := \sum_{i=1}^{n} f\left(\frac{q_i}{p_i}\right) p_i,$$

where we define $0f(0/0) := 0$ in the formula above. If there is an $i$ such that $p_i = 0$ but $q_i > 0$, we say $D_f(q\|p) = \infty$. The $\chi^2$-divergence is generated by $f_{\chi^2}(x) = x^2 - 1$ and the KL divergence is generated by $f_{\mathrm{KL}}(x) = x \ln x + \iota_+(x)$ where $\iota_+$ denotes the convex indicator that is zero for $x \geq 0$ and $+\infty$ otherwise, and we define $x \ln x = 0$ for all $x < 0$.

**The Dual Problem.** We describe the inner maximization first, that is

$$\max_{q \in \mathcal{P}(\sigma)} \left\{ q^\top l - \nu D_f(q\|\mathbf{1}_n) \right\}. \tag{11}$$

Its properties will inform the algorithmic implementation for the minimization over $w$ in (10). In the case of an $f$-divergence between $q$ and the uniform weights $\mathbf{1}_n/n$, we have

$$D_f(q\|\mathbf{1}_n/n) := \frac{1}{n} \sum_{i=1}^{n} f(nq_i). \tag{12}$$

We now derive the dual problem to Equation (11). This will lead to an algorithm to solve the optimization problem efficiently. Throughout, we denote $f^*(y) := \sup_{x \in \mathbb{R}} \{xy - f(x)\}$ as the convex conjugate of $f$.

We consider the following functions whose conjugates are strictly convex. Recall that if $f$ is smooth, i.e., with Lispchtiz continuous gradients, then its conjugate is strongly convex, hence strictly convex. More generally $f^*$ is strictly convex if $f$ is convex and essentially smooth, that is, with gradient norm tending to $+\infty$ at its boundaries, see e.g. (Rockafellar, 1976) for a detailed presentation. For simple cases such as the $\chi^2$ or KL divergence presented, strict convexity of the convex conjugate is naturally satisfied:

$$f_{\chi^2}(x) = x^2 - 1 \text{ and } f_{\chi^2}^*(y) = y^2/4 + 1 \qquad \qquad (\chi^2\text{-divergence})$$

$$f_{\mathrm{KL}}(x) = x \ln x + \iota_+(x) \text{ and } f_{\mathrm{KL}}^*(y) = \exp(y - 1). \qquad (\text{KL-divergence})$$

**Proposition 3.** *Let $l \in \mathbb{R}^n$ be a vector and $\pi$ be a permutation that sorts its entries in non-decreasing order, i.e., $l_{\pi(1)} \leq \ldots \leq l_{\pi(n)}$. Consider a function $f$ strictly convex with strictly convex conjugate defining a divergence $D_f$. Then, the maximization over the permutahedron subject to the shift penalty can be expressed as*

$$\max_{q \in \mathcal{P}(\sigma)} \left\{ q^\top l - \nu D_f(q\|\mathbf{1}_n/n) \right\} = \min_{\substack{c \in \mathbb{R}^n \\ c_1 \leq \ldots \leq c_n}} \sum_{i=1}^{n} g_i(c_i ; l), \tag{13}$$

*where we define $g_i(c_i ; l) := \sigma_i c_i + \frac{\nu}{n} f^*\left(\frac{l_{\pi(i)} - c_i}{\nu}\right)$. The optima of both problems, denoted*

$$c^{opt}(l) = \arg\min_{\substack{c \in \mathbb{R}^n \\ c_1 \leq \ldots \leq c_n}} \sum_{i=1}^{n} g_i(c_i; l), \; q^{opt} = \arg\max_{q \in \mathcal{P}(\sigma)} q^\top l - \nu D_f(q\|\mathbf{1}_n/n),$$

*are related as $q^{opt}(l) = \nabla(\nu D_f(\cdot\|\mathbf{1}_n/n))^*(l - c_{\pi^{-1}}^{opt}(l))$, that is,*

$$q_i^{opt}(l) = \frac{1}{n} [f^*]'\left(\frac{1}{\nu}(l_i - c_{\pi^{-1}(i)}^{opt}(l))\right). \tag{14}$$

*Proof.* Let $\iota_{\mathcal{P}(\sigma)}$ denote the indicator function of the permutahedron $\mathcal{P}(\sigma)$, which is $0$ inside $\mathcal{P}(\sigma)$ and $+\infty$ outside of $\mathcal{P}(\sigma)$. Its convex conjugate is the support function of the permutahedron, i.e.,

$$\iota_{\mathcal{P}(\sigma)}^*(l) = \max_{q \in \mathcal{P}(\sigma)} q^\top l.$$

For two closed convex functions $h_1$ and $h_2$ that are bounded from below, the convex conjugate of their sum is the infimal convolution of their conjugate (Hiriart-Urruty & Lemaréchal, 2004, Proposition 6.3.1):

$$(h_1 + h_2)^*(x) = \inf_{y \in \mathbb{R}^d} \{h_1^*(y) + h_2^*(x - y)\} .$$

Provided that $h_1 + h_2$ is strictly convex, we have that the maximizer defining the conjugate is unique and equal to the gradient, that is,

$$\nabla(h_1 + h_2)^*(x) = \arg\max_{z \in \mathbb{R}^d} \left\{ z^\top x - (h_1 + h_2)(z) \right\} .$$

If, in addition, $h_1^* + h_2^*$ is strictly convex and $h_2^*$ is differentiable, we have, by Danskin's theorem (Bertsekas, 1997),

$$\nabla(h_1 + h_2)^*(x) = \nabla h_2^*(x - y^\star(x)) \text{ for } y^\star(x) = \arg\min_{y \in \mathbb{R}^d} \{h_1^*(y) + h_2^*(x - y)\} .$$

Consider then $h_1(q) = \iota_{\mathcal{P}(\sigma)}(q)$ and $h_2(q) = \Omega_f(q) := \nu D_f(q \| \mathbf{1}_n / n)$. Provided that $f$ is strictly convex with $f^*$ strictly convex, $D_f$ is also strictly convex with $D_f^*$ strictly convex since $D_f$ just decomposes as a sum of $f$ on independent variables. We have then

$$\sup_{q \in \mathcal{P}(\sigma)} \left\{ q^\top l - \Omega_f(q) \right\} = \sup_{q \in \mathbb{R}^n} \left\{ q^\top l - (\iota_{\mathcal{P}(\sigma)}(q) + \Omega_f(q)) \right\}$$

$$= (\iota_{\mathcal{P}(\sigma)} + \Omega_f)^*(l)$$

$$= \inf_{y \in \mathbb{R}^n} \left\{ \iota_{\mathcal{P}(\sigma)}^*(y) + \Omega_f^*(l - y) \right\}$$

$$= \inf_{y \in \mathbb{R}^n} \left\{ \max_{q \in \mathcal{P}(\sigma)} q^\top y + \Omega_f^*(l - y) \right\}$$

$$= \inf_{y \in \mathbb{R}^n} \left\{ \sum_{i=1}^n \sigma_i y_{(i)} + \Omega_f^*(l - y) \right\}, \tag{15}$$

where $y_{(1)} \leq \ldots \leq y_{(n)}$ are the ordered values of $y \in \mathbb{R}^n$. Moreover we have that

$$\arg\max_{q \in \mathcal{P}(\sigma)} \left\{ q^\top l - \Omega_f(q) \right\} = \nabla \Omega_f^*(l - y^\star(l)) \text{ for } y^\star(l) = \arg\min_{y \in \mathbb{R}^n} \left\{ \sum_{i=1}^n \sigma_i y_{(i)} + \Omega_f^*(l - y) \right\} .$$

Since for any $x \in \mathbb{R}^n$, $\Omega_f$ is decomposable into a sum of identical functions evaluated at the coordinates $(x_1, \ldots, x_n)$, that is, $\Omega_f(x) = \sum_{i=1}^n \omega(x_i)$, its convex conjugate is $\Omega_f^*(y) = \sum_{i=1}^n \omega^*(y_i)$. In our case, $\omega(x_i) = \frac{\nu}{n} f(n x_i)$ from Equation (12), so $\omega^*(y_i) = (\nu/n) f^*(y_i/\nu)$.

Next, by convexity of $\omega^*$, we have that if for scalars $l_i, l_j, y_i, y_j$ such that $l_i \leq l_j$ and $y_i \geq y_j$, then using Lem. 33, we have that

$$\omega^*(l_i - y_i) + \omega^*(l_j - y_j) \geq \omega^*(l_i - y_j) + \omega^*(l_j - y_i).$$

Hence for $y$ to minimize $\Omega_f^*(l - y) = \sum_{i=1}^n \omega^*(l_i - y_i)$, the coordinates of $y$ must be ordered as $l$. That is, if $\pi$ is an argsort for $l$, s.t. $l_{\pi(1)} \leq \ldots \leq l_{\pi(n)}$, then $y_{\pi(1)} \leq \ldots \leq y_{\pi(n)}$. Since $\iota_{\mathcal{P}(\sigma)}^*(y) = \sum_{i=1}^n \sigma_i y_{(i)}$ does not depend on the ordering of $y$, the solution of (15) must also be

ordered as $l$ such that the dual problem (15) can be written as

$$\inf_{\substack{y \in \mathbb{R}^n \\ y_{\pi(1)} \leq \ldots \leq y_{\pi(n)}}} \sum_{i=1}^n \sigma_i y_{\pi(i)} + \frac{\nu}{n} \ f^* \left( \frac{l_{\pi(i)} - y_{\pi(i)}}{\nu} \right) = \inf_{\substack{c \in \mathbb{R}^n \\ c_1 \leq \ldots \leq c_n}} \sum_{i=1}^n \sigma_i c_i + \frac{\nu}{n} \ f^* \left( \frac{l_{\pi(i)} - c_i}{\nu} \right)$$

$$= \min_{\substack{c \in \mathbb{R}^n \\ c_1 \leq \ldots \leq c_n}} \sum_{i=1}^n g_i(c_i; l),$$

where we used a change of variables such that the solutions of the left and right hand sides are related as $y^\star(l) = c^\star_{\pi^{-1}}(l)$. $\square$

While strict convexity of the function $f$ ensures naturally the existence of the maximizer $q^{\text{opt}}$ defined above, this does not help quantify the continuity of the maximizer with respect to the given vector of losses. For that, we need to consider strong convexity of the $f$-divergence on the maximization set. The following proposition simply links the strong convexity of $f$ to the strong convexity of the associated weighted divergence in $\ell_2$ norm.

**Proposition 4.** *Assume that $f : \mathbb{R} \to \mathbb{R}$ is $\alpha_n$-strongly convex on $[0, n]$. Then, $q \mapsto \nu D_f(q \| \mathbf{1}_n / n)$ is $(\nu n \alpha_n)$-strongly convex with respect to $\|\cdot\|_2$.*

*Proof.* Due to the $\alpha_n$-strong convexity of $f$, for any $q, \rho \in [0, 1]^n$ and any $\theta \in (0, 1)$ and any $i \in [n]$,

$$f \left( \theta n q_i + (1 - \theta) n \rho_i \right) \leq \theta f(n q_i) + (1 - \theta) f(n \rho_i) - \frac{\alpha_n}{2} \theta (1 - \theta)(n q_i - n \rho_i)^2.$$

We average this inequality over $i$, yielding

$$\frac{1}{n} \sum_{i=1}^n f \left( n(\theta q_i + (1 - \theta) \rho_i) \right) \leq \theta \frac{1}{n} \sum_{i=1}^n f(n q_i) + (1 - \theta) \frac{1}{n} \sum_{i=1}^n f(n \rho_i) - \frac{\alpha_n}{2} \theta (1 - \theta) \| n q_i - n \rho_i \|^2.$$

Defining $\Omega_f(q) := D_f(q \| \mathbf{1}_n / n)$, the statement above can be succinctly written as

$$\Omega_f(\theta q + (1 - \theta) \rho) \leq \theta \Omega_f(q) + (1 - \theta) \Omega_f(\rho) - \frac{\alpha_n n}{2} \theta (1 - \theta) \| q_i - \rho_i \|^2.$$

Therefore, $\Omega_f$ is $(\alpha_n n)$-strongly convex with respect to $\|\cdot\|_2$ on $[0, 1]^n$, so $q \mapsto \nu D_f(q \| \mathbf{1}_n / n)$ is $(\nu n \alpha_n)$-strongly convex. $\square$

The Pool Adjacent Violators (PAV) algorithm is designed exactly for the minimization (13). The algorithm is described for the $\chi^2$-divergence and KL-divergence with implementation steps in Appx. C. Both the argsort $\pi$ and the inverse argort $\pi^{-1}$ are mappings from $[n] = \{1, \ldots, n\}$ onto itself, but the interpretation of these indices are different for the input and output spaces $[n]$. The argsort $\pi$ can be thought of as an *index finder*, in the sense that for a vector $l \in \mathbb{R}^n$, because $l_{\pi(1)} \leq \ldots \leq l_{\pi(n)}$, $\pi(i)$ can be interpreted as the index of an element of $l$ which achieves the rank $i$ in the sorted vector. On the other hand, $\pi^{-1}(i)$ can be thought of as a *rank finder*, in that $\pi^{-1}(i) = \text{rank}(i)$ is the position that $l_i$ takes in the sorted form of $l$. To summarize:

$$\pi : \quad \underbrace{[n]}_{\text{ranks of losses}} \quad \to \quad \underbrace{[n]}_{\text{indices of training examples}} \qquad \text{while } \pi^{-1} : \quad \underbrace{[n]}_{\text{indices of training examples}} \quad \to \quad \underbrace{[n]}_{\text{ranks of losses}}$$

We may equivalently write (14) as

$$q_i^{\text{opt}}(l) = \frac{1}{n} [f^*]' \left( \frac{1}{\nu}(l_i - c_{\text{rank}(i)}^{\text{opt}}(l)) \right). \tag{16}$$

Finally, as seen in Appx. C, it will be helpful to compute $q^{\text{opt}}$ in sorted order. Because the $f$-divergence is agnostic to the ordering of the $q$ vector (as it is being compared to the uniform weights),

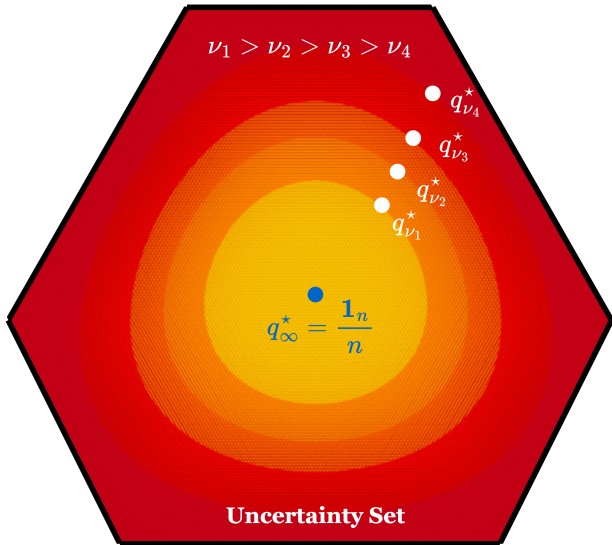

Figure 6: **Visualization of dual solution for varying shift cost**. The hexagon represents a three-dimensional uncertainty set (see Fig. 7) whereas the white dots signify solutions of (11) for a fixed loss vector and varying shift cost $\nu$. As $\nu$ increases, the solution tends closer to the uniform vector $\mathbf{1}_n/n$, whereas as $\nu$ decreases, the shifted distribution is allowed to drift closer to the boundary of the uncertainty set.

$q$ can also be sorted by $\pi$. Thus, we may also write

$$q^{\text{opt}}_{(i)}(l) = \frac{1}{n}[f^*]' \left( \frac{1}{\nu}(l_{(i)} - c_i^{\text{opt}}(l)) \right). \tag{17}$$

The effect of varying the shift cost $\nu$ on the solution of (11) is visualized in Fig. 6.

**The Primal Function.** When the divergence generator $f$ is strongly convex and the loss function $\ell : \mathbb{R}^d \to \mathbb{R}^n$ is convex and differentiable, we have that (10) is differentiable, as we show next.

**Lemma 5.** *When the map $q \mapsto \nu D(q\|\mathbf{1}_n/n)$ is strongly convex over $\mathcal{P}(\sigma)$, then $\mathcal{R}_\sigma$ is continuously differentiable with gradient given by*

$$\nabla \mathcal{R}_\sigma(l) = \underset{q \in \mathcal{P}(\sigma)}{\arg\max} \left\{ q^\top l - \nu D(q\|\mathbf{1}_n/n) \right\} \in \mathbb{R}^n.$$

*Proof.* Because $\mathcal{P}(\sigma)$ as defined in (20) is closed and convex, the strongly concave function $q \mapsto q^\top l - \nu D(q\|\mathbf{1}_n/n)$ has a unique maximizer. Because $\mathcal{P}(\sigma)$ is closed subset of the compact set $\Delta^n$, it is also compact. By Danskin's theorem (Bertsekas, 1997, Proposition B.25), we have that $F_\sigma$ is continuously differentiable with the given gradient formula. $\square$

**Lemma 6.** *Let $\ell : \mathbb{R}^d \to \mathbb{R}^n$ be differentiable with Jacobian $w \mapsto \nabla \ell(w) \in \mathbb{R}^{n \times d}$. Let each $\ell_i : \mathbb{R}^d \to \mathbb{R}$ be convex. Assume $\nu > 0$. Let $f$ be $\alpha_n$-strongly convex on the interval $[0, n]$. Then, the function $F_\sigma$ from Equation (10) is differentiable with its gradient equal to*

$$\nabla F_\sigma(w) = (\nabla \ell(w))^\top q^{opt}(\ell(w)) + \mu w.$$

*Furthermore $l \mapsto q^{opt}(l)$ is $(\alpha_n n \nu)^{-1}$-Lipschitz continuous w.r.t. $\|\cdot\|_2$.*

*Proof.* First, by Prop. 4, we have that $q \mapsto \nu D_f(q\|\mathbf{1}_n)$ is $(\nu \alpha_n n)$-strongly convex with respect to $\|\cdot\|_2$ on $[0, 1]^n$. Next, due to the convexity of each $\ell_i$ and the non-negativity of any $q \in \mathcal{P}(\sigma)$, we have that

$$w \mapsto \max_{q \in \mathcal{P}(\sigma)} \left\{ q^\top \ell(w) - \nu \Omega_f(q) \right\}$$

is convex, as is its pointwise maximum (over $q$) of a family of convex functions $q^\top \ell(w)$. We have by Lem. 5 that $F_\sigma$ is continuously differentiable with

$$\nabla F_\sigma(w) = \nabla \ell(w)^\top q^{\mathrm{opt}}(\ell(w)) + \mu w \,.$$

Moreover, by Nesterov (2005, Theorem 1), we have that $l \mapsto q^{\mathrm{opt}}(l)$ is Lipschitz continuous with Lipschitz constant equal to the inverse of the strong convexity constant of $\nu \Omega_f$, which is $\nu \alpha_n n$. $\qquad \square$

Returning to our canonical examples, we have that for the $\chi^2$, $f_{\chi^2}(x) = x^2 - 1$ is 2-strongly convex on $\mathbb{R}$ and that $f_{\mathrm{KL}}(x) = x \ln x$ is $(1/n)$-strongly convex on $[0, n]$. Thus, the function $l \mapsto q^{\mathrm{opt}}(l)$ will have Lipschitz constant $2n\nu$ and $\nu$, respectively.

**Smoothness Properties.** By applying Lem. 6 to Lipschitz continuous losses, we may achieve the following guarantee regarding the changes in $q^{\mathrm{opt}}$ with respect to $w$.

**Lemma 7.** *Let $f$ be $\alpha_n$-strongly convex on the interval $[0, n]$. For any $w_1, \ldots, w_n, w_1', \ldots, w_n' \in \mathbb{R}^d$ construct $\bar{\ell}(w_1, \ldots, w_n) = \left( \ell_i(w_i) \right)_{i=1}^n \in \mathbb{R}^n$, as well as $\bar{\ell}(w_1', \ldots, w_n')$ where each $\ell_i$ is $G$-Lipschitz w.r.t. $\|\cdot\|_2$. Then, we have*

$$\left\| q^{opt}(\bar{\ell}(w_1, \ldots, w_n)) - q^{opt}(\bar{\ell}(w_1', \ldots, w_n')) \right\|_2^2 = \frac{G^2}{n^2 \alpha_n^2 \nu^2} \sum_{i=1}^n \|w_i - w_i'\|_2^2 \,.$$

*Proof.* By the Lipschitz property of $q^{\mathrm{opt}}$ (Lem. 6), we have,

$$\left\| q^{\mathrm{opt}}(\bar{\ell}(w_1, \ldots, w_n)) - q^{\mathrm{opt}}(\bar{\ell}(w_1', \ldots, w_n')) \right\|_2^2 \le \frac{1}{n^2 \alpha_n^2 \nu^2} \left\| \bar{\ell}(w_1, \ldots, w_n) - \bar{\ell}(w_1', \ldots, w_n') \right\|_2^2$$

$$\le \frac{1}{n^2 \alpha_n^2 \nu^2} \sum_{i=1}^n \left( \ell_i(w_i) - \ell_i(w_i') \right)_2^2$$

$$\le \frac{G^2}{n^2 \alpha_n^2 \nu^2} \sum_{i=1}^n \|w_i - w_i'\|_2^2 \,.$$

$\qquad \square$

As a special case of Lem. 7, we may consider $w_1 = \cdots = w_n = w \in \mathbb{R}^d$ and $w_1' = \cdots = w_n' = w' \in \mathbb{R}^d$, in which case the result reads

$$\left\| q^{\mathrm{opt}}(\ell(w)) - q^{\mathrm{opt}}(\ell(w')) \right\|_2^2 = \frac{G^2}{n \alpha_n^2 \nu^2} \|w - w'\|_2^2 \,.$$

**Properties under No Shift Penalty.** Next, we use the smoothness properties above to prove Prop. 2 by virtue of the following proposition, which states the equivalence of the minimizers of "no-cost" and "low-cost" objectives.

**Proposition 8.** *Let $w_\nu^\star$ be the unique minimizer of (8) with shift cost $\nu \ge 0$ and $\chi^2$-divergence penalty. Define $\ell_{(1)}(w_0^\star) < \ldots < \ell_{(n)}(w_0^\star)$ to be the order statistics of $\ell_1(w_0^\star), \ldots, \ell_n(w_0^\star)$, which are assumed to be distinct. Consider $\nu_0$ such that*

$$n\nu_0 \left( \sigma_{i+1} - \sigma_i \right) < \ell_{(i+1)}(w_0^\star) - \ell_{(i)}(w_0^\star) \text{ for } i = 1, \ldots, n. \tag{18}$$

*We have that $w_0^\star = w_\nu^\star$ for all $\nu \le \nu_0$.*

*Proof.* For a vector $l \in \mathbb{R}^n$ and $\nu \geq 0$, consider

$$
\begin{aligned}
h_\nu(l) &:= \max_{q \in \mathcal{P}(\sigma)} q^\top l - \nu n \|q - \mathbf{1}_n/n\|_2^2 \\
&= \max_{q \in \mathcal{P}(\sigma)} q^\top (l + 2\nu \mathbf{1}_n) - \nu n \|q\|_2^2 - (\nu/n) \|\mathbf{1}_n\|_2^2 \\
&= \max_{q \in \mathcal{P}(\sigma)} q^\top l - \nu n \|q\|_2^2 + \nu \\
&:= g_\nu(l) + \nu,
\end{aligned}
$$

where we used that $q^\top \mathbf{1} = 1$ for all $q \in \mathcal{P}(\sigma)$. For $\nu > 0$, by Danskin's theorem (Bertsekas, 1997, Proposition B.25),

$$
\nabla h_\nu(l) = \nabla g_\nu(l) = \arg\max_{q \in \mathcal{P}(\sigma)} q^\top l - \nu n \|q\|_2^2 = \arg\max_{q \in \mathcal{P}(\sigma)} q^\top (l/2n\nu) - \frac{1}{2} \|q\|_2^2 .
$$

Without loss of generality, assume that $l$ is sorted, so that by applying the duality given by Prop. 3 and that $f^*(t) = t^2/4 + 1$, we have that

$$
\begin{aligned}
c^{\mathrm{opt}}(l) &= \arg\min_{\substack{c \in \mathbb{R}^n \\ c_1 \leq \ldots \leq c_n}} \sigma_i c_i + \frac{\nu}{4n} \left( \frac{l_i - c_i}{\nu} \right)^2 \\
q_i^{\mathrm{opt}}(l) &= \frac{1}{2n\nu} (l_i - c_i^{\mathrm{opt}}(l))
\end{aligned}
$$

By differentiating with respect to $c$, we have that if unconstrained, $c_i^{\mathrm{opt}}(l) = l_i + 2n\nu\sigma_i$ is the primal solution and the dual solution is given by $q_i^{\mathrm{opt}}(l) = \sigma_i$, which is equal to the gradient of $l \mapsto g_0(l)$ when $l$ has distinct elements. Thus, we derive a condition under which $c_i^{\mathrm{opt}}(l) = l_i + 2n\nu\sigma_i$ is monotonically non-decreasing (i.e. has the same sorted order as $l$), which will be true if $\nu$ is small enough. Specifically, we have that if

$$
2n\nu_0 (\sigma_{i+1} - \sigma_i) < \ell_{(i+1)}(w_0^\star) - \ell_{(i)}(w_0^\star) \text{ for } i = 1, \ldots, n, \tag{19}
$$

for some $\nu_0 > 0$, then $c_i^{\mathrm{opt}}(l) = l_i + 2n\nu\sigma_i$ is monotonically non-decreasing. Consequently, for any $\nu \leq \nu_0$,

$$
\nabla g_\nu(\ell(w_0^\star)) = \nabla g_0(\ell(w_0^\star)).
$$

Denote our objective as

$$
\mathcal{L}_{\sigma,\nu}(w) = h_\nu(\ell(w)) + \frac{\mu}{2} \|w\|_2^2 ,
$$

where we explicitly show the dependence on the shift cost $\nu \geq 0$. For $\nu = 0$, since the losses are differentiable and $\ell(w_0^\star)$ is composed of distinct coordinates, $\mathcal{L}_{\sigma,0}$ is differentiable at $w_0^\star$ with gradient $\nabla\ell(w_0^\star)^\top \nabla h_0(\ell(w_0^\star)) + \mu w_0^\star$ (Mehta et al., 2023, Proposition 2), where $\nabla\ell(w_0^\star) \in \mathbb{R}^{n \times d}$ denotes the Jacobian of $\ell$ at $w_0^\star$. Using the chain rule, we successively deduce

$$
\begin{aligned}
\nabla\mathcal{L}_{\sigma,0}(w_0^\star) = 0 &\iff \nabla\ell(w_0^\star)^\top \nabla h_0(\ell(w_0^\star)) + \mu w_0^\star = 0 \\
&\iff \nabla\ell(w_0^\star)^\top \nabla g_0(\ell(w_0^\star)) + \mu w_0^\star = 0 \\
&\iff \nabla\ell(w_0^\star)^\top \nabla g_\nu(\ell(w_0^\star)) + \mu w_0^\star = 0 \\
&\iff \nabla\ell(w_0^\star)^\top \nabla h_\nu(\ell(w_0^\star)) + \mu w_0^\star = 0 \\
&\iff \nabla\mathcal{L}_{\sigma,\nu}(w_0^\star) = 0.
\end{aligned}
$$

Applying the first-order optimality conditions of $\mathcal{L}_{\sigma,0}$ and $\mathcal{L}_{\sigma,\nu}$, as well as the uniqueness of $w_0^\star$ completes the proof. $\square$

Prop. 2 of the main paper then follows by combining Prop. 8 above with the convergence guarantee Thm. 1 of Prospect. Indeed, Thm. 1 shows that Prospect is able to converge linearly for arbitrarily small $\nu > 0$ and as long as $\nu \leq \nu_0$. Under Prop. 8, the minimizer will be equal to $w_0^\star$.

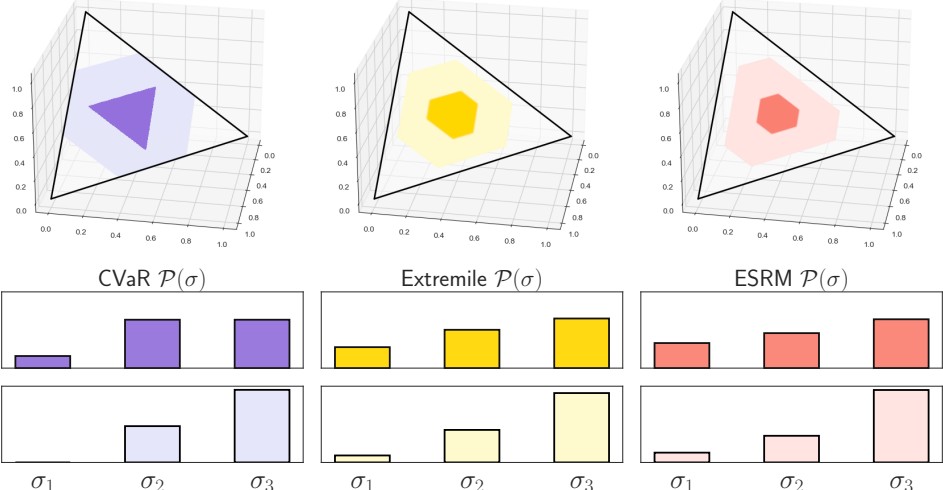

Figure 7: **Visualization of uncertainty sets**. **Top:** Illustration of the permutahedra $\mathcal{P}(\sigma)$ within the three-dimensional probability simplex for the CVaR (21) (**top left**) $p = 0.25$ (light purple) and $p = 0.5$ (light purple), the extremile (22) (**top center**) with $b = 2.5$ (light yellow) and $b = 1.5$ (dark yellow), and the ESRM (23) (**top right**) with $\gamma = 3$ (light pink) and $\gamma = 1$ (dark pink). The size of $\mathcal{P}(\sigma)$ increases for more non-uniform spectra $\sigma$. Each point within the colored shapes represents a vector of probabilities $q$ over $n = 3$ samples. The vertices of the polytopes are various permutations of the $\sigma$ vectors, whereas the interior of the polytopes are elements of the convex hull of these permutations. **Bottom:** The $\sigma$ associated to each colored shape is given by the same-colored bars in the bottom rows, i.e. the CVaR (**bottom left**) $p = 0.25$ (light purple) and $p = 0.5$ (light purple) from Equation (21), the extremile (**bottom center**) with $b = 2.5$ (light yellow) and $b = 1.5$ (dark yellow) from Equation (22), and the ESRM (**bottom right**) with $\gamma = 3$ (light pink) and $\gamma = 1$ (dark pink) from Equation (23). The parameters $(p, b, \gamma)$ are risk-sensitivity parameters that increase or decrease the size of the uncertainty sets.

We interpret this phenomenon as the "hidden smoothness" of $F_\sigma$, in that the non-differentiable points of the map $w \mapsto \max_{q \in \mathcal{P}(\sigma)} q^\top \ell(w)$ are precisely the points at which $\ell_i(w) = \ell_j(w)$ for some $i \neq j$, as the subdifferential may contain multiple elements (Mehta et al., 2023, Proposition 2). Thus, if the losses are well-separated enough (in comparison to the spectrum $\sigma$) at the minimizer $w_0^\star$, the objective for the non-smooth setting $\nu = 0$ and regularized setting $\nu > 0$ result in the same minimizer.

**Spectral Risk Measures.** Recall that for a spectral risk measure with spectrum $\sigma = (\sigma_1, \ldots, \sigma_n)$, the uncertainty set $\mathcal{P}(\sigma)$ is given by:

$$\mathcal{P}(\sigma) = \text{ConvexHull}\left\{ \left(\sigma_{\pi(1)}, \ldots, \sigma_{\pi(n)}\right) \; : \; \pi \text{ is a permutation on } [n] \right\}. \tag{20}$$

As for particular examples: for $p \in [0, 1]$, the $p$-*CVaR* (a.k.a. superquantile) (Rockafellar & Royset, 2013; Kawaguchi & Lu, 2020; Laguel et al., 2021) requires that $k = np$ elements of $\sigma$ be non-zero with equal probability and that the remaining $n - k$ are zero. The $b$-*extremile* (Daouia et al., 2019) and $\gamma$-*exponential spectral risk measure* (Cotter & Dowd, 2006) define their spectra the equations below.

$$\sigma_i = \begin{cases} \frac{1}{np} & \text{if } i \in \{\lceil n(1-p) \rceil, \ldots, n\} \\ 1 - \frac{\lfloor np \rfloor}{np} & \text{if } \lfloor n(1-p) \rfloor < i < \lceil n(1-p) \rceil \end{cases} \qquad p\text{-CVaR}, p \in (0,1) \tag{21}$$

$$\sigma_i = \left(\frac{i}{n}\right)^b - \left(\frac{i-1}{n}\right)^b \qquad\qquad\qquad b\text{-extremile}, b \geq 1 \tag{22}$$

$$\sigma_i = \frac{e^\gamma (e^{\gamma i/n} - e^{\gamma(i-1)/n})}{1 - e^{-\gamma}} \qquad\qquad\qquad \gamma\text{-ESRM}, \gamma > 0 \tag{23}$$

The multiple cases in the CVaR definition account for the instance in which $np$ is not an integer. In light of Lem. 7, when $\nu > 0$, we have that the objective based on the uncertainty set (20) is differentiable with Lipschitz continuous gradient (i.e. smooth). To be more specific, we may call this the *regularized* or *smoothed* spectral risk measure. On the other hand, as described in Sec. 2, SRMs were historically computed for any loss vector $l \in \mathbb{R}^n$ as

$$\mathcal{R}_\sigma(l) = \max_{q \in \mathcal{P}(\sigma)} q^\top l = \sum_{i=1}^n \sigma_i l_{(i)}, \tag{24}$$

where $l_{(1)} \leq \ldots \leq l_{(n)}$ are the order elements of $l$. For this reason, SRMs may also be called $L$-risks (Maurer et al., 2021), based on classical $L$-estimators (linear combinations of order statistics) from the statistics literature (Shorack, 2017). In fact, they were originally introduced as functionals $\mathbb{L}_s$ on arbitrary real-valued probability measures (Acerbi & Tasche, 2002) given by a weighted integral of the quantile function

$$\mathbb{L}_s[F] = \int_0^1 F^{-1}(p) s(p) \, \mathrm{d}p,$$

where $F^{-1}(p) = \inf\{x \in \mathbb{R} : F(x) > p\}$ is the *quantile* function of cumulative distribution function (CDF) $F$ and $s$ is a non-negative, non-decreasing function satisfying $\int_0^1 s(p) \, \mathrm{d}p = 1$. To recover (24) we view $l_1, \ldots, l_n$ as a random sample and define $F_n$ as the empirical distribution induced by the sample. Then, the quantile function $F_n^{-1}(p)$ is given by the order statistic $l_{(i)}$ when $p \in \left(\frac{i-1}{n}, \frac{i}{n}\right)$ and the discontinuity points of $F_n$ are defined to make it left-continuous. Applying $\mathbb{L}_s$ to the empirical CDF yields

$$\mathbb{L}_s[F_n] = \int_0^1 F_n^{-1}(p) s(p) \, \mathrm{d}p = \sum_{i=1}^n \int_{(i-1)/n}^{i/n} F_n^{-1}(p) s(p) \, \mathrm{d}p$$

$$= \sum_{i=1}^n l_{(i)} \cdot \int_{(i-1)/n}^{i/n} s(p) \, \mathrm{d}p = \sum_{i=1}^n \sigma_i l_{(i)}$$

for $\sigma_i = \int_{(i-1)/n}^{i/n} s(p) \, \mathrm{d}p$. Thus, the formulae for $\sigma_i$ for each SRM are defined by taking an $n$-bin discretization of a continuous spectrum $s$ over $[0, 1)$. Notably, the permutation-based description of spectral risk measures in (24) is a unique feature of the their discrete versions. For a visualization of the feasible set $\mathcal{P}(\sigma)$ for the CVaR, extremile, and ESRM, see Fig. 7.

## C  EFFICIENT IMPLEMENTATION OF PROSPECT

In this section, we describe Prospect including computational details, in a way that is amenable to implementation. For convenience, the conceptual description of the algorithm from Sec. 3 is restated in Algorithm 1.

**Efficient Implementation.** We exactly solve the maximization problem

$$q = q^{\text{opt}}(l) = \arg\max_{q \in \mathcal{P}(\sigma)} \left\{ q^\top l - (\nu/n) \sum_{i=1}^n f(nq_i) \right\}. \tag{25}$$

by a sequence of three steps:

- **Sorting:** Find $\pi$ such that $l_{\pi(1)} \leq \ldots \leq l_{\pi(n)}$.

- **Isotonic regression:** Apply Pool Adjacent Violators (PAV) (Subroutine 1) to solve the isotonic regression minimization problem (13), yielding solution $c = c^{\text{opt}}(l)$.

- **Conversion:** Use (14) to convert $c$ back to $q = q^{\text{opt}}(l)$.

The sorting step runs in $O(n \ln n)$ elementary operations whereas the isotonic regression and conversion steps run in $O(n)$ operations. Crucially, retrieving $q$ from the output $c = c^{\text{opt}}(l)$ in the third

---

**Algorithm 1** Prospect (restated from Sec. 3)

---

**Inputs:** Initial $w_0$, spectrum $\sigma$, number of iterations $T$, regularization $\mu > 0$, shift cost $\nu > 0$.
**Hyperparameter:** Stepsize $\eta > 0$.
1: **Initialize** $l \leftarrow \ell(w_0)$ and $g_i \leftarrow \nabla \ell_i(w_0) + \mu w_0$ for $i = 1, \ldots, n$.
2: Set $q \leftarrow \arg\max_{\bar{q} \in \mathcal{P}(\sigma)} \bar{q}^\top l - \nu D(q \| \mathbf{1}_n/n)$ and $\rho \leftarrow q$.
3: Set $\bar{g} \leftarrow \sum_{i=1}^n \rho_i g_i \in \mathbb{R}^d$.
4: Set $w \leftarrow w_0$.
5: **for** $T$ iterations **do**
6:     Sample $i, j \sim \text{Unif}[n]$ independently.
7:     $v \leftarrow nq_i(\nabla \ell_i(w) + \mu w) - n\rho_i g_{i_t} + \bar{g}$.            ▷ **Iterate Update**
8:     $w \leftarrow w - \eta v$.
9:     $l_j \leftarrow \ell_j(w)$.            ▷ **Bias Reducing Update**
10:     $q \leftarrow \arg\max_{\bar{q} \in \mathcal{P}(\sigma)} \bar{q}^\top l - \nu D(\bar{q} \| \mathbf{1}_n/n)$.
11:     $\bar{g} \leftarrow \bar{g} - \rho_i g_i + q_i (\nabla \ell_i(w) + \mu w)$.            ▷ **Variance Reducing Update**
12:     $g_i \leftarrow \nabla \ell_i(w) + \mu w$.
13:     $\rho_i \leftarrow q_i$.
**Output:** Final point $w$.

---

step can be done by a single $O(n)$-time pass by setting

$$q_{\pi(i)} = \frac{1}{n}[f^*]'\left(\frac{1}{\nu}(l_{\pi(i)} - c_i)\right)$$

for $i = 1, \ldots, n$, as opposed to computing the inverse $\pi^{-1}$ and use (14) directly, which in fact requires another sorting operation and can be avoided. Because only one element of $l$ changes on every iteration, we may sort it by simply bubbling the value of the index that changed into its correct position to generate the newly sorted version. The full algorithm is given Algorithm 2. We give a brief explanation on the PAV algorithm for general $f$-divergences below.

**Pool Adjacent Violators (PAV) Algorithm.** First, recall the optimization problem we wish to solve:

$$\min_{\substack{c \in \mathbb{R}^n \\ c_1 \leq \ldots \leq c_n}} \sum_{i=1}^n g_i(c_i; l), \quad \text{where} \quad g_i(c_i; l) := \sigma_i c_i + \frac{\nu}{n} f^*\left(\frac{l_{\pi(i)} - c_i}{\nu}\right). \tag{26}$$

The objective can be thought of as fitting a real-valued monotonic function to the points $(1, l_{\pi(1)}), \ldots, (n, l_{\pi(n)})$, which would require specifying its values $(c_1, \ldots, c_n)$ on $(1, \ldots, n)$ and defining the function as any $x \in [c_j, c_{j+1}]$ on $(j, j+1)$. Because $l_{\pi(1)} \leq \ldots \leq l_{\pi(n)}$, if we evaluated our function $(c_1, \ldots, c_n)$ on a loss such as $\sum_{i=1}^n (l_{\pi(i)} - c_i)^2$, we may easily solve the problem by returning $c_1 = l_{\pi(1)}, \ldots, c_n = l_{\pi(n)}$. However, by specifying functions $g_1, \ldots, g_n$ we allow our loss function to change in different regions of the inputs space $\{1, \ldots, n\}$. In such cases, the monotonicity constraint $c_1 \leq \ldots \leq c_n$ is often violated because individually minimizing $g_i(c_i)$ for each $c_i$ has no guarantee of yielding a function that is monotonic.

The idea behind the PAV algorithm is to attempt a pass at minimizing each $g_i$ individually, and correcting *violations* as they appear. To provide intuition, define $c_i^* \in \arg\min_{c_i \in \mathbb{R}} g_i(c_i)$, and consider $i < j$ such that $c_i^* > c_j^*$. If $f^*$ is strictly convex, then $g_i(x) > g_i(c_i^*)$ for any $x < c_i^*$ and similarly $g_j(x) > g_j(c_j^*)$ for any $x > c_j^*$. Thus, to correct the violation, we decrease $c_i^*$ to $\bar{c}_i$ and increase $c_j^*$ to $\bar{c}_j$ until $\bar{c}_i = \bar{c}_j$. We determine this midpoint precisely by

$$\bar{c}_i = \bar{c}_j = \arg\min_{x \in \mathbb{R}} g_i(x) + g_j(x)$$

as these are exactly the contributions made by these terms in the overall objective. The computation above is called *pooling* the indices $i$ and $j$. We may generalize this viewpoint to *violating chains*, that is collections of contiguous indices $(i, i+1, \ldots, i+m)$ such that $c_j^* < c_i^*$ for all $j < i$ and $c_j^* > c_{i+m}^*$ for all $j > i+m$, but $c_i^* > c_{i+m}^*$. One approach is use dynamic programming to identify

such chains and then compute the pooled quantities

$$\bar{c}_i = \arg\min_{x \in \mathbb{R}} \sum_{k=1}^{m} g_{i+k}(x).$$

This requires two passes through the vector: one for identifying violators and the other for pooling. The Pool Adjacent Violators algorithm, on the other hand, is able to perform both operations in one pass by greedily pooling violators as they appear. This can be viewed as a meta-algorithm, as it hinges on the notion that the solution of "larger" pooling problems can be easily computed from solutions of "smaller" pooling problems. Precisely, for indices $S \subseteq [n] = \{1, \ldots, n\}$ define

$$\mathrm{Sol}(S) = \arg\min_{x \in \mathbb{R}} \sum_{i \in S} g_i(x).$$

We rely on the existence of an operation Pool, such that for any $S, T \subseteq [n]$ such that $S \cap T = \emptyset$, we have that

$$\mathrm{Sol}(S \cup T) = \mathrm{Pool}\left(\mathrm{Sol}(S), m(S), \mathrm{Sol}(T), m(T)\right), \tag{27}$$

where $m(S)$ denotes "metadata" associated to $S$, and that the number of elementary operations in the Pool function is $O(1)$ with respect to $|S| + |T|$. We review our running examples.

For the $\chi^2$-divergence, we have that $f_{\chi^2}(x) = x^2 - 1$ and $f_{\chi^2}^*(y) = y^2/4 + 1$, so

$$\mathrm{Sol}(S) = \arg\min_{x \in \mathbb{R}} \left\{ x \left( \sum_{i \in S} \sigma_i \right) + |S| + \frac{1}{4n\nu} \sum_{i \in S} (l_{\pi(i)} - x)^2 \right\}$$

$$= \frac{1}{|S|} \left[ \sum_{i \in S} l_{\pi(i)} - 2n\nu \sum_{i \in S} \sigma_i \right]$$

$$\mathrm{Sol}(S \cup T) = \frac{1}{|S| + |T|} \left[ \sum_{i \in S \cup T} l_{\pi(i)} - 2n\nu \sum_{i \in S \cup T} \sigma_i \right]$$

$$= \frac{|S| \, \mathrm{Sol}(S) + |T| \, \mathrm{Sol}(T)}{|S| + |T|}.$$

Thus, the metadata $m(S) = |S|$ used in the pooling step eq. (27) is the size of each subset.

For the KL divergence, $f_{\mathrm{KL}}(x) = x \ln x$ and $f_{\mathrm{KL}}^*(y) = e^{-1} \exp(y)$, so so

$$\mathrm{Sol}(S) = \arg\min_{x \in \mathbb{R}} \left\{ x \left( \sum_{i \in S} \sigma_i \right) + \frac{\nu}{ne} \sum_{i \in S} \exp\left(l_{\pi(i)}/\nu\right) \exp\left(-x/\nu\right) \right\}$$

$$= \nu \left[ \ln \sum_{i \in S} \exp\left(l_{\pi(i)}/\nu\right) - \ln \sum_{i \in S} \sigma_i - \ln n - 1 \right]$$

$$\mathrm{Sol}(S \cup T) = \nu \left[ \ln \sum_{i \in S \cup T} \exp\left(l_{\pi(i)}/\nu\right) - \ln \sum_{i \in S \cup T} \sigma_i - \ln n - 1 \right]$$

$$= \nu \left[ \ln \left( \sum_{i \in S} \exp\left(l_{\pi(i)}/\nu\right) + \sum_{i \in T} \exp\left(l_{\pi(i)}/\nu\right) \right) - \ln \left( \sum_{i \in S} \sigma_i + \sum_{i \in T} \sigma_i \right) - \ln n - 1 \right].$$

Here, we carry the metadata $m(S) = \left( \ln \sum_{i \in S} \exp\left(l_{\pi(i)}/\nu\right), \ln \sum_{i \in S} \sigma_i \right)$, which can easily be combined and plugged into the function

$$(m_1, m_2), (m_1', m_2') \mapsto \nu \left[ \ln\left(\exp m_1 + \exp m_1'\right) - \ln\left(\exp m_2 + \exp m_2'\right) - \ln n - 1 \right]. \tag{28}$$

for two instances of metadata $(m_1, m_2)$ and $(m_1', m_2')$. We carry the "logsumexp" instead of just the sum of exponential quantities for numerical stability, and Equation (28) applies this operation as well. It might be that $\sum_{i \in S} \sigma_i = 0$, e.g. for the superquantile. In this case, we may interpret

---

**Algorithm 2** Prospect (with exact implementation details)

**Inputs:** Initial points $w_0$, spectrum $\sigma$, stepsize $\eta > 0$, number of iterations $T$, regularization parameter $\mu > 0$, shift cost $\nu > 0$, loss/gradient oracles $\ell_1, \ldots, \ell_n$ and $\nabla \ell_1, \ldots, \nabla \ell_n$.

1: $l \leftarrow \ell(w_0)\mathbb{R}^n$.
2: $g \leftarrow (\nabla \ell_i(w_0) + \mu w_0)_{i=1}^n \in \mathbb{R}^{n \times d}$.
3: $\pi \leftarrow \text{argsort}(l)$.
4: $c \leftarrow \text{PAV}(l, \pi, \sigma)$.                                                    ▷ Subroutine 1 or Subroutine 2
5: $q \leftarrow \text{Convert}(c, l, \pi, \nu, \mathbf{0}_n)$.                                   ▷ Subroutine 3
6: $\rho \leftarrow q$.
7: $\bar{g} \leftarrow \sum_{i=1}^n \rho_i g_i \in \mathbb{R}^d$.
8: **for** $T$ iterations **do**
9:       Sample $i, j \sim \text{Unif}[n]$.
10:      $v \leftarrow nq_i(\nabla \ell_i(w) + \mu w) - n\rho_i g_i - \bar{g}$.            ▷ **Iterate Update**
11:      $w \leftarrow w - \eta v^{(t)}$.
12:      $l_j \leftarrow \ell_j(w)$.                                               ▷ **Bias Reducing Update**
13:      $\pi \leftarrow \text{Bubble}(\pi, l)$.                                    ▷ Subroutine 4
14:      $c \leftarrow \text{PAV}(l, \pi, \sigma)$.
15:      $q \leftarrow \text{Convert}(c, l, \pi, \nu, q)$.
16:      $\bar{g} \leftarrow \bar{g} - \rho_i g_i + q_i(\nabla \ell_i(w) + \mu w)$.    ▷ **Variance Reducing Update**
17:      $g_i \leftarrow \nabla \ell_i(w) + \mu w$.
18:      $\rho_i \leftarrow q_i$.

**Output:** Final point $w$.

---

$\text{Sol}(S) = -\infty$ and evaluate $\exp(-\infty) = 0$ in the conversion formula (26). Two examples of the PAV algorithm are given in Subroutine 1 and Subroutine 2, respectively. These operate by selecting the unique values of the optimizer and partitions of indices that achieve that value.

**Hardware Acceleration.** Finally, note that all of the subroutines in Algorithm 2 (Subroutine 1/Subroutine 2, Subroutine 3, and Subroutine 4) all require primitive operations such as control flow and linear scans through vectors. Because these steps are outside of the purview of oracle calls or matrix multiplications, they benefit from just-in-time compilation on the CPU. We accelerate these subroutines using the Numba package in Python and are able to achieve an approximate 50%-60% decrease in runtime across benchmarks.

---

**Subroutine 1** Pool Adjacent Violators (PAV) Algorithm for $\chi^2$ divergence

**Inputs:** Losses $(\ell_i)_{i \in [n]}$, argsort $\pi$, and spectrum $(\sigma_i)_{i \in [n]}$.

1: Initialize partition endpoints $(b_0, b_1) = (0, 1)$, partition value $v_1 = l_{\pi(1)} - 2n\nu\sigma_1$, number of parts $k = 1$.
2: **for** $i = 2, \ldots, n$ **do**
3:      Add part $k = k + 1$.
4:      Compute $v_k = l_{\pi(i)} - 2n\nu\sigma_i$.
5:      **while** $k \geq 2$ and $v_{k-1} \geq v_k$ **do**
6:          $v_{k-1} = \frac{(b_k - b_{k-1})v_{k-1} + (i - b_k)v_k}{i - b_{k-1}}$.
7:          Set $k = k - 1$.
8:      $b_k = i$.

**Output:** Vector $c$ containing $c_i = v_k$ for $b_{k-1} < i \leq b_k$.

---

---

**Subroutine 2** Pool Adjacent Violators (PAV) Algorithm for KL divergence

---

**Inputs:** Losses $(\ell_i)_{i \in [n]}$, argsort $\pi$, and spectrum $(\sigma_i)_{i \in [n]}$.
1: Initialize partition endpoints $(b_0, b_1) = (0, 1)$, number of parts $k = 1$.
2: Initialize partition value $v_1 = \nu \left( l_{\pi(1)}/\nu - \ln \sigma_1 - \ln n - 1 \right)$.
3: Initialize metadata $m_1 = \ell_{\pi(1)}/\nu$ and $t_1 = \ln \sigma_1$.
4: **for** $i = 2, \ldots, n$ **do**
5:     Add part $k = k + 1$.
6:     Compute $v_k = \nu \left( l_{\pi(i)}/\nu - \ln \sigma_i - \ln n - 1 \right)$.
7:     Compute $m_k = \ell_{\pi(i)}/\nu$ and $t_k = \ln \sigma_i$
8:     **while** $k \geq 2$ and $v_{k-1} \geq v_k$ **do**
9:         $m_{k-1} = \text{logsumexp}(m_{k-1}, m_k)$ and $t_{k-1} = \text{logsumexp}(t_{k-1}, t_k)$.
10:        $v_{k-1} = \nu \left( m_{k-1} - t_{k-1} - \ln n - 1 \right)$.
11:        Set $k = k - 1$.
12:     $b_k = i$.
**Output:** Vector $c$ containing $c_i = v_k$ for $b_{k-1} < i \leq b_k$.

---

**Subroutine 3** Convert

---

**Require:** Sorted vector $c \in \mathbb{R}$, vector $l \in \mathbb{R}^n$, argsort $\pi$ of $l$, shift cost $\nu \geq 0$, vector $q \in \mathbb{R}^n$.
1: **for** $i = 1, \ldots, n$ **do**
2:     Set $q_{\pi(i)} = (1/n)[f^*]' \left( (l_{\pi(i)} - c_i)/\nu \right)$.
3: **return** $q$.

---

**Subroutine 4** Bubble

---

**Require:** Index $j_{\text{init}}$, sorting permutation $\pi$, loss table $l$.
1: $j = j_{\text{init}}$.                $\triangleright$ If $l_{\pi(j_{\text{init}})}$ too small, bubble left.
2: **while** $j > 1$ and $l_{\pi(j)} < l_{\pi(j-1)}$ **do**
3:     Swap $\pi(j)$ and $\pi(j-1)$.
4: $j = j_{\text{init}}$.                $\triangleright$ If $l_{\pi(j_{\text{init}})}$ too large, bubble right.
5: **while** $j < n$ and $l_{\pi(j)} > l_{\pi(j+1)}$ **do**
6:     Swap $\pi(j)$ and $\pi(j+1)$.
7: **return** $\pi$

---

---

**Algorithm 3** Prospect (with iteration counters specified to accompany convergence analysis)

---

**Inputs:** Initial points $w^{(0)}$, stepsize $\eta > 0$, number of iterations $T$.

1: Set $z_i^{(0)} = \zeta_i^{(0)} = w^{(0)}$ for all $i \in [n]$.

2: $q^{(0)} = \arg\max_{q \in \mathcal{P}(\sigma)} q^\top \ell(w^{(0)}) - \bar{\nu}\Omega(q)$, $\rho^{(0)} = q^{(0)}$.

3: Set $l^{(0)} = (\ell_i(\zeta_i^{(0)}))_{i=1}^n \in \mathbb{R}^n$, $g^{(0)} = (\nabla r_i(z_i^{(0)}))_{i=1}^n \in \mathbb{R}^{d \times n}$, $\bar{g}^{(0)} = \sum_{i=1}^n \rho_i^{(0)} g_i^{(0)} \in \mathbb{R}^d$.

4: **for** $t = 0, \ldots, T-1$ **do**

5: $\quad i_t \sim \mathrm{Unif}([n])$, $j_t \sim \mathrm{Unif}([n])$.

6:

7: $\quad v^{(t)} = n q_{i_t}^{(t)} \nabla r_{i_t}(w^{(t)}) - (n\rho_{i_t}^{(t)} \nabla r_{i_t}(z_{i_t}^{(t)}) - \bar{g}^{(t)})$. $\qquad\qquad\qquad$ ▷ **Iterate Update**

8: $\quad w^{(t+1)} = w^{(t)} - \eta v^{(t)}$.

9:

10: $\quad \zeta_{j_t}^{(t+1)} = w^{(t)}$ and $\zeta_j^{(t+1)} = \zeta_j^{(t)}$ for $j \neq j_t$. $\qquad\qquad\qquad$ ▷ **Bias Reducing Update**

11: $\quad l^{(t+1)} = \ell(\zeta^{(t+1)})$.

12: $\quad q^{(t+1)} = \arg\max_{q \in \mathcal{P}(\sigma)} q^\top l^{(t+1)} - \bar{\nu}\Omega(q)$.

13:

14: $\quad z_{i_t}^{(t+1)} = w^{(t)}$ and $z_i^{(t+1)} = z_i^{(t)}$ for $i \neq i_t$. $\qquad\qquad\qquad$ ▷ **Variance Reducing Update**

15: $\quad g^{(t+1)} = (\nabla r_i(z^{(t+1)}))_{i=1}^n$.

16: $\quad \rho_{i_t}^{(t+1)} = q_{i_t}^{(t)}$ and $\rho_i^{(t+1)} = \rho_i^{(t)}$ for $i \neq i_t$.

17: $\quad \bar{g}^{(t+1)} = \sum_{i=1}^n \rho_i^{(t+1)} g_i^{(t+1)}$.

**Output:** Final point $w^{(T)}$

---

## D CONVERGENCE ANALYSIS OF PROSPECT

This section provides the main convergence analysis for Prospect. For readability, we reference the version of the algorithm presented in Alg. 3, which is written to match quantities appearing in the proof. We begin with a high-level overview, whereas the remaining subsections contain technical lemmas of interest along with key steps in the proof.

### D.1 OVERVIEW

Notation used throughout the proof is collected in Tab. 1, and is also introduced as it appears. In the following, we denote $M = L + \mu$ the smoothness constant of the regularized losses $r_i$. We denote $\mathbb{E}_t$ the expectation w.r.t to the randomness induced by picking $i_t, j_t$ given $w^{(t)}$, i.e. the conditional expectation given $w^{(t)}$. The optimum of (8) is denoted $w^\star$ and satisfies

$$\nabla(q^{\star\top} r(w^\star)) = 0, \text{ for } q^\star = \arg\max_{q \in \mathcal{P}(\sigma)} q^\top \ell(w^\star) - \bar{\nu}\Omega(q). \qquad (29)$$

Denote in addition $l^\star = \ell(w^\star)$. For simplicity, we use the shorthand

$$\Omega(q) := \frac{1}{n\alpha_n} D_f(q \| \mathbf{1}_n/n)$$

for any $f$-divergence $D_f$, where $\alpha_n$ is the strong convexity constant of the generator $f$ over the interval $[0, n]$. By Prop. 4, this gives that $\Omega$ a 1-strongly convex function over the probability simplex. All forthcoming statements will reference the setting of Algorithm 3. Note that when implementing the algorithm, storing the iterates $\{z_i^{(t)}\}_{i=1}^n$ and $\{\zeta_i^{(t)}\}_{i=1}^n$ is not necessary.

**Proof Outline.** We argue convergence by way of defining a Lyapunov function $V^{(t)}$, which will upper bound the quantity $\|w^{(t)} - w^\star\|_2^2$, which will be called the "main term". Specifically, define

$$V^{(t)} = \|w^{(t)} - w^\star\|_2^2 + c_1 S^{(t)} + c_2 T^{(t)} + c_3 U^{(t)} + c_4 R^{(t)}$$

where $c_1$, $c_2$, $c_3$, and $c_4$ are constants to be determined later, and

$$S^{(t)} = \frac{1}{n} \sum_{i=1}^{n} \|n\rho_i^{(t)} \nabla r_i(z_i^{(t)}) - nq_i^* \nabla r_i(w^\star)\|_2^2, \quad T^{(t)} = \sum_{i=1}^{n} \|\zeta_i^{(t)} - w^\star\|_2^2,$$

$$U^{(t)} = \frac{1}{n} \sum_{j=1}^{n} \|w^{(t)} - \zeta_j^{(t)}\|_2^2, \quad R^{(t)} = 2\eta n(q^{(t)} - q^\star)^\top (l^{(t)} - l^\star).$$

Though not included in the Lyapunov function, we will also introduce

$$Q^{(t)} = \frac{1}{n} \sum_{i=1}^{n} \|nq_i^{(t)} \nabla r_i(w^{(t)}) - nq_i^\star \nabla r_i(w^\star)\|_2^2.$$

When the shift cost $\nu$ is large, we will be able to simplify the analysis by excluding the terms $U^{(t)}$ and $R^{(t)}$. The colors are used for the convenience of the reader so that the quantities above are easy to track from result to result. Each term in the Lyapunov function is motivated by terms that appear when bounding the main term $\|w^{(t)} - w^\star\|_2^2$, which appears in **Step 2** below. The outline of the proof is as follows.

1. We introduce a lemma of general interest which is the key technical step in the analysis: bounding the bias of the gradient estimate $v^{(t)}$ given in line 7 of Algorithm 3.

2. We expand the main term and identify "descent" and "noise" terms, as in a standard analysis of stochastic gradient methods. The descent term will be treated using the technical lemma from the previous step, whereas the noise term will be upper bounded using standard techniques. In either case, additional non-negative terms are introduced that will be incorporated into the Lyapunov function.

3. We the bound the evolution of the Lyapunov terms that are not the main term. For the large shift cost setting, only $S^{(t)}$ and $T^{(t)}$ are needed, while $U^{(t)}$ and $R^{(t)}$ can be ignored.

4. In the final step, we tune any free constants that are encountered in previous steps. At this point, we split the proof into two subsections, one for the large shift cost and one for any shift cost.

### D.2  KEY TECHNICAL LEMMA: BIAS OF THE PROSPECT GRADIENT ESTIMATE

In this subsection, we present the key technical step that allows for the unconditional linear convergence of Prospect. When analyzing stochastic gradient methods in the smooth and strongly convex setting, we typically expand

$$\mathbb{E}_t \|w^{(t+1)} - w^\star\|_2^2 = \|w^{(t)} - w^\star\|_2^2 \underbrace{-2\eta \left\langle \mathbb{E}_t[v^{(t)}], w^{(t)} - w^\star \right\rangle}_{\text{descent term}} + \underbrace{\eta^2 \mathbb{E}_t \|v^{(t)}\|_2^2}_{\text{noise term}}. \tag{30}$$

First, note that the expectation of the primal gradient estimate $v^{(t)}$ is $\nabla r(w^{(t)})^\top q^{(t)}$, where $q^{(t)} = \arg\max_{q \in \mathcal{P}(\sigma)} q^\top l^{(t)} - \bar{\nu}\Omega(q)$, where $l^{(t)} \in \mathbb{R}^n$ denotes the estimate of the full loss vector. Applying standard convex inequalities to the descent term yields

$$-\left\langle \nabla r(w^{(t)})^\top q^{(t)}, w^{(t)} - w^\star \right\rangle \leq -\frac{\mu M}{\mu + M} \|w - w_\star\|_2^2$$

$$-\frac{1}{\mu + L} \sum_{i=1}^{n} q_i^{(t)} \left\| \nabla r_i(w^{(t)}) - \nabla r_i(w^\star) \right\|_2^2$$

$$-\left\langle \nabla r(w^\star)^\top q^{(t)}, w^{(t)} - w^\star \right\rangle.$$

The first two terms on the right-hand side are negative, which provide decrease in the expected distance-to-optimum value on every iterate. In the empirical risk minimization setting, the final term on the right-hand side would be zero due to the first-order optimality conditions on $w^\star$, as $q^{(t)} = \mathbf{1}_n/n$, implying the decrease of $\mathbb{E}_t \|w^{(t+1)} - w^\star\|_2^2$ for small enough $\eta$. However, because

$q^{(t)}$ is a potentially non-uniform vector estimated using the table of losses $l^{(t)}$ (as opposed to the loss vector $\ell(w^\star)$ at optimum), the term $-\left\langle \nabla r(w^\star)^\top q^{(t)}, w^{(t)} - w^\star \right\rangle$ is non-zero. Additionally, this term is multiplied only by the learning rate $\eta$, instead of the noise terms which are multiplied by $\eta^2$. Thus, this bias term must be bounded carefully in order to achieve the convergence guarantee under this regime. This is the subject of Lem. 9, the main result of this subsection.

**Lemma 9** (Bias Bound). *Consider any $w \in \mathbb{R}^d$, $l \in \mathbb{R}^n$, and $\bar{q} \in \mathcal{P}(\sigma)$. Define*

$$q := q^{opt}(l) = \arg\max_{p \in \mathcal{P}(\sigma)} p^\top l - \bar{\nu}\Omega(p).$$

*For any $\beta_1 \in [0, 1]$,*

$$- (\nabla r(w)^\top q - \nabla r(w^\star)^\top \bar{q})^\top (w - w^\star)$$
$$\leq -(q - \bar{q})^\top (\ell(w) - \ell(w^\star)) - \frac{\mu}{2} \|w - w^\star\|_2^2$$
$$- \frac{\beta_1}{4(M+\mu)\kappa_\sigma} \frac{1}{n} \sum_{i=1}^n \|nq_i \nabla r_i(w) - nq_i^\star \nabla r_i(w^\star)\|_2^2 + \frac{2\beta_1 G^2}{\bar{\nu}(M+\mu)\kappa_\sigma} n(q - q^\star)^\top (l - l^\star).$$

*Proof.* First, for any $q_i > 0$, we have that $w \mapsto q_i r_i(w)$ is $(q_i M)$-smooth and $(q_i \mu)$-strongly convex, so by applying standard convex inequalities (Lem. 30) we have that

$$q_i r_i(w^\star) \geq q_i r_i(w) + q_i \nabla r_i(w)^\top (w^\star - w)$$
$$+ \frac{1}{2q_i(M+\mu)} \|q_i \nabla r_i(w) - q_i \nabla r_i(w^\star)\|_2^2 + \frac{q_i \mu}{4} \|w - w^\star\|_2^2$$
$$\geq q_i r_i(w) + q_i \nabla r_i(w)^\top (w^\star - w)$$
$$+ \frac{1}{2\sigma_n(M+\mu)} \|q_i \nabla r_i(w) - q_i \nabla r_i(w^\star)\|_2^2 + \frac{q_i \mu}{4} \|w - w^\star\|_2^2$$

as $q_i \leq \sigma_n$. The second inequality holds for $q_i = 0$ as well, so by summing the inequality over $i$ and using that $\sum_i q_i = 1$, we have that

$$q^\top r(w^\star) \geq q^\top r(w) + q^\top \nabla r(w)(w^\star - w)$$
$$+ \frac{1}{2\sigma_n(M+\mu)} \sum_{i=1}^n \|q_i \nabla r_i(w) - q_i \nabla r_i(w^\star)\|_2^2 + \frac{\mu}{4} \|w - w^\star\|_2^2.$$

Applying the same argument replacing $q$ by $\bar{q}$ and swapping $w$ and $w^\star$ yields

$$\bar{q}^\top r(w) \geq \bar{q}^\top r(w^\star) + \bar{q}^\top \nabla r(w^\star)(w - w^\star)$$
$$+ \frac{1}{2\sigma_n(M+\mu)} \sum_{i=1}^n \|\bar{q}_i \nabla r_i(w) - \bar{q}_i \nabla r_i(w^\star)\|_2^2 + \frac{\mu}{4} \|w - w^\star\|_2^2.$$

Summing the two inequalities yields

$$- (q - \bar{q})^\top (r(w) - r(w^\star))$$
$$\geq - (\nabla r(w)^\top q - \nabla r(w^\star)^\top \bar{q})^\top (w - w^\star) + \frac{\mu}{2} \|w - w^\star\|_2^2$$
$$+ \frac{1}{2\sigma_n(M+\mu)} \left[ \sum_{i=1}^n \|q_i \nabla r_i(w) - q_i \nabla r_i(w^\star)\|_2^2 + \sum_{i=1}^n \|\bar{q}_i \nabla r_i(w) - \bar{q}_i \nabla r_i(w^\star)\|_2^2 \right].$$

Dropping the $\sum_{i=1}^n \|\bar{q}_i \nabla r_i(w) - \bar{q}_i \nabla r_i(w^\star)\|_2^2$ term and applying a weight of $\beta_1 \in [0,1]$ to $\sum_{i=1}^n \|q_i \nabla r_i(w) - q_i \nabla r_i(w^\star)\|_2^2$ still satisfies the inequality, which can equivalently be written as

$$- \left( \nabla r(w)^\top q - \nabla r(w^\star)^\top \bar{q} \right)^\top (w - w^\star) \leq -(q - \bar{q})^\top (r(w) - r(w^\star)) - \frac{\mu}{2} \|w - w^\star\|_2^2$$

$$- \frac{\beta_1}{2\sigma_n(M+\mu)} \sum_{i=1}^n \|q_i \nabla r_i(w) - q_i \nabla r_i(w^\star)\|_2^2. \quad (31)$$

Next, because

$$\|q_i \nabla r_i(w) - q_i^\star \nabla r_i(w^\star)\|_2^2 \leq 2 \|q_i \nabla r_i(w) - q_i \nabla r_i(w^\star)\|_2^2 + 2(q_i - q_i^\star)^2 \|\nabla r_i(w^\star)\|_2^2,$$

we have that (by summing over $i$) that

$$-\sum_{i=1}^n \|q_i \nabla r_i(w) - q_i \nabla r_i(w^\star)\|_2^2 \leq -\frac{1}{2} \sum_{i=1}^n \|q_i \nabla r_i(w) - q_i^\star \nabla r_i(w^\star)\|_2^2 + 4G^2 \|q - q^\star\|_2^2, \quad (32)$$

where we used that each $\|\nabla r_i(w^\star)\|_2 \leq 2G$. To see this, use that $\nabla r(w^\star)^\top q^\star = 0$ and $\nabla r(w^\star) = \nabla \ell(w^\star) + \mu w^\star$, so

$$\|\nabla r_i(w^\star)\|_2 = \|\nabla \ell_i(w^\star) + \mu w^\star\|_2 = \left\| \nabla \ell_i(w^\star) - \sum_{j=1}^n q_i^\star \nabla \ell_j(w^\star) \right\|_2 \leq 2G.$$

Because the map $q^{\text{opt}}$ is the gradient of a convex and $(1/\bar{\nu})$-smooth map, we also have that

$$\|q - q^\star\|_2^2 = \left\| q^{\text{opt}}(l) - q^{\text{opt}}(\ell(w^\star)) \right\|_2^2 \leq \frac{1}{\bar{\nu}} (q - q^\star)^\top (l - \ell(w^\star)), \quad (33)$$

so we apply the above to (32) to achieve

$$-\sum_{i=1}^n \|q_i \nabla r_i(w) - q_i \nabla r_i(w^\star)\|_2^2$$

$$\leq -\frac{1}{2} \sum_{i=1}^n \|q_i \nabla r_i(w) - q_i^\star \nabla r_i(w^\star)\|_2^2 + \frac{4G^2}{\bar{\nu}} (q - q^\star)^\top (l - \ell(w^\star)), \quad (34)$$

We also use (33) to claim non-negativity of $(q - q^\star)^\top (l - \ell(w^\star))$. Finally, because $\sum_i q_i = \sum_i q_i^\star = 1$, we have that

$$(q - \bar{q})^\top (r(w) - r(w^\star)) = (q - \bar{q})^\top \left( \ell(w) + \frac{\mu}{2} \|w\|_2^2 \mathbf{1}_n - \ell(w^\star) - \frac{\mu}{2} \|w^\star\|_2^2 \mathbf{1}_n \right)$$

$$= (q - \bar{q})^\top (\ell(w) - \ell(w^\star)) + (q - \bar{q})^\top \mathbf{1}_n \left( \|w\|_2^2 - \|w^\star\|_2^2 \right)$$

$$= (q - \bar{q})^\top (\ell(w) - \ell(w^\star)). \quad (35)$$

Combine (31), (34), and (35) along with $\kappa_\sigma = n\sigma_n$ to achieve the claim. $\qquad \square$

In the next step, we apply Lem. 9, as well as a noise bound to give the initial per-iterate progress bound on the distance-to-optimum quantity $\mathbb{E}_t \|w^{(t+1)} - w^\star\|_2^2$.

### D.3 BOUNDING THE DISTANCE-TO-OPTIMUM

To outline the remainder of the proof, observe the expansion (30). With the bias bound for the descent term in hand, we now upper bound the noise term.

**Lemma 10** (Noise Bound). *Consider the notations of Alg. 3, we have for any $\beta > 0$,*

$$\mathbb{E}_t \|v^{(t)}\|_2^2 \leq (1+\beta) \mathbb{E}_t \|n q_{i_t}^{(t)} \nabla r_{i_t}(w^{(t)}) - n q_{i_t}^\star \nabla r_{i_t}(w^\star)\|_2^2$$

$$+ (1 + \beta^{-1}) \mathbb{E}_t \|n \rho_{i_t}^{(t)} \nabla r_{i_t}(z_{i_t}^{(t)}) - n q_{i_t}^\star \nabla r_{i_t}(w^\star)\|_2^2.$$

*Proof.* In the following, we use the identity $\mathbb{E}\|X - \mathbb{E}[X]\|_2^2 = \mathbb{E}\|X\|_2^2 - \|\mathbb{E}[X]\|_2^2$ in equations denoted with $(\star)$. We denote by $\beta$ an arbitrary positive number stemming from using Young's inequality $\|a+b\|_2^2 \leq (1+\beta)\|a\|_2^2 + (1+\beta^{-1})\|b\|_2^2$ in equation $(\circ)$. Noting that $\sum_{i=1}^n q_i^\star \nabla r_i(w^\star) = 0$, we get,

$$
\mathbb{E}_t\left[\|v^{(t)} - \nabla(q^{*\top} r)(w^\star)\|_2^2\right]
$$

$$
= \mathbb{E}_t\Big[\|nq_{i_t}^{(t)}\nabla r_{i_t}(w^{(t)}) - nq_{i_t}^\star \nabla r_{i_t}(w^\star)
$$
$$
\qquad + nq_{i_t}^\star \nabla r_{i_t}(w^\star) - n\rho_{i_t}^{(t)}\nabla r_{i_t}(z_{i_t}^{(t)}) - (\nabla(q^{\star\top} r)(w^\star) - \bar{g}^{(t)})\|_2^2\Big]
$$

$$
\overset{(\star)}{=} \|\nabla(q^{(t)\top} r)(w^{(t)}) - \nabla(q^{\star\top} r)(w^\star)\|_2^2
$$
$$
\quad + \mathbb{E}_t\Big[\|nq_{i_t}^{(t)}\nabla r_{i_t}(w^{(t)}) - nq_{i_t}^\star \nabla r_{i_t}(w^\star) - (\nabla(q^{(t)\top} r)(w^{(t)}) - \nabla(q^{\star\top} r)(w^\star))
$$
$$
\qquad + nq_{i_t}^\star \nabla r_{i_t}(w^\star) - n\rho_{i_t}^{(t)}\nabla r_{i_t}(z_{i_t}^{(t)}) - (\nabla(q^{\star\top} r)(w^\star) - \bar{g}^{(t)})\|_2^2\Big]
$$

$$
\overset{(\circ)}{\leq} \|\nabla(q^{(t)\top} r)(w^{(t)}) - \nabla(q^{\star\top} r)(w^\star)\|_2^2
$$
$$
\quad + (1+\beta)\mathbb{E}_t\left[\|nq_{i_t}^{(t)}\nabla r_{i_t}(w^{(t)}) - nq_{i_t}^\star \nabla r_{i_t}(w^\star) - (\nabla(q^{(t)\top} r)(w^{(t)}) - \nabla(q^{\star\top} r)(w^\star))\|_2^2\right]
$$
$$
\quad + (1+\beta^{-1})\mathbb{E}_t\left[\|nq_{i_t}^\star \nabla r_{i_t}(w^\star) - n\rho_{i_t}^{(t)}\nabla r_{i_t}(z_{i_t}^{(t)}) - (\nabla(q^{\star\top} r)(w^\star) - \bar{g}^{(t)})\|_2^2\right]
$$

$$
\overset{(\star)}{=} -\beta\|\nabla(q^{(t)\top} r)(w^{(t)}) - \nabla(q^{\star\top} r)(w^\star)\|_2^2
$$
$$
\quad + (1+\beta)\mathbb{E}_t\left[\|nq_{i_t}^{(t)}\nabla r_{i_t}(w^{(t)}) - nq_{i_t}^\star \nabla r_{i_t}(w^\star)\|_2^2\right]
$$
$$
\quad + (1+\beta^{-1})\mathbb{E}_t\left[\|nq_{i_t}^\star \nabla r_{i_t}(w^\star) - n\rho_{i_t}^{(t)}\nabla r_{i_t}(z_{i_t}^{(t)})\|_2^2\right]
$$
$$
\quad - (1+\beta^{-1})\|\nabla(q^{\star\top} r)(w^\star) - \bar{g}^{(t)}\|_2^2.
$$

$\square$

We then combine the analyses of the first and second-order terms to yield the main result of this subsection.

**Lemma 11** (Analysis of distance-to-optimum term). *For any constants $\beta_1 \in [0,1]$ and $\beta_2 > 0$, and any $\bar{q} \in \mathcal{P}(\sigma)$, we have that*

$$
\mathbb{E}_t\|w^{(t+1)} - w^\star\|_2^2 \leq (1 - \eta\mu)\|w^{(t)} - w^\star\|_2^2
$$
$$
\quad - 2\eta(w^{(t)} - w^\star)^\top \nabla r(w^\star)\bar{q}
$$
$$
\quad - \eta\left(\frac{\beta_1}{2(M+\mu)\kappa_\sigma} - \eta(1+\beta_2)\right)Q^{(t)} + \eta^2(1+\beta_2^{-1})S^{(t)}
$$
$$
\quad + \frac{2\beta_1 G^2}{\bar{\nu}(M+\mu)\kappa_\sigma}R^{(t)} - 2\eta(q^{(t)} - \bar{q})^\top(\ell(w) - \ell(w^\star)).
$$

*Proof.* Recall the expansion given in (30), which is:

$$
\mathbb{E}_t\|w^{(t+1)} - w^\star\|_2^2 = \|w^{(t)} - w^\star\|_2^2 - 2\eta\left\langle \mathbb{E}_t[v^{(t)}], w^{(t)} - w^\star \right\rangle + \eta^2\mathbb{E}_t\|v^{(t)}\|_2^2. \tag{36}
$$

Observe that

$$
\mathbb{E}_t[v^{(t)}] = \sum_{i=1}^n q_i^{(t)}\nabla r(w^{(t)}) = \nabla r(w^{(t)})^\top q^{(t)}
$$

By Lem. 9 with $l = l^{(t)}$, $q = q^{(t)}$, $w = w^{(t)}$, and multiplying by $2\eta$, we have that

$$-2\eta(w^{(t)} - w^\star)^\top \nabla r(w^{(t)})^\top q^{(t)} \leq -2\eta(w^{(t)} - w^\star)^\top \nabla r(w^\star)\bar{q} - 2\eta(q^{(t)} - \bar{q})^\top (\ell(w^{(t)}) - \ell(w^\star))$$

$$- \mu\eta \left\| w^{(t)} - w^\star \right\|_2^2 - \frac{\eta\beta_1}{2(M+\mu)\kappa_\sigma} Q^{(t)}$$

$$+ \frac{2\beta_1 G^2}{\bar{\nu}(M+\mu)\kappa_\sigma} R^{(t)}.$$

The noise term is bounded by applying Lem. 10, so that for some $\beta_2 > 0$,

$$\eta^2 \mathbb{E}_t \|v^{(t)}\|_2^2 \leq \eta^2(1 + \beta_2)Q^{(t)} + \eta^2(1 + \beta_2^{-1})S^{(t)}.$$

Combine the two displays above to get the desired result. □

Now, we see the appearance of $S^{(t)}$, $T^{(t)}$, $U^{(t)}$, and $R^{(t)}$ in Lem. 11. We incorporate them into the Lyapunov function and describe their evolution in the next section.

### D.4 BOUNDING THE EVOLUTION OF THE LYAPUNOV FUNCTION TERMS

We now describe the evolution of the terms $S^{(t)}$, $T^{(t)}$, $U^{(t)}$, $R^{(t)}$ from iterate $t$ to iterate $t + 1$.

The first two terms simply measure the closeness of the iterates $\{z_{i_t}^{(t)}\}_{i=1}^n$ and $\{\zeta_{i_t}^{(t)}\}_{i=1}^n$ within the table to the optimum $w^\star$, measured either in closeness in weighted gradients $(S^{(t)} = \frac{1}{n}\sum_{i=1}^n \|n\rho_{i_t}^{(t)}\nabla r_{i_t}(z_{i_t}^{(t)}) - nq_{i_t}^\star \nabla r_{i_t}(w^\star)\|_2^2)$ or directly $(T^{(t)} = \sum_{i=1}^n \|\zeta_i^{(t)} - w^\star\|_2^2)$. Recall that $Q^{(t)} = \frac{1}{n}\sum_{i=1}^n \|nq_i^{(t)}\nabla r_i(w^{(t)}) - nq_i^\star \nabla r_i(w^\star)\|_2^2$.

**Lemma 12.** *The following hold.*

$$\mathbb{E}_t \left[ S^{(t+1)} \right] = \frac{1}{n}Q^{(t)} + \left(1 - \frac{1}{n}\right) S^{(t)},$$

$$\mathbb{E}_t \left[ T^{(t+1)} \right] = \|w^{(t)} - w^\star\|_2^2 + \left(1 - \frac{1}{n}\right) T^{(t)}.$$

*Proof.* Write

$$\mathbb{E}_t \left[ S^{(t+1)} \right]$$

$$= \frac{1}{n}\sum_{i=1}^n \mathbb{E}_t \left[ \|n\rho_i^{(t+1)}\nabla r_i(z_i^{(t+1)}) - nq_i^*\nabla r_i(w^\star)\|_2^2 \right]$$

$$= \frac{1}{n}\sum_{i=1}^n \left[ \frac{1}{n}\|nq_i^{(t)}\nabla r_i(w^{(t)}) - q_i^\star \nabla r_i(w^\star)\|_2^2 + \left(1 - \frac{1}{n}\right) \|n\rho_i^{(t)}\nabla r_{i_t}(z_i^{(t)}) - nq_i^*\nabla r_i(w^\star)\|_2^2 \right]$$

$$= \frac{1}{n}Q^{(t)} + \left(1 - \frac{1}{n}\right) S^{(t)}.$$

Similarly,

$$\mathbb{E}_t \left[ T^{(t+1)} \right] = \sum_{i=1}^n \mathbb{E}_t \left[ \|\zeta_i^{(t+1)} - w^\star\|_2^2 \right]$$

$$= \sum_{i=1}^n \left[ \frac{1}{n}\|w^{(t)} - w^\star\|_2^2 + \left(1 - \frac{1}{n}\right) \|\zeta_i^{(t)} - w^\star\|_2^2 \right]$$

$$= \|w^{(t)} - w^\star\|_2^2 + \left(1 - \frac{1}{n}\right) T^{(t)}.$$

□

Next, we handle $U^{(t)} = \frac{1}{n}\sum_{j=1}^{n}\|w^{(t)} - \zeta_j^{(t)}\|_2^2$, which can be ignored if we assume a particular lower bound on $\bar{\nu}$.

**Lemma 13.** *For any value of $\beta_2 > 0$, we have that*

$$\mathbb{E}_t\left[U^{(t+1)}\right] \le \eta^2(1 + \beta_2)Q^{(t)} + \eta^2(1 + \beta_2^{-1})S^{(t)}$$
$$+ \frac{\eta M^2}{\mu n}\left(1 - \frac{1}{n}\right)T^{(t)} + \left(1 - \frac{1}{n}\right)\frac{G^2}{2\bar{\nu}\mu n}R^{(t)} + \left(1 - \frac{1}{n}\right)U^{(t)}.$$

*Proof.* First, note that a separate index $j_t$ (independent of $i_t$) is used to update $\{\zeta_j^{(t)}\}_{j=1}^n$, so we may first take the expected value with respect to $j_t$ conditioned on $i_t$:

$$\mathbb{E}_t\left[U^{(t+1)}\right] = \mathbb{E}_t\left[\frac{1}{n}\sum_{j=1}^{n}\|w^{(t+1)} - \zeta_j^{(t+1)}\|_2^2\right]$$

$$= \frac{1}{n}\mathbb{E}_t\left[\frac{1}{n}\sum_{j=1}^{n}\left\|w^{(t+1)} - \zeta_j^{(t+1)}\right\|_2^2 \mid j_t = j\right] + \left(1 - \frac{1}{n}\right)\mathbb{E}_t\left[\frac{1}{n}\sum_{j=1}^{n}\|w^{(t+1)} - \zeta_j^{(t+1)}\|_2^2 \mid j_t \ne j\right]$$

$$= \frac{1}{n}\mathbb{E}_t\left[\left\|w^{(t+1)} - w^{(t)}\right\|_2^2\right] + \left(1 - \frac{1}{n}\right)\mathbb{E}_t\left[\frac{1}{n}\sum_{j=1}^{n}\|w^{(t+1)} - \zeta_j^{(t)}\|_2^2\right]$$

$$= \frac{\eta^2}{n}\mathbb{E}_t\left[\left\|v^{(t)}\right\|_2^2\right] + \left(1 - \frac{1}{n}\right)\mathbb{E}_t\left[\frac{1}{n}\sum_{j=1}^{n}\|w^{(t+1)} - \zeta_j^{(t)}\|_2^2\right].$$

Next, we expand the second term.

$$\frac{1}{n}\mathbb{E}_t\left[\sum_{j=1}^{n}\|w^{(t+1)} - \zeta_j^{(t)}\|_2^2\right]$$

$$= \frac{1}{n}\mathbb{E}_t\left[\sum_{j=1}^{n}\|w^{(t+1)} - w^{(t)}\|_2^2\right] + \frac{2}{n}\mathbb{E}_t\left[\sum_{j=1}^{n}(w^{(t+1)} - w^{(t)})^\top(w^{(t)} - \zeta_j^{(t)})\right] + \frac{1}{n}\mathbb{E}_t\left[\sum_{j=1}^{n}\|\zeta_j^{(t)} - w^{(t)}\|_2^2\right]$$

$$= \eta^2\mathbb{E}_t\left[\|v^{(t)}\|_2^2\right] - \frac{2\eta}{n}\sum_{j=1}^{n}\nabla(q^{(t)\top}r)(w^{(t)})^\top(w^{(t)} - \zeta_j^{(t)}) + \frac{1}{n}\sum_{j=1}^{n}\|\zeta_j^{(t)} - w^{(t)}\|_2^2.$$

The first term is simply the noise term that appears in Lem. 10, whereas the last term is $U^{(t)}$. Next, we have

$$-2\nabla(q^{(t)\top}r)(w^{(t)})^\top(w^{(t)} - \zeta_j^{(t)}) = -2(\nabla(q^{(t)\top}r)(w^{(t)}) - \nabla(q^{(t)\top}r)(\zeta_j^{(t)}))^\top(w^{(t)} - \zeta_j^{(t)})$$
$$- 2(\nabla(q^{(t)\top}r)(\zeta_j^{(t)}) - \nabla(q^{(t)\top}r)(w^\star))^\top(w^{(t)} - \zeta_j^{(t)})$$
$$- 2(\nabla(q^{(t)\top}r)(w^\star) - \nabla(q^{\star\top}r)(w^\star))^\top(w^{(t)} - \zeta_j^{(t)}),$$

where the last term is introduced because $\nabla(q^{\star\top}r)(w^\star) = 0$. We bound each of the three terms. First,

$$-2(\nabla(q^{(t)\top}r)(w^{(t)}) - \nabla(q^{(t)\top}r)(\zeta_j^{(t)}))^\top(w^{(t)} - \zeta_j^{(t)}) \le -2\mu\left\|w^{(t)} - \zeta_j^{(t)}\right\|_2^2$$

because $q^{(t)\top}r$ is $\mu$-strongly convex (Nesterov, 2018, Theorem 2.1.9). Second,

$$-2(\nabla(q^{(t)\top}r)(\zeta_j^{(t)}) - \nabla(q^{(t)\top}r)(w^\star))^\top(w^{(t)} - \zeta_j^{(t)}) \le \beta_4\|\nabla(q^{(t)\top}r)(\zeta_j^{(t)}) - \nabla(q^{(t)\top}r)(w^\star)\|_2^2 + \beta_4^{-1}\|\zeta_j^{(t)} - w^{(t)}\|_2^2$$
$$\le \beta_4 M^2\|\zeta_j^{(t)} - w^\star\|_2^2 + \beta_4^{-1}\|\zeta_j^{(t)} - w^{(t)}\|_2^2$$

by Young's inequality with parameter $\beta_4$ and the $M$-Lipschitz continuity of $\nabla(q^{(t)\top} r)$. Third,

$$-2(\nabla(q^{(t)\top} r)(w^\star) - \nabla(q^{\star\top} r)(w^\star))^\top (w^{(t)} - \zeta_j^{(t)}) = -2(\nabla((q^{(t)} - q^\star)^\top \ell)(w^\star))^\top (w^{(t)} - \zeta_j^{(t)})$$
$$\leq \beta_5 \|\nabla((q^{(t)} - q^\star)^\top \ell)(w^\star)\|_2^2 + \beta_5^{-1} \|\zeta_j^{(t)} - w^{(t)}\|_2^2$$
$$\leq \beta_5 G^2 \|q^{(t)} - q\|_2^2 + \beta_5^{-1} \|\zeta_j^{(t)} - w^{(t)}\|_2^2,$$

by Young's inequality with parameter $\beta_5$ and the $G$-Lipschitz continuity of each $\ell_i$. Combining with the above, we have

$$-2 \sum_{j=1}^n \nabla(q^{(t)\top} r)(w^{(t)})^\top (w^{(t)} - \zeta_j^{(t)}) \leq \beta_4 M^2 T^{(t)} + (\beta_4^{-1} + \beta_5^{-1} - 2\mu)U^{(t)} + \beta_5 G^2 n \|q^{(t)} - q^\star\|_2^2$$
$$\leq \mu^{-1} M^2 T^{(t)} + \mu^{-1} G^2 n \|q^{(t)} - q^\star\|_2^2$$

when we set $\beta_4 = \beta_5 = \mu^{-1}$. Hence, we get

$$\mathbb{E}_t\left[U^{(t+1)}\right]$$
$$= \frac{\eta^2}{n} \mathbb{E}_t\left[\left\|v^{(t)}\right\|_2^2\right] + \left(1 - \frac{1}{n}\right) \mathbb{E}_t\left[\frac{1}{n}\sum_{j=1}^n \|w^{(t+1)} - \zeta_j^{(t)}\|_2^2\right]$$
$$\leq \eta^2 \mathbb{E}_t\left[\left\|v^{(t)}\right\|_2^2\right] - \frac{\eta}{n}\left(1 - \frac{1}{n}\right) 2 \sum_{j=1}^n \nabla(q^{(t)\top} r)(w^{(t)})^\top (w^{(t)} - \zeta_j^{(t)}) + \left(1 - \frac{1}{n}\right) U^{(t)}$$
$$\leq \eta^2 \mathbb{E}_t\left[\left\|v^{(t)}\right\|_2^2\right] + \frac{\eta}{n}\left(1 - \frac{1}{n}\right)\left[\mu^{-1} M^2 T^{(t)} + \mu^{-1} G^2 n \|q^{(t)} - q^\star\|_2^2\right] + \left(1 - \frac{1}{n}\right) U^{(t)}$$
$$= \eta^2 \mathbb{E}_t\left[\left\|v^{(t)}\right\|_2^2\right] + \left(1 - \frac{1}{n}\right) \frac{\eta M^2}{\mu n} T^{(t)} + \left(1 - \frac{1}{n}\right) \frac{G^2}{2n\mu} 2n\eta \|q^{(t)} - q^\star\|_2^2 + \left(1 - \frac{1}{n}\right) U^{(t)}$$
$$= \eta^2 \mathbb{E}_t\left[\left\|v^{(t)}\right\|_2^2\right] + \left(1 - \frac{1}{n}\right) \frac{\eta M^2}{\mu n} T^{(t)} + \left(1 - \frac{1}{n}\right) \frac{G^2}{2n\mu\bar{\nu}} R^{(t)} + \left(1 - \frac{1}{n}\right) U^{(t)}$$
$$\leq \eta^2(1 + \beta_2)Q^{(t)} + \eta^2(1 + \beta_2^{-1})S^{(t)}$$
$$+ \frac{\eta M^2}{\mu n}\left(1 - \frac{1}{n}\right) T^{(t)} + \left(1 - \frac{1}{n}\right) \frac{G^2}{2\bar{\nu}\mu n} R^{(t)} + \left(1 - \frac{1}{n}\right) U^{(t)},$$

where the two last steps follow from Lem. 10 and Lem. 34 to claim $\|q^{(t)} - q^\star\|_2^2 \leq \frac{1}{\bar{\nu}}(q^{(t)} - q^\star)(l^{(t)} - l^\star)$. $\qquad\square$

Finally, we consider $R^{(t)} = 2\eta n(q^{(t)} - q^\star)^\top (l^{(t)} - l^\star)$. This can be viewed as a measurement of orthogonality between the vectors $q^{(t)} - q^\star$ and $l^{(t)} - l^\star$, which in turn can be viewed as the directions to the optimal gradient and optimal solution of a constrained optimization problem. Indeed, we may define

$$l^\star = \arg\min_{l \in \mathcal{L}}\left[h(l) := \max_{q \in \mathcal{P}(\sigma)} q^\top l - \bar{\nu}\Omega(q)\right],$$

and

$$\mathcal{L} = \left\{l \in \mathbb{R}^n : l \geq \ell(w) \text{ for some } w \in \mathbb{R}^d\right\},$$

where the inequality is taken element-wise. The set $\mathcal{L}$ is a convexification of the set $\ell(\mathbb{R}^d)$ which shares a minimizer and has the same minimum value. Indeed, letting $\bar{l}$ be any minimizer of $h$, select $\bar{w}$ such that $\bar{l} = \ell(\bar{w})$. Define $\bar{q} = \nabla h(\bar{l}) = \arg\max_{q \in \mathcal{P}(\sigma)} q^\top \bar{l} - \bar{\nu}\Omega(\bar{q})$, and due to the non-negativity of $\bar{q}$, we have that

$$\min_{l \in \mathcal{L}} h(l) = h(\bar{l}) = \bar{q}^\top \bar{l} - \bar{\nu}\Omega(\bar{q}) \geq \bar{q}^\top \ell(\bar{w}) - \bar{\nu}\Omega(\bar{q}).$$

Taking the maximum over $\bar{q}$ shows that $\min_{l \in \mathcal{L}} h(l) = h(\ell(\bar{w}))$. For convexity, for any $l_1, l_2 \in \mathcal{L}$ satisfying $l_1 \geq \ell(w_1)$ and $l_2 \geq \ell(w_2)$, and any $\theta \in (0, 1)$, apply the following inequalities element-wise:

$$\theta l_1 + (1 - \theta) l_2 \geq \theta \ell(w_1) + (1 - \theta) \ell(w_2) \geq \ell(\theta w_1 + (1 - \theta) w_2),$$

with $\theta w_1 + (1 - \theta) w_2 \in \mathbb{R}^d$. By convexity, $(q^{(t)} - q^\star)^\top (l^{(t)} - l^\star) \geq 0$. Finally, this term is of particular importance because the term $-(q - q^\star)^\top (\ell(w) - \ell(w^\star))$ that appears in Lem. 9 can be used for cancellation in this case. The next result describes its evolution.

**Lemma 14.** *For any $\beta_3 > 0$, it holds that*

$$\mathbb{E}_t \left[ R^{(t+1)} \right] \leq 2\eta (q^{(t)} - q^\star)^\top (\ell(w^{(t)}) - l^\star) + \left( 1 - \frac{1}{n} \right) R^{(t)}$$
$$+ \frac{\eta G^2 n}{2\bar{\nu}} \beta_3^{-1} T^{(t)} + \frac{2\eta G^2 n}{\bar{\nu}} (1 + \beta_3) U^{(t)}.$$

*Proof.* First, decompose

$$(q^{(t+1)} - q^\star)^\top (l^{(t+1)} - l^\star) = (q^{(t)} - q^\star)^\top (l^{(t+1)} - l^\star) + (q^{(t+1)} - q^{(t)})^\top (l^{(t+1)} - l^{(t)}) \quad (37)$$
$$+ (q^{(t+1)} - q^{(t)})^\top (l^{(t)} - l^\star). \quad (38)$$

Because $q^{(t)} = q^{\text{opt}}(l^{(t)})$ for all $t$, and $q^{\text{opt}}(\cdot)$ is the gradient of a convex and $(1/\bar{\nu})$-smooth function, we have for the second term of (38) that

$$(q^{(t+1)} - q^{(t)})^\top (l^{(t+1)} - l^{(t)}) \leq \frac{1}{\bar{\nu}} \|l^{(t+1)} - l^{(t)}\|_2^2.$$

Next, using Young's inequality, that is, $a^\top b \leq \frac{\beta_3}{2} \|a\|_2^2 + \frac{\beta_3^{-1}}{2} \|b\|_2^2$ for any $\beta_3 > 0$, we have for the third term term of (38) that

$$(q^{(t+1)} - q^{(t)})^\top (l^{(t)} - l^\star) \leq \frac{\beta_3}{2} \|q^{(t+1)} - q^{(t)}\|_2^2 + \frac{\beta_3^{-1}}{2} \|l^{(t)} - l^\star\|_2^2$$
$$\leq \frac{\beta_3}{2\bar{\nu}^2} \|l^{(t+1)} - l^{(t)}\|_2^2 + \frac{\beta_3^{-1}}{2} \|l^{(t)} - l^\star\|_2^2.$$

Note that we have

$$\mathbb{E}_t \left[ l^{(t+1)} \right] = \frac{1}{n} \ell(w^{(t)}) + \left( 1 - \frac{1}{n} \right) l^{(t)}.$$

Hence, we get,

$$\frac{1}{2\eta n} \mathbb{E}_t \left[ R^{(t+1)} \right] = \frac{1}{n} (q^{(t)} - q^\star)^\top (l(w^{(t)}) - l^\star) + \left( 1 - \frac{1}{n} \right) (q^{(t)} - q^\star)^\top (l^{(t)} - l^\star)$$
$$+ \mathbb{E}_t \left[ (q^{(t+1)} - q^{(t)})^\top (l^{(t+1)} - l^{(t)}) \right] + \mathbb{E}_t \left[ (q^{(t+1)} - q^{(t)})^\top (l^{(t)} - l^\star) \right]$$
$$\leq \frac{1}{n} (q^{(t)} - q^\star)^\top (l(w^{(t)}) - l^\star) + \left( 1 - \frac{1}{n} \right) (q^{(t)} - q^\star)^\top (l^{(t)} - l^\star)$$
$$+ \left( \frac{1}{\bar{\nu}} + \frac{\beta_3}{2\bar{\nu}^2} \right) \mathbb{E}_t \left[ \left\| l^{(t+1)} - l^{(t)} \right\|_2^2 \right] + \frac{\beta_3^{-1}}{2} \|l^{(t)} - l^\star\|_2^2$$
$$= \frac{1}{n} (q^{(t)} - q^\star)(\ell(w^{(t)}) - l^\star) + \left( 1 - \frac{1}{n} \right) (q^{(t)} - q^\star)^\top (l^{(t)} - l^\star)$$
$$+ \frac{1}{n\bar{\nu}} \left( 1 + \frac{\beta_3}{2\bar{\nu}} \right) \sum_{j=1}^{n} (\ell_j(w^{(t)}) - \ell_j(\zeta_j))^2$$
$$+ \frac{\beta_3^{-1}}{2} \sum_{j=1}^{n} (\ell_j(\zeta_j) - \ell_j(w^\star))^2.$$

Then, apply the $G$-Lipschitz continuity of each $\ell_i$ to achieve

$$\frac{1}{2\eta n}\mathbb{E}_t\left[R^{(t+1)}\right] \le \frac{1}{n}(q^{(t)} - q^\star)(\ell(w^{(t)}) - l^\star) + \left(1 - \frac{1}{n}\right)(q^{(t)} - q^\star)^\top(l^{(t)} - l^\star)$$

$$+ \frac{G^2}{n\bar{\nu}}\left(1 + \frac{\beta_3}{2\bar{\nu}}\right)\sum_{j=1}^{n}\|w^{(t)} - \zeta_j^{(t)}\|_2^2$$

$$+ \frac{G^2\beta_3^{-1}}{2}\sum_{j=1}^{n}\|\zeta_j^{(t)} - w^\star\|_2^2.$$

Replacing $\beta_3$ by $2\bar{\nu}\beta_3$ gives the claim. $\qquad\square$

## D.5 TUNING CONSTANTS AND FINAL RATE

Recall that our Lyapunov function is given by

$$V^{(t)} = \|w^{(t)} - w^\star\|_2^2 + c_1 S^{(t)} + c_2 T^{(t)} + c_3 U^{(t)} + c_4 R^{(t)}.$$

Recall in addition the definitions

$$S^{(t)} = \frac{1}{n}\sum_{i=1}^{n}\|n\rho_i^{(t)}\nabla r_i(z_i^{(t)}) - nq_i^*\nabla r_i(w^\star)\|_2^2, \quad T^{(t)} = \sum_{i=1}^{n}\|\zeta_i^{(t)} - w^\star\|_2^2,$$

$$U^{(t)} = \frac{1}{n}\sum_{j=1}^{n}\|w^{(t)} - \zeta_j^{(t)}\|_2^2, \quad R^{(t)} = 2\eta n(q^{(t)} - q^\star)^\top(l^{(t)} - l^\star).$$

We will derive a value $\tau > 0$ such that for all $t \ge 0$,

$$\mathbb{E}_t\left[V^{(t+1)}\right] \le (1 - \tau^{-1})V^{(t)}.$$

In order to describe our rates, we define the condition number $\kappa := M/\mu$ and recall that $\kappa_\sigma = n\sigma_n$.

### D.5.1 STEP 3A: ANALYZE LARGE SHIFT COST SETTING.

The following gives the convergence rate for large shift cost.

**Theorem 15.** *Suppose the shift cost satisfies*

$$\bar{\nu} \ge 8nG^2/\mu.$$

*Then, the sequence of iterates produced by Algorithm 3 with $\eta = 1/(12(\mu + M)\kappa_\sigma)$ achieves*

$$\mathbb{E}\|w^{(t)} - w^\star\|_2^2 \le (1 + \sigma_n^{-1} + \sigma_n^{-2})\exp(-t/\tau)\|w^{(0)} - w^\star\|_2^2.$$

*with*

$$\tau = 2\max\{n, 24\kappa_\sigma(\kappa + 1)\}.$$

*Proof.* First, invoke Lem. 11 with $q' = q^{(t)}$ and $\beta_1 = 1$ to obtain

$$\mathbb{E}_t\|w^{(t+1)} - w^\star\|_2^2 \le (1 - \eta\mu)\|w^{(t)} - w^\star\|_2^2 \tag{39}$$

$$- 2\eta(w^{(t)} - w^\star)^\top\nabla r(w^\star)q^{(t)} + \frac{2G^2}{\bar{\nu}(M + \mu)\kappa_\sigma}R^{(t)} \tag{40}$$

$$- \eta\left(\frac{1}{2(M + \mu)\kappa_\sigma} - \eta(1 + \beta_2)\right)Q^{(t)} + \eta^2(1 + \beta_2^{-1})S^{(t)}. \tag{41}$$

We will first bound (40), by using that $\nabla r(w^\star)q^\star = 0$ and Young's inequality with parameter $a > 0$ to write

$$\left| (w^{(t)} - w^\star)^\top \nabla r(w^\star)q^{(t)} \right| = \left| (w^{(t)} - w^\star)^\top \nabla r(w^\star)(q^{(t)} - q^\star) \right|$$

$$\leq \frac{a}{2} \left\| \nabla r(w^\star)^\top (q^{(t)} - q^\star) \right\|_2^2 + \frac{1}{2a} \left\| w^{(t)} - w^\star \right\|_2^2$$

$$\leq \frac{aG^2\gamma_*^2}{2\bar{\nu}^2} T^{(t)} + \frac{1}{2a} \left\| w^{(t)} - w^\star \right\|_2^2,$$

where we used in the second inequality that:

$$\left\| \nabla r(w^\star)^\top (q^{(t)} - q^\star) \right\|_2^2 = \left\| \nabla \ell(w^\star)^\top (q^{(t)} - q^\star) \right\|_2^2 \leq \gamma_*^2 \left\| q^{(t)} - q^\star \right\|_2^2 \leq \frac{\gamma_*^2}{\bar{\nu}^2} \left\| l^{(t)} - l^\star \right\|_2^2$$

$$\leq \frac{G^2\gamma_*^2}{\bar{\nu}^2} \sum_{i=1}^n \| \zeta_i^{(t)} - w^\star \|_2^2 = \frac{G^2\gamma_*^2}{\bar{\nu}^2} T^{(t)}.$$

We also have by Cauchy-Schwartz and Lipschitz continuity that

$$R^{(t)} = 2\eta n(q^{(t)} - q^\star)^\top (l^{(t)} - l^\star) \leq \frac{2\eta n}{\bar{\nu}} \left\| l^{(t)} - l^\star \right\|_2^2 \leq \frac{2\eta nG^2}{\bar{\nu}} T^{(t)}.$$

Combining the above displays yields

$$- 2\eta(w^{(t)} - w^\star)^\top \nabla r(w^\star)q^{(t)} + \frac{2G^2}{\bar{\nu}(M + \mu)\kappa_\sigma} R^{(t)}$$

$$\leq \frac{\eta G^2}{\bar{\nu}^2} \left[ a\gamma_*^2 + \frac{4nG^2}{(M + \mu)\kappa_\sigma} \right] T^{(t)} + \frac{\eta}{a} \left\| w^{(t)} - w^\star \right\|_2^2.$$

We take $\beta_2 = 2$, $c_3 = c_4 = 0$, and apply Lem. 12 to achieve

$$\mathbb{E}_t \left[ V^{(t+1)} \right] - (1 - \tau^{-1})V^{(t)} \leq \left[ \tau^{-1} - \eta\mu + \eta a^{-1} + c_2 \right] \| w^{(t)} - w^\star \|_2^2$$

$$+ \left[ \tau^{-1} + \frac{3\eta^2}{2c_1} - \frac{1}{n} \right] c_1 S^{(t)}$$

$$+ \left[ \tau^{-1} + \frac{\eta G^2}{\bar{\nu}^2 c_2} \left( a\gamma_*^2 + \frac{4nG^2}{(M + \mu)\kappa_\sigma} \right) - \frac{1}{n} \right] c_2 T^{(t)}$$

$$+ \left[ - \frac{\eta}{2(M + \mu)\kappa_\sigma} + 3\eta^2 + \frac{c_1}{n} \right] Q^{(t)},$$

where $\tau > 0$ is a to-be-specified rate constant. We now need to set the various free parameters $a$, $c_1$, $c_2$, and $\eta$ to make each of the squared bracketed terms be non-positive. We enforce $\tau \geq 2n$ throughout. By setting

$$\eta = \frac{1}{12(\mu + M)\kappa_\sigma} \quad \text{and} \quad c_1 = \frac{n\eta}{4(\mu + M)\kappa_\sigma},$$

we have that the bracketed constants before $c_1 S^{(t)}$ and $Q^{(t)}$ vanish. Then, setting

$$a^{-1} = \frac{\mu}{2} \quad \text{and} \quad c_2 = \frac{1}{48(\kappa + 1)\kappa_\sigma}$$

make the bracketed constant before $\| w^{(t)} - w^\star \|_2^2$, assuming that we enforce

$$\tau \geq 48(\kappa + 1)\kappa_\sigma.$$

We turn to the final constant after substituting the values of $a$, $c_2$, and $\eta$. We need that

$$\frac{\eta G^2}{\bar{\nu}^2 c_2} \left( a\gamma_*^2 + \frac{8nG^2}{(M + \mu)\kappa_\sigma} \right) = \frac{8G^2}{\bar{\nu}^2\mu^2} \left( \gamma_*^2 + \frac{2nG^2}{(\kappa + 1)\kappa_\sigma} \right) \leq \frac{1}{2n},$$

which occurs when

$$\bar{\nu}^2 \geq \frac{16nG^2}{\mu^2} \left[ \gamma_*^2 + \frac{2nG^2}{(\kappa+1)\kappa_\sigma} \right].$$

Because $\gamma_*^2 \leq nG^2 \leq 2nG^2$, this is achieved when

$$\nu \geq \frac{8nG^2}{\mu},$$

completing the proof of the claim

$$\mathbb{E}_t \left[ V^{(t+1)} \right] \leq (1 - \tau^{-1}) V^{(t)}.$$

To complete the proof, we bound the initial terms. Because $c_3 = c_4 = 0$, we need only to bound $S^{(0)}$ and $T^{(0)}$.

$$S^{(0)} = \frac{1}{n} \sum_{i=1}^{n} \| n\rho_i^{(0)} \nabla r_i(z_i^{(0)}) - nq_i^\star \nabla r_i(w^\star) \|_2^2$$

$$= \frac{1}{n} \sum_{i=1}^{n} \| nq_i^{(0)} \nabla r_i(w^{(0)}) - nq_i^* \nabla r_i(w^\star) \|_2^2$$

$$\leq \frac{2}{n} \sum_{i=1}^{n} \| nq_i^{(0)} \nabla (r_i(w^{(0)}) - \nabla r_i(w^\star)) \|_2^2 + \frac{2}{n} \sum_{i=1}^{n} \| n(q_i^{(0)} - q_i^\star) \nabla r_i(w^\star) \|_2^2$$

$$\leq 2n \sum_{i=1}^{n} (q_i^{(0)})^2 M^2 \| w^{(0)} - w^\star \|_2^2 + 8nG^2 \| q^{(0)} - q^\star \|_2^2$$

$$\leq \left[ 2n \| \sigma \|_2^2 M^2 + \frac{8n^2 G^4}{\bar{\nu}^2} \right] \| w^{(0)} - w^\star \|_2^2$$

$$\leq \left[ 2n \| \sigma \|_2^2 M^2 + \mu^2/8 \right] \| w^{(0)} - w^\star \|_2^2 \leq 3nM^2 \| w^{(0)} - w^\star \|_2^2.$$

This means ultimately that

$$c_1 S^{(0)} \leq \frac{n^2}{16(1 + \kappa^{-1})^2 \kappa_\sigma^2} \| w^{(0)} - w^\star \|_2^2.$$

Next, we have

$$c_2 T^{(0)} = \frac{n}{48(\kappa+1)\kappa_\sigma} \left\| w^{(0)} - w^\star \right\|_2^2.$$

Thus, we can write

$$V^{(0)} \leq \left[ 1 + \frac{n^2}{16(1 + \kappa^{-1})^2 \kappa_\sigma^2} + \frac{n}{48(\kappa+1)\kappa_\sigma} \right] \left\| w^{(0)} - w^\star \right\|_2^2$$

$$\leq (1 + \sigma_n^{-1} + \sigma_n^{-2}) \left\| w^{(0)} - w^\star \right\|_2^2,$$

completing the proof. $\qquad\square$

### D.5.2 STEP 3B: ANALYZE SMALL SHIFT COST SETTING.

To describe the rate, define $\delta := nG^2/(\mu\bar{\nu})$. The quantity $\delta$ captures the effect of the primal regularizer $\mu$ and dual regularizer $\bar{\nu}$ as compared to the inherent continuity properties of the underlying losses.

**Theorem 16.** *Assume that $n \geq 2$ and that the shift cost $\bar{\nu} \leq 8nG^2/\mu$. The sequence of iterates produced by Algorithm 3 with*

$$\eta = \frac{1}{16n\mu} \min \left\{ \frac{1}{6[8\delta + (\kappa + 1)\kappa_\sigma]}, \frac{1}{4\delta^2 \max\{2n\kappa^2, \delta\}} \right\}$$

*we have*

$$\mathbb{E}_t \left[ V^{(t+1)} \right] \leq (1 - \tau^{-1})V^{(t)},$$

$$\mathbb{E}_t \left\| w^{(t)} - w^\star \right\|_2^2 \leq \left( 5 + 16\delta + \frac{6\kappa^2}{\sigma_n} \right) \exp\left(-t/\tau\right) \left\| w^{(0)} - w^\star \right\|_2^2$$

*for*

$$\tau = 32n \max \left\{ 6[8\delta + (\kappa + 1)\kappa_\sigma], 4\delta^2 \max\{2n\kappa^2, \delta\}, 1/16 \right\}.$$

*Proof.* First, we apply Lem. 11 with $q' = q^\star$, as well as Lem. 14, Lem. 12, and Lem. 13, set $c_4 = 1$, and consolidate all constants to write

$$\mathbb{E}_t \left[ V^{(t+1)} \right] - (1 - \tau^{-1})V^{(t)} \leq (\tau^{-1} - \eta\mu + c_2) \left\| w^{(t)} - w^\star \right\|_2^2 \tag{42}$$

$$+ \left[ \tau^{-1} - \frac{1}{n} + \frac{2\beta_1 G^2}{\bar{\nu}(M + \mu)\kappa_\sigma} + \left(1 - \frac{1}{n}\right) \frac{G^2 c_3}{2\bar{\nu}\mu n} \right] R^{(t)} \tag{43}$$

$$+ \left[ \tau^{-1} + \frac{1 + c_3}{c_1} \eta^2 (1 + \beta_2^{-1}) - \frac{1}{n} \right] c_1 S^{(t)} \tag{44}$$

$$+ \left[ \tau^{-1} + \frac{\eta G^2 n}{2c_2 \bar{\nu}} \beta_3^{-1} + \frac{c_3 \eta M^2}{c_2 \mu n} \left(1 - \frac{1}{n}\right) - \frac{1}{n} \right] c_2 T^{(t)} \tag{45}$$

$$+ \left[ \tau^{-1} + \frac{2\eta G^2 n}{c_3 \bar{\nu}} (1 + \beta_3) - \frac{1}{n} \right] c_3 U^{(t)} \tag{46}$$

$$+ \left[ -\frac{\eta \beta_1}{2(M + \mu)\kappa_\sigma} + \eta^2 (1 + c_3)(1 + \beta_2) + \frac{c_1}{n} \right] Q^{(t)}. \tag{47}$$

We first set $c_1 = \frac{n\eta\beta_1}{4(M+\mu)\kappa_\sigma}$ and $c_2 = \eta\mu/2$ to clean up (42) and (47). We also drop the terms $(1 - 1/n) \leq 1$. Then, we notice in (43) that to achieve

$$\frac{2\beta_1 G^2}{\bar{\nu}(M + \mu)\kappa_\sigma} \leq \frac{1}{4n},$$

we need that $\beta_1 \leq ((M + \mu)\kappa_\sigma)/(8nG^2/\bar{\nu})$. Combined with the requirement that $\beta_1 \in [0, 1]$, we set $\beta_1 = ((M + \mu)\kappa_\sigma)/(8nG^2/\bar{\nu} + (M + \mu)\kappa_\sigma)$. We set $\beta_2 = 2$, and can rewrite the expression

above.

$$\mathbb{E}_t\left[V^{(t+1)}\right] - (1-\tau^{-1})V^{(t)} \le \left(\tau^{-1} - \frac{\eta\mu}{2}\right)\left\|w^{(t)} - w^\star\right\|_2^2$$
$$+ \left[\tau^{-1} - \frac{3}{4n} + \frac{G^2 c_3}{2\bar{\nu}\mu n}\right]R^{(t)}$$
$$+ \left[\tau^{-1} + \frac{6(1+c_3)(M+\mu)\kappa_\sigma}{n\beta_1}\eta - \frac{1}{n}\right]c_1 S^{(t)}$$
$$+ \left[\tau^{-1} + \frac{G^2 n}{\mu\bar{\nu}}\beta_3^{-1} + \frac{c_3 M^2}{\mu^2 n} - \frac{1}{n}\right]c_2 T^{(t)}$$
$$+ \left[\tau^{-1} + \frac{2\eta G^2 n}{c_3\bar{\nu}}(1+\beta_3) - \frac{1}{n}\right]c_3 U^{(t)}$$
$$+ \left[-\frac{\eta\beta_1}{4(M+\mu)\kappa_\sigma} + 3\eta^2(1+c_3)\right]Q^{(t)}.$$

Next, set the learning rate to be

$$\eta \le \frac{\beta_1}{12(1+c_3)(M+\mu)\kappa_\sigma} \tag{48}$$

to cancel out $Q^{(t)}$ and achieve

$$\mathbb{E}_t\left[V^{(t+1)}\right] - (1-\tau^{-1})V^{(t)} \le \left(\tau^{-1} - \frac{\eta\mu}{2}\right)\left\|w^{(t)} - w^\star\right\|_2^2$$
$$+ \left[\tau^{-1} - \frac{3}{4n} + \frac{G^2 c_3}{2\bar{\nu}\mu n}\right]R^{(t)}$$
$$+ \left[\tau^{-1} - \frac{1}{2n}\right]c_1 S^{(t)}$$
$$+ \left[\tau^{-1} + \frac{G^2 n}{\mu\bar{\nu}}\beta_3^{-1} + \frac{c_3 M^2}{\mu^2 n} - \frac{1}{n}\right]c_2 T^{(t)}$$
$$+ \left[\tau^{-1} + \frac{2\eta G^2 n}{c_3\bar{\nu}}(1+\beta_3) - \frac{1}{n}\right]c_3 U^{(t)}.$$

Requiring now that $\tau \ge 2n$, we may also cancel the $S^{(t)}$ term. We substitute $\delta = nG^2/(\mu\bar{\nu})$ to achieve

$$\mathbb{E}_t\left[V^{(t+1)}\right] - (1-\tau^{-1})V^{(t)} \le \left(\tau^{-1} - \frac{\eta\mu}{2}\right)\left\|w^{(t)} - w^\star\right\|_2^2$$
$$+ \left[-\frac{1}{4n} + \frac{c_3\delta}{2n^2}\right]R^{(t)}$$
$$+ \left[-\frac{1}{2n} + \frac{\delta}{\beta_3} + \frac{c_3 M^2}{\mu^2 n}\right]c_2 T^{(t)}$$
$$+ \left[-\frac{1}{2n} + \frac{2\mu\eta\delta}{c_3}(1+\beta_3)\right]c_3 U^{(t)}.$$

It remains to select $c_3$ and $\beta_3$. As such, we set $\beta_3 = 4n\delta$ and use that $1 + 4n\delta \le 8n\delta$ when $n \ge 2$ and $\delta \ge 1/8$ as assumed, and so

$$
\begin{aligned}
\mathbb{E}_t \left[ V^{(t+1)} \right] - (1 - \tau^{-1}) V^{(t)} &\le \left( \tau^{-1} - \frac{\eta\mu}{2} \right) \left\| w^{(t)} - w^\star \right\|_2^2 \\
&+ \left[ -\frac{1}{4n} + \frac{c_3 \delta}{2n^2} \right] R^{(t)} \\
&+ \left[ -\frac{1}{4n} + \frac{c_3 \kappa^2}{n} \right] c_2 T^{(t)} \\
&+ \left[ -\frac{1}{2n} + \frac{16n\mu\eta\delta^2}{c_3} \right] c_3 U^{(t)}.
\end{aligned}
$$

We require now that

$$
c_3 = \frac{1}{2} \min \left\{ \frac{1}{2\kappa^2}, \frac{n}{\delta} \right\},
$$

which cancels $T^{(t)}$ and $R^{(t)}$, leaving

$$
\begin{aligned}
\mathbb{E}_t \left[ V^{(t+1)} \right] - (1 - \tau^{-1}) V^{(t)} &\le \left( \tau^{-1} - \frac{\eta\mu}{2} \right) \left\| w^{(t)} - w^\star \right\|_2^2 \\
&+ \left[ -\frac{1}{2n} + 32\mu\eta\delta^2 \max \left\{ 2n\kappa^2, \delta \right\} \right] c_3 U^{(t)}.
\end{aligned}
$$

From the above, we retrieve the requirement that

$$
\eta \le \frac{1}{64n\mu\delta^2 \max \left\{ 2n\kappa^2, \delta \right\}}. \tag{49}
$$

It now remains to set $\eta$. By substituting in the values for $\beta_1$ and $c_3$ into (48), we have that

$$
\begin{aligned}
\eta \overset{\text{want}}{\le} \frac{\beta_1}{12(1 + c_3)(M + \mu)\kappa_\sigma} &= \frac{1}{12(1 + c_3)[8\mu\delta + (M + \mu)\kappa_\sigma]} \\
&\ge \frac{1}{(12 + 6n/\delta)[8\mu\delta + (M + \mu)\kappa_\sigma]} \\
&\ge \frac{1}{(12 + 48n)[8\mu\delta + (M + \mu)\kappa_\sigma]} \\
&\ge \frac{1}{96n[8\mu\delta + (M + \mu)\kappa_\sigma]}.
\end{aligned}
$$

The combination of (49) and the above display yields

$$
\begin{aligned}
\eta &= \min \left\{ \frac{1}{96n[8\mu\delta + (M + \mu)\kappa_\sigma]}, \frac{1}{64n\mu\delta^2 \max \left\{ 2n\kappa^2, \delta \right\}} \right\} \\
&= \frac{1}{16n\mu} \min \left\{ \frac{1}{6[8\delta + (\kappa + 1)\kappa_\sigma]}, \frac{1}{4\delta^2 \max \left\{ 2n\kappa^2, \delta \right\}} \right\}.
\end{aligned}
$$

We need finally that $\tau \ge 2/(\mu\eta)$, resulting in the requirement

$$
\tau \ge 32n \max \left\{ 6[8\delta + (\kappa + 1)\kappa_\sigma], 4\delta^2 \max \left\{ 2n\kappa^2, \delta \right\} \right\}.
$$

This is achieved by setting

$$
\tau = 32n \max \left\{ 6[8\delta + (\kappa + 1)\kappa_\sigma], 4\delta^2 \max \left\{ 2n\kappa^2, \delta \right\}, 1/16 \right\}.
$$

completing the proof of the claim

$$
\mathbb{E}_t \left[ V^{(t+1)} \right] \le (1 - \tau^{-1}) V^{(t)}.
$$

Next, we bound the initial terms to achieve the final rate. First, we bound $\eta$ which is used in all of the terms. Because $\delta \geq 1/8$,

$$\eta \leq \frac{1}{16n\mu} \cdot \frac{1}{4\delta^2 \max\{2n\kappa^2, \delta\}} \leq \frac{1}{64n\mu\delta^3} \leq \frac{8}{n\mu}. \tag{50}$$

Then,

$$
\begin{aligned}
S^{(0)} &= \frac{1}{n} \sum_{i=1}^{n} \|n\rho_i^{(0)}\nabla r_i(z_i^{(0)}) - nq_i^\star \nabla r_i(w^\star)\|_2^2 \\
&= \frac{1}{n} \sum_{i=1}^{n} \|nq_i^{(0)}\nabla r_i(w^{(0)}) - nq_i^* \nabla r_i(w^\star)\|_2^2 \\
&\leq \frac{2}{n} \sum_{i=1}^{n} \|nq_i^{(0)}\nabla(r_i(w^{(0)}) - \nabla r_i(w^\star))\|_2^2 + \frac{2}{n} \sum_{i=1}^{n} \|n(q_i^{(0)} - q_i^\star)\nabla r_i(w^\star)\|_2^2 \\
&\leq 2n \sum_{i=1}^{n} (q_i^{(0)})^2 M^2 \|w^{(0)} - w^\star\|_2^2 + 8nG^2 \|q^{(0)} - q^\star\|_2^2 \\
&\leq \left[ 2n\|\sigma\|_2^2 M^2 + \frac{8n^2G^2}{\bar{\nu}^2} \right] \|w^{(0)} - w^\star\|_2^2 \\
&\leq \left[ 2n\|\sigma\|_2^2 M^2 + \mu^2/8 \right] \|w^{(0)} - w^\star\|_2^2 \leq 3nM^2 \|w^{(0)} - w^\star\|_2^2.
\end{aligned}
$$

Continuing with $\beta_1 \leq 1$ and (50),

$$
\begin{aligned}
c_1 S^{(0)} &= \frac{n\eta\beta_1}{4(M+\mu)\kappa_\sigma} S^{(0)} \\
&\leq \frac{2}{\mu(M+\mu)\kappa_\sigma} \cdot 3nM^2 \|w^{(0)} - w^\star\|_2^2 \\
&\leq \frac{6n\kappa^2}{(1+\kappa)\kappa_\sigma} \|w^{(0)} - w^\star\|_2^2 \\
&\leq \frac{6\kappa^2}{\sigma_n} \|w^{(0)} - w^\star\|_2^2.
\end{aligned}
$$

Next, we have $T^{(0)} = n \left\|w^{(0)} - w^\star\right\|_2^2$ and by (50),

$$
\begin{aligned}
c_2 T^{(0)} &= \frac{\eta\mu}{2} \cdot n \left\|w^{(0)} - w^\star\right\|_2^2 \\
&\leq 4 \left\|w^{(0)} - w^\star\right\|_2^2.
\end{aligned}
$$

Because $U^{(0)} = 0$, it is bounded trivially. For $R^{(0)}$, with $c_4 = 1$ we have

$$
\begin{aligned}
R^{(0)} &= 2n\eta(q^{\text{opt}}(\ell(w^{(0)})) - q^{\text{opt}}(\ell(w^\star)))^\top (\ell(w^{(0)}) - \ell(w^\star)) \\
&\leq \frac{2n\eta}{\bar{\nu}} \left\|\ell(w^{(0)}) - \ell(w^\star)\right\|_2^2 \\
&\leq \frac{2n^2\eta G^2}{\bar{\nu}} \left\|w^{(0)} - w^\star\right\|_2^2 \\
&\leq \frac{16nG^2}{\mu\bar{\nu}} \left\|w^{(0)} - w^\star\right\|_2^2 \\
&= 16\delta \left\|w^{(0)} - w^\star\right\|_2^2.
\end{aligned}
$$

Combining each of these terms together, we have that

$$V^{(0)} \leq \left(5 + 16\delta + \frac{6\kappa^2}{\sigma_n}\right) \left\|w^{(0)} - w^\star\right\|_2^2,$$

completing the proof. □

### D.6 PROOF OF MAIN RESULT

The objective is once again

$$
\begin{aligned}
F_\sigma(w) &= \max_{q \in \mathcal{P}(\sigma)} q^\top \ell(w) - \nu D_f(q\|\mathbf{1}_n/n) + \frac{\mu}{2}\|w\|_2^2 \\
&= \max_{q \in \mathcal{P}(\sigma)} q^\top \ell(w) - n\alpha_n\nu \frac{1}{n\alpha_n} D_f(q\|\mathbf{1}_n/n) + \frac{\mu}{2}\|w\|_2^2 \\
&= \max_{q \in \mathcal{P}(\sigma)} q^\top \ell(w) - n\alpha_n\nu\Omega(q) + \frac{\mu}{2}\|w\|_2^2 \\
&= \max_{q \in \mathcal{P}(\sigma)} q^\top \ell(w) - \bar{\nu}\Omega(q) + \frac{\mu}{2}\|w\|_2^2,
\end{aligned}
$$

where $\Omega(q) = D_f(q\|\mathbf{1}_n/n)/n\alpha_n$ is the penalty scaled to be 1-strongly convex and we simply notate $\bar{\nu} = n\alpha_n\nu$. The previous subsections give the convergence analysis in the cases of large and small values of $\bar{\nu}$. They are combined below.

**Theorem 1.** *Prospect with a small enough step size is guaranteed to converge linearly for all $\nu > 0$. If, in addition, the shift cost is $\nu \geq \Omega(G^2/\mu\alpha_n)$, then the sequence of iterates $(w^{(t)})_{t\geq 1}$ generated by Prospect and learning rate $\eta = (12\mu(1+\kappa)\kappa_\sigma)^{-1}$ converges linearly at a rate $\tau = 2\max\{n, 24\kappa_\sigma(\kappa+1)\}$, i.e.,*

$$\mathbb{E}\|w^{(t)} - w^\star\|_2^2 \leq (1 + \sigma_n^{-1} + \sigma_n^{-2})\exp(-t/\tau)\|w^{(0)} - w^\star\|_2^2.$$

*Proof.* Combine Thm. 15 (the analysis for $\bar{\nu} \geq 8nG^2/\mu$) and Thm. 16 (the analysis for $\bar{\nu} \leq 8nG^2/\mu$) to achieve convergence for any value of $\bar{\nu}$. Substitute $\bar{\nu} = n\alpha_n\nu$ so that the condition $\bar{\nu} \geq 8nG^2/\mu$ reads as $\nu \geq G^2/(\mu\alpha_n)$. □

## E   IMPROVING PROSPECT WITH MOREAU ENVELOPES

As mentioned in Sec. 3, we may want to generalize Prospect to non-smooth settings which arise either when the shift cost $\nu = 0$ or when the underlying losses $(\ell_i)$ are non-smooth. The former case is already addressed by Prop. 8 in Appx. B. The latter case can be handled by considering a variant of Prospect applied to the *Moreau envelope* of the losses, as defined below. Not only does this extend the algorithm to non-smooth losses, it also allows even in the smooth setting for a less stringent lower bound on $\nu$ required for the $O((n + \kappa_\sigma\kappa)\ln(1/\varepsilon))$ rate. The rest of this section contains necessary background, implementation details, and the adjustments to the analysis.

### E.1   OVERVIEW

We first describe the method and practical details of the implementation, followed by the convergence analysis in the next section.

**Notation.**   The Moreau envelope and the proximal (prox) operator of a convex function $f : \mathbb{R}^d \to \mathbb{R}$ are respectively defined for a constant $\eta > 0$ as

$$\mathcal{M}_\eta[f](w) = \min_{z\in\mathbb{R}^d}\left\{f(z) + \frac{1}{2\eta}\|w - z\|_2^2\right\}, \tag{51}$$

$$\mathrm{prox}_{\eta f}(w) = \arg\min_{z\in\mathbb{R}^d}\left\{f(z) + \frac{1}{2\eta}\|w - z\|_2^2\right\}. \tag{52}$$

A fundamental property is that the gradient of the Moreau envelope is related to the proximal operator:

$$\nabla \mathcal{M}_\eta[f](w) = \frac{1}{\eta}(w - \text{prox}_{\eta f}(w)). \tag{53}$$

For simplicity, we denote $\bar{\nu} = 2n\nu$.

**Algorithm Description.** The algorithm is nearly equivalent to Algorithm 3, but makes the following changes. We sample $i_t \sim q^{(t)}$ non-uniformly in the sense that $\mathbb{P}[i_t = i] = q_i^{(t)}$. We do not store an additional vector of weights $\rho^{(t)}$, and use $q^{(t)}$ in all associated steps. This does not change the expectation of the update direction or the control variate, but creates minor changes in the analysis of the variance term. In the iterate update, we replace the gradient descent-like update with

$$u^{(t)} = w^{(t)} + \eta(g_{i_t}^{(t)} - \bar{g}^{(t)})$$
$$w^{(t+1)} = \text{prox}_{\eta r_{i_t}}(u^{(t)}).$$

The vector $u^{(t)}$ adds the control variate to $w^{(t)}$ before passing it to the proximal operator. The second change is that the elements of the gradient table are updated using $j_t$ and the gradients of the Moreau envelope. That is,

$$g_{j_t}^{(t+1)} = \nabla \mathcal{M}_\eta[r_{j_t}]\big(w^{(t)} + \eta(g_{j_t}^{(t)} - \bar{g}^{(t)})\big),$$
$$g_j^{(t+1)} = g_j^{(t)} \text{ for } j \neq j_t.$$

Plugging these changes into Algorithm 3 produces the Prospect-Moreau variant.

**Implementation Details.** The proximal operators can be computed in closed form or algorithmically for common losses. We list here the implementations for some losses of interest. The proximal operators for the binary or multiclass logistic losses cannot be obtained in closed form, we approximate them by one Newton step.

*Squared loss.* For the squared loss, defined as $\ell(w) = \frac{1}{2}(w^\top x - y)^2$ for $x \in \mathbb{R}^d, y \in \mathbb{R}$, then

$$\text{prox}_{\eta \ell}(w) = w - \frac{\eta x}{1 + \eta \|x\|^2}\left(x^\top w - y\right).$$

*Binary logistic loss.* For the binary logistic loss defined for $x \in \mathbb{R}^d$, $y \in \{0, 1\}$, $w \in \mathbb{R}^d$ as $\ell(w) = -y\ln(\sigma(x^\top w)) - (1 - y)\ln(1 - \sigma(x^\top w)) = -yx^\top w + \ln(1 + e^{x^\top w})$, we approximate the proximal operator by one Newton step, whose formulation reduces to

$$\text{prox}_{\eta \ell}(w) \approx w - \frac{\eta g}{1 + \eta q\|x\|_2^2}x$$

*Multinomial logistic loss.* For the multinomial logistic loss of a linear model defined by $W$ on a sample $(x, y)$ as $\ell(W) = -y^\top Wx + \ln(\exp(Wx)^\top \mathbf{1})$. for $x \in \mathbb{R}^d, y \in \{0, 1\}^k, y^\top \mathbf{1} = 1, W \in \mathbb{R}^{k \times d}$, we consider approximating the proximal operator by one Newton-step, whose formulation reduces to

$$\text{prox}_{\eta \ell}(W) \approx W - \eta z^* x^\top$$
$$z^* = z_1 - \lambda^* z_2,$$
$$z_1 = -y \oslash z_3 + z_2, \ z_2 = \sigma(Wx) \oslash z_3, \ z_3 = (\mathbf{1} + \eta\|x\|_2^2\sigma(Wx)), \ \lambda^* = \frac{z_1^\top \mathbf{1}}{z_2^\top \mathbf{1}}.$$

*Regularized losses.* For a convex $\ell : \mathbb{R}^d \to \mathbb{R}$, define $r(w) = \ell(w) + (\mu/2)\|w\|^2$. Then, we have,

$$\text{prox}_{\eta r}(w) = \text{prox}_{\frac{\eta \ell}{1 + \eta \mu}}\left(\frac{w}{1 + \eta \mu}\right).$$

### E.2 CONVERGENCE ANALYSIS

Prospect-Moreau satisfies the following convergence bound. Recall that $\gamma_\star = \|\nabla\ell(w^\star)\|_2$.

**Theorem 17.** *Suppose the smoothing parameter $\bar{\nu}$ is set large enough as*

$$\bar{\nu} \geq \frac{\gamma_* G}{M} \min\left\{\sqrt{\frac{2n\kappa}{4\kappa_\sigma^* - 1}}, 2\kappa\right\},$$

*and define a constant*

$$\tau = 2 + \max\{2(n-1),\ \kappa(4\kappa_\sigma^* - 1)\},$$

*for $\kappa_\sigma^* = \sigma_n/\sigma_1$. Then, the sequence of iterates $(w^{(t)})$ generated by Prospect-Moreau with learning rate $\eta = M^{-1}\min\{1/(4\kappa_\sigma^* - 1), \kappa/(n-1)\}$ satisfies*

$$\mathbb{E}\left\|w^{(t)} - w^\star\right\|_2^2 \leq (n + 3/2)\exp(-t/\tau)\left\|w^{(0)} - w^\star\right\|_2^2.$$

We now prove Thm. 17, using similar techniques as in Appx. D.

**Additional Notation.** Further, we define $w_i^* = w^* + \eta\nabla r_i(w^*)$. By analyzing the first-order conditions of the prox, it is easy to see that

$$\text{prox}_{\eta r_i}(w_i^*) = w^*. \tag{54}$$

We will use the Lyapunov function

$$V^{(t)} = \left\|w^{(t)} - w^*\right\|^2 + c_1\sum_{i=1}^n\left\|z_i^{(t)} - w^*\right\|^2 + \frac{c_2}{M^2}\sum_{i=1}^n\left\|g_i^{(t)} - \nabla r_i(w^*)\right\|^2. \tag{55}$$

The first step is to analyze the effect of the update on $w^{(t)}$ as the first term of the Lyapunov function.

**Proposition 18.** *The iterates of Prospect-Moreau satisfy*

$$(1 + \mu\eta)\mathbb{E}_t\left\|w^{(t+1)} - w^*\right\|^2 \leq \left\|w^{(t)} - w^*\right\|^2 + 2\eta^2\sigma_n\sum_{i=1}^n\left\|g_i^{(t)} - \nabla r_i(w^*)\right\|^2$$

$$+ \frac{2\eta^2\gamma_*^2 G^2}{\bar{\nu}^2}\sum_{i=1}^n\left\|z_i^{(t)} - w^*\right\|^2$$

$$- \eta^2\left(1 + \frac{1}{M\eta}\right)\sigma_1\sum_{i=1}^n\left\|\nabla\mathcal{M}_\eta[r_i]\left(w^{(t)} - \eta(g_i^{(t)} - \bar{g}^{(t)})\right) - \nabla r_i(w^*)\right\|^2.$$

*Proof.* We use the co-coercivity of the prox operator (Thm. 31) to get

$$(1 + \mu\eta)\mathbb{E}_t\left\|w^{(t+1)} - w^*\right\|^2 = (1 + \mu\eta)\mathbb{E}_t\left\|\text{prox}_{\eta r_{i_t}}(u^{(t)}) - \text{prox}_{\eta r_{i_t}}(w_{i_t}^*)\right\|^2$$

$$\leq \mathbb{E}_t\langle u^{(t)} - w_{i_t}^*, \text{prox}_{\eta r_{i_t}}(u^{(t)}) - \text{prox}_{\eta r_{i_t}}(w_{i_t}^*)\rangle$$

$$= \mathbb{E}_t\langle u^{(t)} - w_{i_t}^*, w^{(t+1)} - w^*\rangle$$

$$= \underbrace{\mathbb{E}_t\langle u^{(t)} - w_{i_t}^{(t)}, w^{(t)} - w^*\rangle}_{=:\mathcal{T}_1} + \underbrace{\mathbb{E}_t\langle u^{(t)} - w_{i_t}^*, w^{(t+1)} - w^{(t)}\rangle}_{=:\mathcal{T}_2}, \tag{56}$$

where we added and subtracted $w^{(t)}$ in the last step.

For the first term, we observe that $\mathbb{E}_t[u^{(t)}] = w^{(t)}$ and $\mathbb{E}_t[w_{i_t}^*] = w^* + \eta\mathbb{E}_t[\nabla r_{i_t}(w^*)]$ so that

$$\mathcal{T}_1 = \left\langle\mathbb{E}_t[u^{(t)} - w_{i_t}^*], w^{(t)} - w^*\right\rangle = \left\|w^{(t)} - w^*\right\|^2 + \eta\left\langle\mathbb{E}_t[\nabla r_{i_t}(w^*)], w^{(t)} - w^*\right\rangle. \tag{57}$$

For $\mathcal{T}_2$, note that

$$w^{(t+1)} - w^{(t)} = -\eta \left( \nabla \mathcal{M}_\eta[r_{i_t}](u^{(t)}) - g_{i_t}^{(t)} + \bar{g}^{(t)} \right) .$$

We manipulate $\mathcal{T}_2$ to set ourselves up to apply co-coercivity of prox-gradient by adding and subtracting $\nabla \mathcal{M}_\eta[r_{i_t}](w_{i_t}^*)$ as follows:

$$\mathcal{T}_2 = -\eta \, \mathbb{E}_t \langle u^{(t)} - w_{i_t}^{(t)}, \nabla \mathcal{M}_\eta[r_{i_t}](u^{(t)}) - g_{i_t}^{(t)} + \bar{g}^{(t)} \rangle$$

$$= \underbrace{-\eta \, \mathbb{E}_t \langle u^{(t)} - w_{i_t}^*, \nabla \mathcal{M}_\eta[r_{i_t}](u^{(t)}) - \nabla \mathcal{M}_\eta[r_{i_t}](w_{i_t}^*) \rangle}_{=:\mathcal{T}_2'}$$

$$\underbrace{-\eta \, \mathbb{E}_t \langle u^{(t)} - w_{i_t}^*, \nabla \mathcal{M}_\eta[r_{i_t}](w_{i_t}^*) - g_{i_t}^{(t)} + \bar{g}^{(t)} \rangle}_{=:\mathcal{T}_2''} .$$

Now, co-coercivity of the prox-gradient (Thm. 32) of the $M$-smooth function $r_{i_t}$ gives

$$\mathcal{T}_2' \leq -\eta^2 \left( 1 + \frac{1}{M\eta} \right) \mathbb{E}_t \left\| \nabla \mathcal{M}_\eta[r_{i_t}](u^{(t)}) - \nabla \mathcal{M}_\eta[r_{i_t}](w_{i_t}^*) \right\|^2 . \tag{58}$$

Next, we use $u^{(t)} = w^{(t)} + \eta(g_{i_t}^{(t)} - \bar{g}^{(t)})$, and $w_i^* = w^* + \eta \nabla r_i(w^*)$ and $\nabla \mathcal{M}_\eta[r_i](w_i^*) = \nabla r_i(w^*)$ to get

$$\mathcal{T}_2'' = -\eta \, \mathbb{E}_t \langle w^{(t)} - w^* - \eta(\nabla r_{i_t}(w^*) - g_{i_t}^{(t)} + \bar{g}^{(t)}), \nabla r_{i_t}(w^*) - g_{i_t}^{(t)} + \bar{g}^{(t)} \rangle$$

$$= -\eta \langle w^{(t)} - w^*, \mathbb{E}_t[\nabla r_{i_t}(w^*)] \rangle + \eta^2 \, \mathbb{E}_t \left\| g_{i_t}^{(t)} - \bar{g}^{(t)} - \nabla r_{i_t}(w^*) \right\|^2 ,$$

where we used that $\mathbb{E}_t[g_{i_t}^{(t)}] = \bar{g}^{(t)}$. Next, we use $\|x + y\|^2 \leq 2 \|x\|^2 + 2 \|y\|^2$ for any vectors $x, y$ and $\mathbb{E}\|X - \mathbb{E}[X]\|^2 \leq \mathbb{E}\|X\|^2$ for any random vector $X$ to get

$$\mathcal{T}_2'' \leq -\eta \langle w^{(t)} - w^*, \mathbb{E}_t[\nabla r_{i_t}(w^*)] \rangle + 2\eta^2 \, \mathbb{E}_t \left\| g_{i_t}^{(t)} - \nabla r_{i_t}(w^*) \right\|^2 + 2\eta^2 \, \|\mathbb{E}_t[\nabla r_{i_t}(w^*)]\|^2 . \tag{59}$$

Plugging (59), (58), and (59) into (56) gives us

$$(1 + \mu\eta)\mathbb{E}_t \left\| w^{(t+1)} - w^* \right\|^2 \leq \left\| w^{(t)} - w^* \right\|^2 + 2\eta^2 \, \mathbb{E}_t \left\| g_{i_t}^{(t)} - \nabla r_{i_t}(w^*) \right\|^2 + 2\eta^2 \, \|\mathbb{E}_t[\nabla r_{i_t}(w^*)]\|^2$$

$$- \eta^2 \left( 1 + \frac{1}{M\eta} \right) \mathbb{E}_t \left\| \nabla \mathcal{M}_\eta[r_{i_t}](u^{(t)}) - \nabla r_{i_t}(w^*) \right\|^2 . \tag{60}$$

Next, we note that $\mathcal{P}(\sigma) \subset [\sigma_1, \sigma_n]^n$ to get,

$$\mathbb{E}_t \|g_{i_t} - \nabla r_{i_t}(w^*)\|^2 = \sum_{i=1}^n q_i^{(t)} \|g_i - \nabla r_i(w^*)\|^2 \leq \sigma_n \sum_{i=1}^n \|g_i - \nabla r_i(w^*)\|^2 , \quad \text{and}$$

$$\mathbb{E}_t \left\| \nabla \mathcal{M}_\eta[r_{i_t}](u^{(t)}) - \nabla r_{i_t}(w^*) \right\|^2 = \sum_{i=1}^n q_i^{(t)} \left\| \nabla \mathcal{M}_\eta[r_i]\big(w^{(t)} - \eta(g_i^{(t)} - \bar{g}^{(t)})\big) - \nabla r_i(w^*) \right\|^2$$

$$\geq \sigma_1 \sum_{i=1}^n \left\| \nabla \mathcal{M}_\eta[r_i]\big(w^{(t)} - \eta(g_i^{(t)} - \bar{g}^{(t)})\big) - \nabla r_i(w^*) \right\|^2 .$$

Moreover, we also have that

$$\|\mathbb{E}_t[\nabla r_{i_t}(w^*)]\|^2 = \left\|\nabla\ell(w^\star)^\top(q^{\mathrm{opt}}(l^{(t)}) - q^{\mathrm{opt}}(\ell(w^\star)))\right\|^2$$

$$\gamma_*^2 \left\|q^{\mathrm{opt}}(l^{(t)}) - q^{\mathrm{opt}}(\ell(w^\star)))\right\|_2^2$$

$$\leq \frac{\gamma_*^2 G^2}{\bar{\nu}^2} \sum_{i=1}^n \left\|z_i^{(t)} - w^*\right\|^2.$$

Plugging these back into (60) completes the proof. $\qquad\square$

Next, we analyze the other two terms of the Lyapunov function. The proof is trivial, so we omit it.

**Proposition 19.** *We have,*

$$\mathbb{E}_t\left[\sum_{i=1}^n \left\|z_i^{(t+1)} - w^*\right\|^2\right] = (1 - n^{-1})\sum_{i=1}^n \left\|z_i^{(t)} - w^*\right\|^2 + \left\|w^{(t)} - w^*\right\|^2,$$

$$\mathbb{E}_t\left[\sum_{i=1}^n \left\|g_i^{(t+1)} - \nabla r_i(w^*)\right\|^2\right] = (1 - n^{-1})\sum_{i=1}^n \left\|g_i^{(t)} - \nabla r_i(w^*)\right\|^2$$

$$+ \frac{1}{n}\sum_{i=1}^n \left\|\nabla\mathcal{M}_\eta[r_i]\big(w^{(t)} - \eta(g_i^{(t)} - \bar{g}^{(t)})\big) - \nabla r_i(w^*)\right\|^2.$$

We are now ready to prove Thm. 17.

*Proof of Thm. 17.* Let $\tau > 1$ be a constant to be determined later and let $\Gamma := \gamma_*^2 G^2/(M^2\bar{\nu}^2)$ denote the effect of the smoothing. Combining Props. 18 and 19, we can write

$$
\begin{aligned}
\mathbb{E}_t[V^{(t)}] - (1 - \tau^{-1})V^{(t)} \leq &-\left\|w^{(t)} - w^*\right\|^2\left(\frac{\mu\eta}{1 + \mu\eta} - c_1 - \tau^{-1}\right) \\
&- \sigma_1\sum_{i=1}^n\left\|\nabla\mathcal{M}_\eta[r_i]\big(w^{(t)} - \eta(g_i^{(t)} - \bar{g}^{(t)})\big) - \nabla r_i(w^*)\right\|^2\left(\frac{\eta^2(1 + (M\eta)^{-1})}{1 + \mu\eta} - \frac{c_2}{n\sigma_1 M^2}\right) \\
&- \sum_{i=1}^n\left\|z_i^{(t)} - w^*\right\|^2\left(c_1(n^{-1} - \tau^{-1}) - \frac{2\eta^2\gamma_*^2 G^2}{(1 + \mu\eta)\bar{\nu}^2}\right) \\
&- \sum_{i=1}^n\left\|g_i^{(t)} - \nabla r_i(w^*)\right\|^2\left(\frac{c_2}{M^2}(n^{-1} - \tau^{-1}) - \frac{2\eta^2\sigma_n}{1 + \mu\eta}\right).
\end{aligned}
$$

$$(61)$$

Let $\eta = b/M$. Our goal is to set the constants $b, c_1, c_2, \tau > 0$ so that the right side above is non-positive and $\tau$ is as small as possible. We will require $\tau \geq 2n$ so that $n^{-1} - \tau^{-1} \geq (2n)^{-1}$. Thus, we can have the right side nonpositive with

$$\frac{b}{b + \kappa} - c_1 - \tau^{-1} \geq 0 \tag{62a}$$

$$b(b + 1) \geq \frac{c_2}{n\sigma_1}\left(1 + \frac{b}{\kappa}\right) \tag{62b}$$

$$\frac{c_1}{2n} - \frac{2b^2\Gamma}{1 + b/\kappa} \geq 0 \tag{62c}$$

$$\frac{c_2}{2n} - \frac{2b^2\sigma_n}{1 + b/\kappa} \geq 0. \tag{62d}$$

Let us set $c_1 = \tau^{-1}$. By setting $c_2 = 4\kappa n \sigma_n b^2/(b+\kappa)$, we ensure that (62d) is satisfied. Next, we satisfy (62a) with

$$\frac{b}{b+\kappa} = 2\tau^{-1} \quad \Longleftrightarrow \quad b = \frac{2\kappa}{\tau - 2}.$$

Now, (62b) is an inequality only in $\tau$. It is satisfied with $\tau \geq \tau_* := 2 + 2\kappa(4\kappa_\sigma^* - 1)$. This lets us fix $\tau = \max\{2n, \tau_*\}$ throughout, which leads to the value of $\eta$ as claimed in the theorem statement. Finally, (62c) requires

$$\frac{4n\kappa^2\Gamma}{\tau - 2} \leq 1 \quad \Longleftrightarrow \quad \bar{\nu} \geq \frac{\sqrt{n}\kappa\gamma_* G}{M} \min\left\{\sqrt{\frac{2}{\kappa(4\kappa_\sigma^* - 1)}}, \frac{2}{\sqrt{n}}\right\}.$$

Thus, under these conditions, the right-hand side of (61) is non-negative. Iterating (61) over $t$ updates, we get

$$\mathbb{E}[V^{(t)}] = (1 - \tau^{-1})^t V^{(0)} \leq \exp(-t/\tau)V^{(0)}.$$

To complete the proof, we note that $c_1 \leq 1/(2n)$ and

$$c_2 = \frac{4\kappa n \sigma_n b^2}{b+\kappa} = 8\frac{\kappa\kappa_\sigma}{\tau}b \leq 8\frac{\kappa\kappa_\sigma}{\kappa(4\kappa_\sigma^* - 1)}\frac{1}{\kappa_\sigma^* - 1} \leq \frac{8}{9}.$$

This lets us use the fact that $\nabla r_i$ is $M$-Lipschitz to bound

$$V^{(0)} = \left\|w^{(0)} - w^*\right\| + c_1 \sum_{i=1}^n \left\|w^{(0)} - w^*\right\|^2 + \frac{c_2}{M^2}\sum_{i=1}^n \left\|\nabla r_i(w^{(0)}) - \nabla r_i(w^*)\right\|^2$$

$$\leq (n + 3/2)\left\|w^{(0)} - w^*\right\|^2.$$

$\square$

# F    IMPROVING THE DIRECT SADDLE-POINT BASELINE

In Sec. 4 we compared against the existing method of Palaniappan & Bach (2016), which views our objective (8) in its min-max form directly and applies variance reduction techniques to both the primal and dual sequences. In order to make the comparison more convincing, we also improve this method both theoretically and empirically by utilizing different learning rates for the primal and dual sequences. In this section, we provide a new convergence analysis for this improved two-hyperparameter variant, which we dub *SaddleSAGA*, under the $\chi^2$-divergence penalty.

## F.1    OVERVIEW

As in Appx. E, we begin with the additional notation and description of the algorithm, followed by the convergence analysis.

**Notation.** For simplicity, we denote $\bar{\nu} = 2n\nu$, and consider directly the min-max problem

$$\min_{w \in \mathbb{R}^d} \max_{q \in \mathcal{P}(\sigma)} \left[\Psi(w, q) := q^\top \ell(w) + \frac{\mu}{2}\|w\|_2^2 - \frac{\bar{\nu}}{2}\|q - \mathbf{1}_n/n\|_2^2\right]. \tag{63}$$

Note that the function $\Psi$ is strongly convex in its first argument and strongly concave in its second argument. A pair $(w^\star, q^\star)$ is called a saddle point of the convex-concave function $\Psi$ if

$$\max_{q \in \mathcal{P}(\sigma)} \Psi(w^\star, q) \leq \Psi(w^\star, q^\star) \leq \min_{w \in \mathbb{R}^d} \Psi(w, q^\star).$$

In our setting, we can verify that the pair $w^\star = \arg\min F_\sigma$ and $q^\star = q^{\mathrm{opt}}(\ell(w^\star))$ is the unique saddle point of $\Psi$.

**Algorithm Description.** The algorithm makes use of proximal operators (as described in Appx. E in addition), which is defined for a convex function $f : \mathbb{R}^d \to \mathbb{R}$, and $x \in \mathbb{R}^d$ as

$$\text{prox}_f(x) = \arg\min_{y \in \mathbb{R}^d} \; f(y) + \frac{1}{2}\|x - y\|_2^2.$$

The method is nearly equivalent to Algorithm 3, but applies the update

$$v^{(t)} = nq_{i_t}^{(t)}\nabla\ell_{i_t}(w^{(t)}) - (n\rho_{i_t}^{(t)}g_{i_t}^{(t)}) - \bar{g}^{(t)})$$
$$w^{(t+1)} = \text{prox}_{\eta\mu\|\cdot\|_2^2}(w^{(t)} - \eta v^{(t)})$$

to the primal iterates and

$$\pi^{(t)} = n\ell_{i_t}(w^{(t)})e_{i_t} - (nl_{i_t}^{(t)}e_{i_t} - l^{(t)})$$
$$q^{(t+1)} = \text{prox}_{\iota_{\mathcal{P}(\sigma)}+\delta\bar{\nu}\|\cdot-\mathbf{1}_n/n\|_2^2/2}(q^{(t)} - \delta\pi^{(t)})$$

to the dual iterates, where $\delta > 0$ is the dual learning rate. The vector $\pi^{(t)}$ plays the role of an update direction, and the proximal update on $q^{(t+1)}$ can be solved with the PAV algorithm, as seen in Appx. C. Overall, the time and space complexity is identical to that of Prospect.

**Rate of Convergence.** We prove the following rate of convergence for SaddleSAGA.

**Theorem 20.** *The iterates $(w^{(t)}, q^{(t)})$ of SaddleSAGA with learning rates*

$$\eta = \min\left\{\frac{1}{\mu}, \frac{1}{6(L\kappa_\sigma + 2G^2n/\bar{\nu})}\right\}, \quad \delta = \min\left\{\frac{1}{\bar{\nu}}, \frac{\mu}{8n^2G^2}\right\}$$

*converge linearly to the saddle point of (63). In particular, for non-trivial regularization $\mu\bar{\nu} \le 8n^2G^2$ and $\mu \le 6(L\kappa_\sigma+2G^2n/\bar{\nu})$, the number of iterations $t$ to get $\|w^{(t)}-w^\star\|_2^2+c\|q^{(t)}-q^\star\|_2^2 \le \varepsilon$ (for some constant c) is at most*

$$O\left(\left(n + \kappa\kappa_\sigma + \frac{n^2G^2}{\mu\bar{\nu}}\right)\ln\frac{1}{\varepsilon}\right).$$

The proof of this statement is given as Cor. 26 later in this section. To compare to the original variant, the rate obtained by Palaniappan & Bach (2016) in terms of our problem's constants is

$$O\left(\left(n + \frac{nG^2}{\mu\bar{\nu}} + n\kappa^2\right)\ln\frac{1}{\varepsilon}\right).$$

Compared to this, the rate we prove for SaddleSAGA improves $\kappa^2$ to $\kappa\kappa_\sigma$ while suffering an additional factor of $n$ in the $n^2G^2/(\mu\bar{\nu})$ term. Empirical comparisons between SaddleSAGA and the original algorithm are given in Appx. I. As compared to Prospect, SaddleSAGA matches the rate of Thm. 1 only when the shift cost $\bar{\nu}$ is large enough.

## F.2 CONVERGENCE ANALYSIS

In the following, we denote by $\mathbb{E}_t[\cdot]$ the expecation of a quantity according to the randomness of $i_t$ conditioned on $w^{(t)}, q^{(t)}$. Throughout the proof, we consider that the losses are $L$-smooth and $G$-Lipschitz continuous.

**Evolution of Distances-to-Optimum.** We start by using the contraction properties of the proximal operator to bound the evolution of the distances to the saddle point $(w^\star, q^\star)$.

**Lemma 21.** *We have,*

$$
\mathbb{E}_t\left[\|w^{(t+1)} - w^\star\|_2^2\right] \le \frac{1}{(1+\eta\mu)^2}\Big(\|w^{(t)} - w^\star\|_2^2
$$
$$
- 2\eta(\nabla(q^{(t)\top}\ell)(w^{(t)}) - \nabla(q^{\star\top}\ell)(w^\star))^\top(w^{(t)} - w^\star)
$$
$$
+ \eta^2\mathbb{E}_t\left[\|v^{(t)} - \nabla(q^{\star\top}\ell)(w^\star)\|_2^2\right]\Big)
$$
$$
\mathbb{E}_t\left[\|q^{(t+1)} - q^\star\|_2^2\right] \le \frac{1}{(1+\delta\bar\nu)^2}\Big(\|q^{(t)} - q^\star\|_2^2
$$
$$
+ 2\delta(\ell(w^{(t)}) - \ell(w^\star))^\top(q^{(t)} - q^\star)
$$
$$
+ \delta^2\mathbb{E}_t\left[\|\pi^{(t)} - \ell(w^\star)\|_2^2\right]\Big).
$$

*Proof.* By considering the first-order optimality conditions of the problem one verifies that $w^\star, q^\star$ satisfy for any $\eta, \delta$,

$$
w^\star = \mathrm{prox}_{\eta\mu\|\cdot\|_2^2/2}(w^\star - \eta\nabla(q^{\star\top}\ell)(w^\star)), \quad q^\star = \mathrm{prox}_{\iota_{\mathcal{P}(\sigma)}+\delta\bar\nu\|\cdot-\mathbf{1}_n/n\|_2^2/2}(q^\star + \delta\ell(w^\star)).
$$

Recall that the proximal operator of a $c$-strongly convex function $h$ is contractive such that $\|\mathrm{prox}_h(z) - \mathrm{prox}_h(z')\|_2 \le \frac{1}{1+c}\|z - z'\|_2$. In our case, it means that

$$
\|w^{(t+1)} - w^\star\|_2 \le \frac{1}{1+\eta\mu}\|w^{(t)} - \eta v^{(t)} - (w^\star - \eta\nabla(q^{\star\top}\ell)(w^\star))\|_2,
$$
$$
\|q^{(t+1)} - q^\star\|_2 \le \frac{1}{1+\delta\bar\nu}\|q^{(t)} + \delta\pi^{(t)} - (q^\star + \delta\ell(w^\star))\|_2.
$$

By taking the squared norm, the expectation, expanding the squared norms and using that $\mathbb{E}_t\left[v^{(t)}\right] = \nabla(q^{(t)\top}\ell)(w^{(t)}), \mathbb{E}_t\left[\pi^{(t)}\right] = \ell(w^{(t)})$, we get the result. $\square$

**Evolution of Variance Term.** We consider the evolution of the additional variance term added to the dual variables.

**Lemma 22.** *We have for any $\beta_2 > 0$,*

$$
\mathbb{E}_t\left[\|\pi^{(t)} - \ell(w^\star)\|_2^2\right] \le (n + (n-1)\beta_2)nG^2\|w^{(t)} - w^\star\|_2^2
$$
$$
+ (n-1)(1 + \beta_2^{-1})\|\ell(w^\star) - l^{(t)}\|_2^2.
$$

*Proof.* As in the proof of Lem. 10, we have for some $\beta_2 > 0$,

$$
\mathbb{E}_t\left[\|\pi^{(t)} - \ell(w^\star)\|_2^2\right] = \mathbb{E}_{i_t}\Big[\|(n\ell_{i_t}(w^{(t)}) - n\ell_{i_t}(w^\star))e_{i_t}
$$
$$
+ (n\ell_{i_t}(w^\star) - n\ell_{i_t}(z_{i_t}^{(t)}))e_{i_t} - (\ell(w^\star) - l^{(t)})\|_2^2\Big]
$$
$$
\le -\beta_2\|\ell(w^{(t)}) - \ell(w^\star)\|_2^2
$$
$$
+ (1 + \beta_2)\mathbb{E}_t\left[\|(n\ell_{i_t}(w^{(t)}) - n\ell_{i_t}(w^\star))e_{i_t}\|_2^2\right]
$$
$$
+ (1 + \beta_2^{-1})\mathbb{E}_t\left[\|(n\ell_{i_t}(w^\star) - n\ell_{i_t}(z_{i_t}^{(t)}))e_{i_t}\|_2^2\right]
$$
$$
- (1 + \beta_2^{-1})\|\ell(w^\star) - l^{(t)}\|_2^2
$$
$$
= (n + (n-1)\beta_2)\|\ell(w^{(t)}) - \ell(w^\star)\|_2^2
$$
$$
+ (n-1)(1 + \beta_2^{-1})\|\ell(w^\star) - l^{(t)}\|_2^2
$$
$$
\le (n + (n-1)\beta_2)nG^2\|w^{(t)} - w^\star\|_2^2
$$
$$
+ (n-1)(1 + \beta_2^{-1})\|\ell(w^\star) - l^{(t)}\|_2^2.
$$

$\square$

**Incorporating Smoothness and Convexity of the Losses.** The improved algorithm we developed here, for the purpose of a fair comparison to our own algorithm, differs from the original one from Palaniappan & Bach (2016) by Cor. 24 stemming from Lem. 23. We exploit the smoothness and convexity of the losses to get a negative term $-\mathbb{E}_t\left[\|nq_{i_1}\nabla\ell_{i_t}(w^{(t)}) - nq_{i_t}^*\nabla\ell_{i_t}(w^\star)\|_2^2\right]$ used to temper the variance of the primal updates at the price of an additional positive term $\|q^{(t)} - q^\star\|_2^2$. The sum of both being positive we can dampen the effect of the additional positive term $\|q^{(t)} - q^\star\|_2^2$ at the price of getting a less negative term $-\mathbb{E}_t\left[\|nq_{i_1}\nabla\ell_{i_t}(w^{(t)}) - nq_{i_t}^*\nabla\ell_{i_t}(w^\star)\|_2^2\right]$.

**Lemma 23.** *For any $q_1, q_2 \in \mathcal{P}(\sigma)$, $w_1, w_2 \in \mathbb{R}^d$, we have,*

$$(q_1 - q_2)^\top(\ell(w_1) - \ell(w_2)) - (\nabla(q_1^\top\ell)(w_1) - \nabla(q_2^\top\ell)(w_2))^\top(w_1 - w_2)$$

$$\leq -\frac{1}{2Ln\sigma_{\max}}\left(\mathbb{E}_{i\sim\text{Unif}[n]}\left[\|nq_{1,i}\nabla\ell_i(w_1) - nq_{2,i}\nabla\ell_i(w_2)\|_2^2 + \|nq_{1,i}\nabla\ell(w_2) - nq_{2,i}\nabla\ell(w_1)\|_2^2\right]\right)$$

$$+ \frac{G^2}{L\sigma_{\max}}\|q_1 - q_2\|_2^2.$$

*Proof.* For any $q \in \mathcal{P}(\sigma)$ and any $w, v \in \mathbb{R}^d$, we have by smoothness and convexity of $q_i\ell_i$, for $q_i > 0$

$$q_i\ell_i(v) \geq q_i\ell_i(w) + q_i\nabla\ell_i(w)^\top(v - w) + \frac{1}{2Lq_i}\|q_i\nabla\ell_i(w) - q_i\nabla\ell_i(v)\|_2^2 \tag{64}$$

$$\geq q_i\ell_i(w) + q_i\nabla\ell_i(w)^\top(v - w) + \frac{1}{2Ln^2\sigma_{\max}}\|nq_i\nabla\ell_i(w) - nq_i\nabla\ell_i(v)\|_2^2. \tag{65}$$

Note that the second inequality is then true even if $q_i = 0$, since in that case all terms are 0. Therefore, for any $q_1, q_2 \in \mathcal{P}(\sigma)$, and any $w_1, w_2$, we have

$$q_1^\top\ell(w_2) \geq q_1^\top\ell(w_1) + \nabla(q_1^\top\ell)(w_1)^\top(w_2 - w_1) + \frac{1}{2Ln\sigma_{\max}}\mathbb{E}_{i\sim\text{Unif}[n]}\left[\|nq_{1,i}\nabla\ell_i(w_1) - nq_{1,i}\nabla\ell_i(w_2)\|_2^2\right],$$

$$q_2^\top\ell(w_1) \geq q_2^\top\ell(w_2) + \nabla(q_2^\top\ell)(w_2)^\top(w_1 - w_2) + \frac{1}{2Ln\sigma_{\max}}\mathbb{E}_{i\sim\text{Unif}[n]}\left[\|nq_{2,i}\nabla\ell(w_1) - nq_{2,i}\nabla\ell(w_2)\|_2^2\right].$$

Combining these inequalities, we get

$$-(q_1 - q_2)^\top(\ell(w_1) - \ell(w_2)) + (\nabla(q_1^\top\ell)(w_1) - \nabla(q_2^\top\ell)(w_2))^\top(w_1 - w_2)$$

$$\geq \frac{1}{2Ln\sigma_{\max}}\left(\mathbb{E}_{i\sim\text{Unif}[n]}\left[\|nq_{1,i}\nabla\ell_i(w_1) - nq_{1,i}\nabla\ell_i(w_2)\|_2^2 + \|nq_{2,i}\nabla\ell(w_1) - nq_{2,i}\nabla\ell(w_2)\|_2^2\right]\right).$$

For any 4 vectors $a, b, c, d$,

$$\|a - b\|_2^2 + \|c - d\|_2^2 = \|a - c\|_2^2 + \|b - d\|_2^2 - 2(a - d)^\top(b - c).$$

Applying this for $a = q_{1,i}\nabla\ell_i(w_1)$, $b = q_{i,1}\nabla\ell_i(w_2)$, $c = q_{2,i}\nabla\ell_i(w_2)$, $d = q_{2,i}\nabla\ell_i(w_1)$, we get

$$-(q_1 - q_2)^\top(\ell(w_1) - \ell(w_2)) + (\nabla(q_1^\top\ell)(w_1) - \nabla(q_2^\top\ell)(w_2))^\top(w_1 - w_2)$$

$$\geq \frac{1}{2Ln\sigma_{\max}}\Big(\mathbb{E}_{i\sim\text{Unif}[n]}\left[\|nq_{1,i}\nabla\ell_i(w_1) - nq_{2,i}\nabla\ell_i(w_2)\|_2^2 + \|nq_{1,i}\nabla\ell(w_2) - nq_{2,i}\nabla\ell(w_1)\|_2^2\right]$$

$$- 2n^2\mathbb{E}_{i\sim\text{Unif}[n]}\left[(q_{1,i} - q_{2,i})^2\nabla\ell_i(w_1)^\top\nabla\ell_i(w_2)\right]\Big).$$

Reorganizing the terms and bounding $\nabla\ell_i(w_1)^\top\nabla\ell_i(w_2)$ by $G^2$ we get the result. $\qquad\square$

**Corollary 24.** *We have for any $\alpha \in [0,1]$*

$$\mathbb{E}_t \left[ \frac{(1+\eta\mu)^2}{\eta} \|w^{(t+1)} - w^\star\|_2^2 + \frac{(1+\delta\bar{\nu})^2}{\delta} \|q^{(t+1)} - q^\star\|_2^2 \right]$$

$$\leq \eta^{-1} \|w^{(t)} - w^\star\|_2^2 + \left( \delta^{-1} + \frac{2\alpha G^2}{L\sigma_{\max}} \right) \|q^{(t)} - q^\star\|_2^2$$

$$+ \eta \mathbb{E}_t \left[ \|v^{(t)} - \nabla(q^{\star\top}\ell)(w^\star)\|_2^2 \right] + \delta \mathbb{E}_t \left[ \|\pi^{(t)} - \ell(w^\star)\|_2^2 \right]$$

$$- \frac{\alpha}{Ln\sigma_{\max}} \mathbb{E}_t \left[ \|nq_{i_1} \nabla\ell_{i_t}(w^{(t)}) - nq_{i_t}^* \nabla\ell_{i_t}(w^\star)\|_2^2 \right].$$

*Proof.* Follows from Lem. 23 □

**Lyapunov Function and Overall Convergence.** Thm. 25 shows that an appropriately defined Lyapunov function incorporating the distances to the optima, decrease exponentially.

**Theorem 25.** *Define the Lyapunov function*

$$V^{(t)} = \frac{(1+\eta\mu)^2}{\eta} \|w^{(t)} - w^\star\|_2^2 + \frac{(1+\delta\bar{\nu})^2}{\delta} \|q^{(t)} - q^\star\|_2^2$$

$$+ c_1 \sum_{i=1}^n \|n\rho_i^{(t)} \nabla\ell_i(z_i^{(t)}) - nq_i^* \nabla\ell_i(w^\star)\|_2^2 + \frac{c_2}{G^2} \|l^{(t)} - \ell(w^\star)\|_2^2,$$

*with $c_1 = \frac{n}{2(L\kappa_\sigma + 2G^2 n/\bar{\nu})}$ and $c_2 = \frac{\mu}{2}$ with $\kappa_\sigma = n\sigma_{\max}$. By taking*

$$\eta = \min\left\{ \frac{1}{\mu}, \frac{1}{6(L\kappa_\sigma + 2G^2 n/\bar{\nu})} \right\}, \quad \delta = \min\left\{ \frac{1}{\bar{\nu}}, \frac{\mu}{8n^2 G^2} \right\},$$

*we have*

$$\mathbb{E}_t \left[ V^{(t+1)} \right] \leq (1 - \tau^{-1}) V^{(t)},$$

*for some $\tau > 1$. In particular, for small regularizations, i.e., $\mu\bar{\nu} \leq 8n^2 G^2$ and $\mu \leq 6(L\kappa_\sigma + 2G^2 n/\bar{\nu})$, we have*

$$\tau = \max\left\{ 2n, 4 + \frac{24L\kappa_\sigma}{\mu} + \frac{48G^2 n}{\mu\bar{\nu}}, 2 + \frac{16G^2 n^2}{\bar{\nu}\mu} \right\}.$$

*Proof.* Let us denote

$$T^{(t)} = \frac{1}{n} \sum_{i=1}^n \|n\rho_i^{(t)} \nabla\ell_i(z_i^{(t)}) - nq_i^* \nabla\ell_i(w^\star)\|_2^2, \quad S^{(t)} = \|l^{(t)} - \ell(w^\star)\|_2^2,$$

we have,

$$\mathbb{E}_t \left[ T^{(t+1)} \right] \leq \frac{1}{n^2} \sum_{i=1}^n \|nq_i^{(t)} \nabla\ell_i(w^{(t)}) - nq_i^* \nabla\ell_i(w^\star)\|_2^2 + \left( 1 - \frac{1}{n} \right) T^{(t)},$$

$$\mathbb{E}_t \left[ S^{(t+1)} \right] \leq G^2 \|w^{(t)} - w^\star\|_2^2 + \left( 1 - \frac{1}{n} \right) S^{(t)}.$$

By combining Cor. 24, Lem. 10, Lem. 22 we have, denoting $\kappa_\sigma = n\sigma_{\max}$,

$$\mathbb{E}_t\left[V^{(t+1)}\right] \le \left(\eta^{-1} + \delta(n + (n-1)\beta_2)nG^2 + c_2\right)\|w^{(t)} - w^\star\|_2^2$$

$$+ \left(\delta^{-1} + \frac{2\alpha n G^2}{L\kappa_\sigma}\right)\|q^{(t)} - q^\star\|_2^2$$

$$+ \left(\eta(1 + \beta_1) + \frac{c_1}{n} - \frac{\alpha}{Ln\sigma_{\max}}\right)\mathbb{E}_{i\sim\mathrm{Unif}[n]}\left[\|nq_{i_1}\nabla\ell_{i_t}(w^{(t)}) - nq_{i_t}^*\nabla\ell_{i_t}(w^\star)\|_2^2\right]$$

$$+ \left(\eta(1 + \beta_1^{-1}) + c_1\left(1 - \frac{1}{n}\right)\right)\frac{1}{n}\sum_{i=1}^n\|n\rho_i^{(t)}\nabla\ell_i(z_i^{(t)}) - nq_i^*\nabla\ell_i(w^\star)\|_2^2$$

$$+ \left(\delta(n-1)(1 + \beta_2^{-1}) + \frac{c_2}{G^2}\left(1 - \frac{1}{n}\right)\right)\|\ell(w^\star) - l^{(t)}\|_2^2.$$

Therefore for some $\tau > 1$, we have

$$\mathbb{E}_t\left[V^{(t+1)}\right] - (1 - \tau^{-1})V^{(t)} \le K_1\|w^{(t)} - w^\star\|_2^2 + K_2\|q^{(t)} - q^\star\|_2^2$$

$$+ K_3\mathbb{E}_{i\sim\mathrm{Unif}[n]}\left[\|nq_{i_1}\nabla\ell_{i_t}(w^{(t)}) - nq_{i_t}^*\nabla\ell_{i_t}(w^\star)\|_2^2\right]$$

$$+ K_4\frac{1}{n}\sum_{i=1}^n\|n\rho_i^{(t)}\nabla\ell_i(z_i^{(t)}) - nq_i^*\nabla\ell_i(w^\star)\|_2^2 + K_5\|\ell(w^\star) - l^{(t)}\|_2^2,$$

with,

$$K_1 = \frac{(1 + \eta\mu)^2}{\eta}\left(\frac{1 + \eta\left((n + (n-1)\beta_2)nG^2\delta + c_2\right)}{(1 + \eta\mu)^2} - (1 - \tau^{-1})\right)$$

$$K_2 = \frac{(1 + \delta\bar{\nu})^2}{\delta}\left(\frac{1 + 2\delta\alpha G^2 n/(L\kappa_\sigma)}{(1 + \delta\bar{\nu})^2} - (1 - \tau^{-1})\right)$$

$$K_3 = \eta(1 + \beta_1) + \frac{c_1}{n} - \frac{\alpha}{L\kappa_\sigma}$$

$$K_4 = c_1\left(\eta(1 + \beta_1^{-1})\frac{1}{c_1} + \left(1 - \frac{1}{n}\right) - (1 - \tau^{-1})\right)$$

$$K_5 = \frac{c_2}{G^2}\left(\delta(n-1)(1 + \beta_2^{-1})\frac{G^2}{c_2} + \left(1 - \frac{1}{n}\right) - (1 - \tau^{-1})\right).$$

Fix $\beta_1 = 2, \beta_2 = 1$. Denote also $\bar{\eta} = \frac{\eta\mu}{1 + \eta\mu} \in (0, 1)$ and $\bar{\delta} = \frac{\delta\bar{\nu}}{1 + \delta\bar{\nu}} \in (0, 1)$ with e.g. $\eta = \frac{\bar{\eta}}{\mu(1 + \bar{\eta})}$. We have then for $c_1/n = \alpha/(2L\kappa_\sigma)$ and $c_2 = \mu/2$,

$$K_1 \le \eta\mu^2\bar{\eta}\left(\bar{\eta}^2 - \left(1 - \frac{2n^2G^2\delta}{\mu}\right)\bar{\eta} + \tau^{-1}\right)$$

$$K_2 \le \delta\bar{\nu}^2\bar{\delta}\left(\bar{\delta}^2 - 2\left(1 - \frac{\alpha G^2 n}{L\kappa_\sigma\bar{\nu}}\right)\bar{\delta} + \tau^{-1}\right)$$

$$K_3 = 3\eta - \frac{\alpha}{2L\kappa_\sigma}$$

$$K_4 = c_1\left(3\eta\frac{L\kappa_\sigma}{n\alpha} - \frac{1}{n} + \tau^{-1}\right)$$

$$K_5 \le \frac{c_2}{G^2}\left(\delta\frac{4nG^2}{\mu} - \frac{1}{n} + \tau^{-1}\right).$$

We can further take $3\eta \leq \alpha/(2L\kappa_\sigma)$ and $\delta \leq \mu/(8n^2G^2)$. By imposing the constraint $\tau \geq 2n$, we can simplify

$$K_1 \leq \eta\mu^2\bar{\eta}\left(\bar{\eta}^2 - \frac{3}{4}\bar{\eta} + \tau^{-1}\right)$$

$$K_2 \leq \delta\bar{\nu}^2\bar{\delta}\left(\bar{\delta}^2 - 2\left(1 - \frac{\alpha G^2 n}{L\kappa_\sigma\bar{\nu}}\right)\bar{\delta} + \tau^{-1}\right)$$

$$K_3 \leq 0, K_4 \leq 0, K_5 \leq 0.$$

Recall that $\alpha$ must be chosen in $[0, 1]$. Taking then

$$\alpha = \frac{L\kappa_\sigma}{L\kappa_\sigma + 2G^2n/\bar{\nu}} \leq \frac{L\kappa_\sigma\bar{\nu}}{2G^2n},$$

we get

$$K_1 \leq \eta\mu^2\bar{\eta}\left(\bar{\eta}^2 - \frac{3}{4}\bar{\eta} + \tau^{-1}\right), \quad K_2 \leq \delta\bar{\nu}^2\bar{\delta}\left(\bar{\delta}^2 - \bar{\delta} + \tau^{-1}\right).$$

By taking $\eta \leq 1/\mu$, $\delta \leq 1/\bar{\nu}$, we get $\bar{\eta} \leq 1/2$, $\bar{\delta} \leq 1/2$ and so $\bar{\eta}^2 - \frac{3}{4}\bar{\eta} \leq -\frac{1}{4}\bar{\eta}$ and $\bar{\delta}^2 - \bar{\delta} \leq -\frac{1}{2}\bar{\delta}$. Therefore taking

$$\eta = \min\left\{\frac{1}{\mu}, \frac{1}{6(L\kappa_\sigma + 2G^2n/\bar{\nu})}\right\}, \quad \delta = \min\left\{\frac{1}{\bar{\nu}}, \frac{\mu}{8n^2G^2}\right\},$$

we get $K_i \leq 0$ for all $i$ as long as $\tau \geq \max\{2n, 4/\bar{\eta}, 2/\bar{\delta}\}$. In our case,

$$\frac{4}{\bar{\eta}} = \begin{cases} 4\left(1 + \frac{6L\kappa_\sigma}{\mu} + \frac{12G^2n}{\mu\bar{\nu}}\right) & \text{if } \mu \leq 6(L\kappa_\sigma + 2G^2n/\bar{\nu}), \\ 8 & \text{otherwise}, \end{cases}$$

$$\frac{2}{\bar{\delta}} = \begin{cases} 2\left(1 + \frac{8G^2n^2}{\bar{\nu}\mu}\right) & \text{if } \mu\bar{\nu} \leq 8n^2G^2, \\ 4 & \text{otherwise}. \end{cases}$$

The result follows. $\qquad\square$

**Corollary 26.** *Under the setting of Thm. 25, after $t$ iterations of SaddleSAGA, we have*

$$\mathbb{E}\left[\frac{(1 + \eta\mu)^2}{\eta}\|w^{(t)} - w^\star\|_2^2 + \frac{(1 + \delta\bar{\nu})^2}{\delta}\|q^{(t)} - q^\star\|_2^2\right]$$

$$\leq \exp(-t/\tau)\left(\frac{(1 + \eta\mu)^2}{\eta}\|w^{(0)} - w^\star\|_2^2 + \frac{(1 + \delta\bar{\nu})^2}{\delta}\|q^{(0)} - q^\star\|_2^2\right.$$

$$\left. + c_1n^2\sum_{i=1}^n \|nq_i^{(0)}\nabla\ell_i(w^{(0)}) - q_i^\star\nabla\ell_i(w^\star)\|_2^2 + \frac{c_2}{G^2}\|\ell(w^{(0)}) - \ell(w^\star)\|_2^2\right).$$

## G  TECHNICAL RESULTS FROM CONVEX ANALYSIS

Herein, we collect several results, mostly from Nesterov (2018), used throughout the paper. In the following, let $\|\cdot\|$ denote an arbitrary norm on $\mathbb{R}^d$ and let $\|\cdot\|_*$ denote its associated dual norm.

The first concerns $L$-smooth function, or those with $L$-Lipschitz continuous gradient.

**Theorem 27.** *(Nesterov, 2018, Theorem 2.1.5) The conditions below are considered for any $x, y \in \mathbb{R}^d$ and $\alpha \in [0, 1]$. The following are equivalent for a differentiable function $f : \mathbb{R}^d \to \mathbb{R}$.*

1. *$f$ is convex and $L$-smooth with respect to $\|\cdot\|$.*

2. *$0 \leq f(y) - f(x) - \langle\nabla f(x), y - x\rangle \leq \frac{L}{2}\|x - y\|^2$.*

3. *$f(x) + \langle\nabla f(x), y - x\rangle + \frac{1}{2L}\|\nabla f(x) - \nabla f(y)\|_*^2 \leq f(y)$.*

4. $\frac{1}{L} \|\nabla f(x) - \nabla f(y)\|_*^2 \leq \langle \nabla f(x) - \nabla f(y), x - y \rangle$.

5. $0 \leq \langle \nabla f(x) - \nabla f(y), x - y \rangle \leq L \|x - y\|^2$.

Next, we detail the properties of strongly convex functions.

**Theorem 28.** *(Nesterov, 2018, Theorem 2.1.10) If $f : \mathbb{R}^d \to \mathbb{R}$ is $\mu$-strongly convex and differentiable, then for any $x, y \in \mathbb{R}^d$,*

- $f(y) \leq f(x) + \langle f(x), y - x \rangle + \frac{1}{2\mu} \|\nabla f(x) - \nabla f(y)\|_*^2$.

- $\langle \nabla f(x) - \nabla f(y), x - y \rangle \leq \frac{1}{\mu} \|\nabla f(x) - \nabla f(y)\|_*^2$.

- $\mu \|x - y\| \leq \|\nabla f(x) - \nabla f(y)\|_*$.

Finally, functions that are both smooth and strongly convex enjoy a number of relevant primal-dual properties.

**Theorem 29.** *(Nesterov, 2018, Theorem 2.1.12) If $f$ is both $L$-smooth and $\mu$-strongly convex, then for any $x, y \in \mathbb{R}^d$,*

$$- \langle \nabla f(x), x - y \rangle = -\frac{\mu L}{\mu + L} \|x - y\|^2 - \frac{1}{\mu + L} \|\nabla f(x) - \nabla f(y)\|^2 - \langle \nabla f(y), x - y \rangle. \qquad (66)$$

**Lemma 30.** *Let $f : \mathbb{R}^d \to \mathbb{R}$ be $\mu$-strongly convex and $M$-smooth. Then, we have for any $w, v \in \mathbb{R}^d$,*

$$f(v) \geq f(w) + \nabla f(w)^\top (v - w) + \frac{1}{2(M + \mu)} \|\nabla f(w) - \nabla f(v)\|_2^2 + \frac{\mu}{4} \|w - v\|_2^2.$$

*Proof.* The function $g = f - \mu \| \cdot \|_2^2 / 2$ is convex and $M - \mu$ smooth. Hence, we have by line 3 of Thm. 27 for any $w, v \in \mathbb{R}^d$,

$$g(v) \geq g(w) + \nabla g(w)^\top (v - w) + \frac{1}{2(M - \mu)} \|\nabla g(v) - \nabla g(w)\|_2^2.$$

Expanding $g$ and $\nabla g$, we get

$$f(v) \geq f(w) + \nabla f(w)^\top (v - w) + \frac{1}{2(M - \mu)} \|\nabla f(w) - \nabla f(v)\|_2^2$$
$$+ \frac{\mu M}{2(M - \mu)} \|w - v\|_2^2 - \frac{\mu}{M - \mu} (\nabla f(w) - \nabla f(v))^\top (w - v).$$

Using Young's inequality, that is, $a^\top b \leq \frac{\alpha}{2} \|a\|_2^2 + \frac{\alpha^{-1}}{2} \|b\|_2^2$, we have

$$f(v) \geq f(w) + \nabla f(w)^\top (v - w) + \frac{1 - \alpha\mu}{2(M - \mu)} \|\nabla f(w) - \nabla f(v)\|_2^2$$
$$+ \frac{\mu(M - \alpha^{-1})}{2(M - \mu)} \|w - v\|_2^2.$$

Taking $\alpha = \frac{2}{\mu + M}$ gives the claim. $\qquad \square$

We state a few properties of the prox operator.

**Theorem 31** (Co-coercivity of the prox). *If $f : \mathbb{R}^d \to \mathbb{R}$ is $\mu$-strongly convex, then we have for any constant $\eta > 0$ that*

$$\langle x - y, \mathrm{prox}_{\eta f}(x) - \mathrm{prox}_{\eta f}(y) \rangle \geq (1 + \eta\mu) \left\| \mathrm{prox}_{\eta f}(x) - \mathrm{prox}_{\eta f}(y) \right\|^2.$$

The same result applied to the convex conjugate $f^\star$ of $f$ and noting that $\nabla \mathcal{M}_\eta[f](x) = \mathrm{prox}_{f^\star/\eta}(x/\eta)$ gives the following result:

| Dataset | $d$ | $n_{\text{train}}$ | $n_{\text{test}}$ | Task | Source |
|---------|-----|---------|--------|------|--------|
| yacht | 6 | 244 | 62 | Regression | UCI |
| energy | 8 | 614 | 154 | Regression | UCI |
| concrete | 8 | 824 | 206 | Regression | UCI |
| kin8nm | 8 | 6,553 | 1,639 | Regression | OpenML |
| power | 4 | 7,654 | 1,914 | Regression | UCI |
| diabetes | 33 | 4,000 | 1,000 | Binary Classification | Fairlearn |
| acsincome | 202 | 4,000 | 1,000 | Regression | Fairlearn |
| amazon | 535 | 10,000 | 10,000 | Multiclass Classification | WILDS |
| iwildcam | 9420 | 20,000 | 5,000 | Multiclass Classification | WILDS |

Table 2: Dataset attributes and dimensionality $d$, train sample size $n_{\text{train}}$, and test sample size $n_{\text{test}}$.

**Theorem 32** (Co-coercivity of the prox). *If $f : \mathbb{R}^d \to \mathbb{R}$ is $L$-smooth, then we have for any constant $\eta > 0$ that*

$$\langle x - y, \nabla \mathcal{M}_\eta[f](x) - \nabla \mathcal{M}_\eta[f](y) \rangle \geq \eta \left( 1 + \frac{1}{L\eta} \right) \| \nabla \mathcal{M}_\eta[f](x) - \nabla \mathcal{M}_\eta[f](y) \|^2 .$$

**Lemma 33.** *For a convex function $f : \mathbb{R} \to \mathbb{R} \cup \{+\infty\}$, if $x_1 \geq x_2$ and $y_2 \geq y_1$, then*

$$f(y_1 - x_1) + f(y_2 - x_2) \geq f(y_2 - x_1) + f(y_1 - x_2).$$

*Proof.* First, observe that

$$y_2 - x_2 \geq y_2 - x_1 \geq y_1 - x_1 \text{ and } y_2 - x_2 \geq y_1 - x_2 \geq y_1 - x_1.$$

Thus, $y_2 - x_1$ and $y_1 - x_2$ both lie between $y_2 - x_2$ and $y_1 - x_1$ and can be expressed as a convex combination of the two endpoints, that is

$$y_2 - x_1 = \alpha(y_2 - x_2) + (1 - \alpha)(y_1 - x_1)$$
$$y_1 - x_2 = \beta(y_2 - x_2) + (1 - \beta)(y_1 - x_1)$$

for some $\alpha, \beta \in [0, 1]$. By solving for $\alpha$ we get $\alpha = 1 - \beta$. Apply the definition of convexity to get

$$f(y_2 - x_1) \leq \alpha f(y_2 - x_2) + (1 - \alpha) f(y_1 - x_1)$$
$$f(y_1 - x_2) \leq (1 - \alpha) f(y_2 - x_2) + \alpha f(y_1 - x_1).$$

Sum both inequalities to achieve the desired result. $\qquad\square$

**Lemma 34.** *Define for $l \in \mathbb{R}^n$,*

$$h(l) = \max_{q \in \mathcal{P}(\sigma)} l^\top q - \frac{\bar{\nu}}{2} \| q - \mathbf{1}_n / n \|_2^2.$$

*The function $h$ is $1/\bar{\nu}$-smooth and convex such that for any $l, l' \in \mathbb{R}^n$,*

$$\bar{\nu} \| \nabla h(l) - \nabla h(l') \|_2^2 \leq (\nabla h(l) - \nabla h(l'))^\top (l - l') \leq \frac{1}{\bar{\nu}} \| l - l' \|_2^2.$$

# H EXPERIMENTAL DETAILS

## H.1 TASKS & OBJECTIVES

In all settings, we consider supervised learning tasks specified by losses of the form

$$\ell_i(w) = h(y_i, w^\top \varphi(x_i)),$$

where we consider an input $x_i \in \mathcal{X}$, a feature map $\varphi : \mathcal{X} \to \mathbb{R}^d$, and a label $y_i \in \mathcal{Y}$. The function $h : \mathcal{Y} \times \mathbb{R} \to \mathbb{R}$ measures the error between the true label and another value which is the prediction in regression and the logit probabilities of the associated classes in classification. In the regression tasks, $\mathcal{Y} = \mathbb{R}$ and we used the squared loss

$$\ell_i(w) = \frac{1}{2}(y_i - w^\top \phi(x_i))^2 \,.$$

For binary classification, we have $\mathcal{Y} = \{-1, 1\}$, denoting a negative and positive class. We used the binary logistic loss

$$\ell_i(w) = -y_i x_i^\top w + \ln(1 + e^{x_i^\top w}) \,.$$

For multiclass classification, $\mathcal{Y} = \{1, \ldots, C\}$ where $C$ is the number of classes. We used the multinomial logistic loss:

$$\ell_i(w) = -\ln p_{y_i}(x_i; w), \text{ where } p_{y_i}(x_i; w) := \frac{\exp\left(w_{\cdot y}^\top x_i\right)}{\sum_{y'=1}^{C} \exp\left(w_{\cdot y'}^\top x_i\right)}, \ w \in \mathbb{R}^{d \times C}$$

The design matrix $(\varphi(x_1), \ldots, \varphi(x_n)) \in \mathbb{R}^{n \times d}$ is standardized to have columns with zero mean and unit variance, and the estimated mean and variance from the training set is used to standardize the test sets as well. Our final objectives are of the form

$$F_\sigma(w) = \max_{q \in \mathcal{P}(\sigma)} \sum_{i=1}^{n} q_i \ell_i(w) - \nu n \left\| q - \mathbf{1}_n/n \right\|_2^2 + \frac{\mu}{2} \left\| w \right\|_2^2$$

for shift cost $\nu \geq 0$ and regularization constant $\mu \geq 0$.

## H.2 DATASETS

We detail the datasets used in the experiments. If not specified below, the input space $\mathcal{X} = \mathbb{R}^d$ and $\varphi$ is the identity map. The sample sizes, dimensions, and source of the datasets are summarized in Tab. 2, where $d$ refers to the dimension of each $\varphi(x_i)$.

(a) `yacht`: prediction of the residuary resistance of a sailing yacht based on its physical attributes Tsanas & Xifara (2012).

(b) `energy`: prediction of the cooling load of a building based on its physical attributes Baressi Segota et al. (2020).

(c) `concrete`: prediction of the compressive strength of a concrete type based on its physical and chemical attributes Yeh (2006).

(d) `kin8nm`: prediction of the distance of an 8 link all-revolute robot arm to a spatial endpoint (Akujuobi & Zhang, 2017).

(e) `power`: prediction of net hourly electrical energy output of a power plant given environmental factors (Tüfekci, 2014).

(f) `diabetes`: prediction of readmission for diabetes patients based on 10 years worth of clinical care data at 130 US hospitals (Rizvi et al., 2014).

(g) `acsincome`: prediction of income of US adults given features compiled from the American Community Survey (ACS) Public Use Microdata Sample (PUMS) (Ding et al., 2021).

(h) `amazon`: prediction of the review score of a sentence taken from Amazon products. Each input $x \in \mathcal{X}$ is a sentence in natural language and the feature map $\varphi(x) \in \mathbb{R}^d$ is generated by the following steps:
   - A BERT neural network Devlin et al. (2019) (fine-tuned on $10,000$ held-out examples) is applied to the text $x_i$, resulting in vector $x_i'$.
   - The $x_1', \ldots, x_n'$ are normalized to have unit norm.
   - Principle Components Analysis (PCA) is applied, resulting in 105 components that explain $99\%$ of the variance, resulting in vectors $x_i'' \in \mathbb{R}^{105}$. The $d$ in Tab. 2 refers to the total dimension of the parameter vectors for all 5 classes.

(i) `iwildcam`: prediction of an animal or flora in an image from wilderness camera traps, with heterogeneity in illumination, camera angle, background, vegetation, color, and relative animal frequencies Beery et al. (2020). Each input $x \in \mathcal{X}$ is an image the feature map $\varphi(x) \in \mathbb{R}^d$ is generated by the following steps:

- A ResNet50 neural network He et al. (2016) that is pretrained on ImageNet Deng et al. (2009) is applied to the image $x_i$, resulting in vector $x_i'$.
- The $x_1', \ldots, x_n'$ are normalized to have unit norm.
- Principle Components Analysis (PCA) is applied, resulting in $d = 157$ components that explain $99\%$ of the variance. The $d$ in Tab. 2 refers to the total dimension of the parameter vectors for all 60 classes.

### H.3 HYPERPARAMETER SELECTION

We fix a minibatch size of 64 SGD and SRDA and an epoch length of $N = n$ for LSVRG. For SaddleSAGA we consider three schemes for selecting the primal and dual learning rates that reduce to searching for a single parameter $\eta > 0$, as described in Appx. I. In practice, the regularization parameter $\mu$ and shift cost $\nu$ are tuned by a statistical metric, i.e. generalization error as measured on a validation set. We study the optimization performance of the methods for multiple values of each in Appx. I.

For the tuned hyperparameters, we use the following method. Let $k \in \{1, \ldots, K\}$ be a seed that determines algorithmic randomness. This corresponds to sampling a minibatch without replacement for SGD and SRDA and a single sampled index for SaddleSAGA, LSVRG, and Prospect. Letting $\mathcal{L}_k(\eta)$ denote the average value of the training loss of the last ten passes using learning rate $\eta$ and seed $k$, the quantity $\mathcal{L}(\eta) = \frac{1}{K}\sum_{k=1}^{K}\mathcal{L}_k(\eta)$ was minimized to select $\eta$. The learning rate $\eta$ is chosen in the set $\{1 \times 10^{-4}, 3 \times 10^{-4}, 1 \times 10^{-3}, 3 \times 10^{-3}, 1 \times 10^{-2}, 3 \times 10^{-2}, 1 \times 10^{-1}, 3 \times 10^{-1}, 1 \times 10^{0}, 3 \times 10^{0}\}$, with two orders of magnitude lower numbers used in `acsincome` due to its sparsity. We discard any learning rates that cause the optimizer to diverge for any seed.

### H.4 COMPUTE ENVIRONMENT

No GPUs were used in the study; Experiments were run on a CPU workstation with an Intel i9 processor, a clock speed of 2.80GHz, 32 virtual cores, and 126G of memory. The code used in this project was written in Python 3 using the PyTorch and Numba packages for automatic differentiation and just-in-time compilation, respectively.

## I ADDITIONAL EXPERIMENTS

**Varying Risk Parameters.** We study the effect of varying the risk parameters, that is $(p, b, \gamma)$ for the $p$-CVaR (Equation (21)), $b$-extremile (Equation (22)), $\gamma$-ESRM (Equation (23)), choosing the spectrum to increase the condition number $\kappa_\sigma = n\sigma_n$ compared to the experiments in the main text. We use $p = 0.25$, $b = 2.5$, and $\gamma = 1/e^{-2}$ to generate "hard" version of the superquantile, extremile, and ESRM. Fig. 8 plots the corresponding training curves for four datasets of varying sample sizes: `yacht`, `energy`, `concrete`, and `iwildcam`. We see that the comparison of methods is the same as the original methods, that is that Prospect performs the best or close to best in terms of optimization trajectories. Except on `concrete`, SaddleSAGA generally matches the performance of Prospect. The trajectory of LSVRG is noticeably noisier than on the original settings; we hypothesize that the bias accrued by this epoch-based algorithm is exacerbated by the skewness in the spectrum, as mentioned in Mehta et al. (2023, Proposition 1).

**Lowering or Removing Shift Cost.** A relevant setting is the low or no shift cost regime, as this allows the adversary to make arbitrary distribution shifts (while still constrained to $\mathcal{P}(\sigma)$). These settings correspond to $\nu = 10^{-3}$ and $\nu = 0$, respectively. The low-cost experiment is displayed in Fig. 9 while Fig. 10 displays these curves for the no-cost experiment. When $\nu = 0$, the optimization problem can equivalently be written as

$$\min_{w \in \mathbb{R}^d}\left[\max_{q \in \mathcal{P}(\sigma)} q^\top \ell(w) + \frac{\mu}{2}\|w\|_2^2 = \sum_{i=1}^{n}\sigma_i \ell_{(i)}(w) + \frac{\mu}{2}\|w\|_2^2\right].$$

In this case, we always have that $q^{\mathrm{opt}}(l) = (\sigma_{\pi^{-1}(1)}, \ldots, \sigma_{\pi^{-1}(n)})$, where $\pi$ sorts $l$. Here, $w$ is chosen to optimize a linear combination of order statistics of the losses. In the low shift cost settings, performance trends are qualitatively similar to those seen from $\nu = 1$. Interestingly, for the no-cost

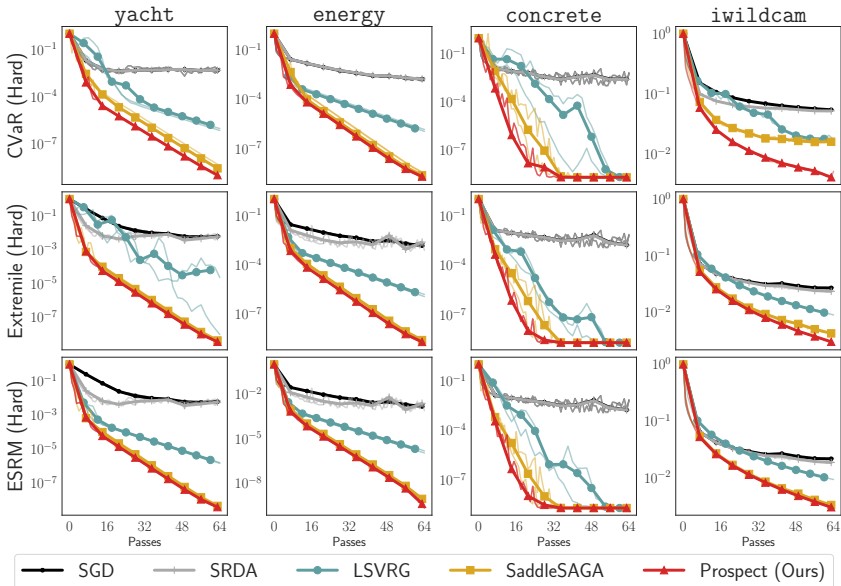

Figure 8: **Harder risk parameter settings.** Each row represents a different "hard" variant of the superquantile, extremile, and ESRM spectra. Columns represent different datasets. Suboptimality (9) is measured on the $y$-axis while the $x$-axis measures the total number of gradient evaluations made divided by $n$, i.e. the number of passes through the training set.

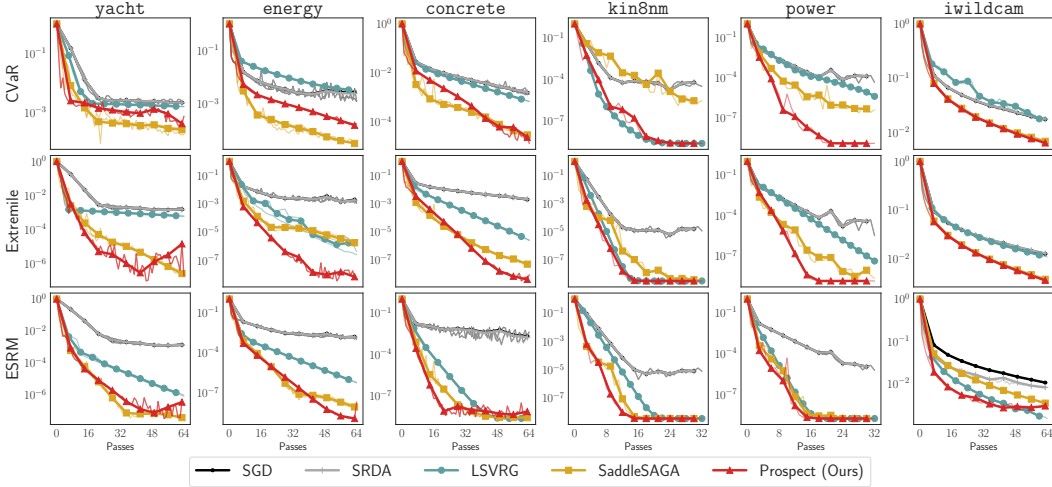

Figure 9: **Low shift cost settings.** Each row represents a different spectral risk objective with $\nu = 10^{-3}$ (instead of $\nu = 1$) while each column represents a different datasets. Suboptimality (9) is measured on the $y$-axis while the $x$-axis measures the total number of gradient evaluations made divided by $n$, i.e. the number of passes through the training set.

setting, LSVRG, SaddleSAGA, and Prospect seem to converge linearly empirically even without smoothness of the objective.

**Lowering Regularization.** Next, we decrease the $\ell_2$-regularization from $\mu = 1/n$ to $\mu = 1/(10n)$ and $\mu = 1/(100n)$. These settings are plotted in Fig. 11 and Fig. 12, respectively. Performance rankings among methods reflect those of the original parameters. For five of the six datasets, that is yacht, energy, concrete, kin8nm, and power, the regression tasks involve optimizing the squared error. This function is already strongly convex, with a constant depending on the smallest eigenvalue of the empirical second-moment matrix. When assuming that the input data vectors are

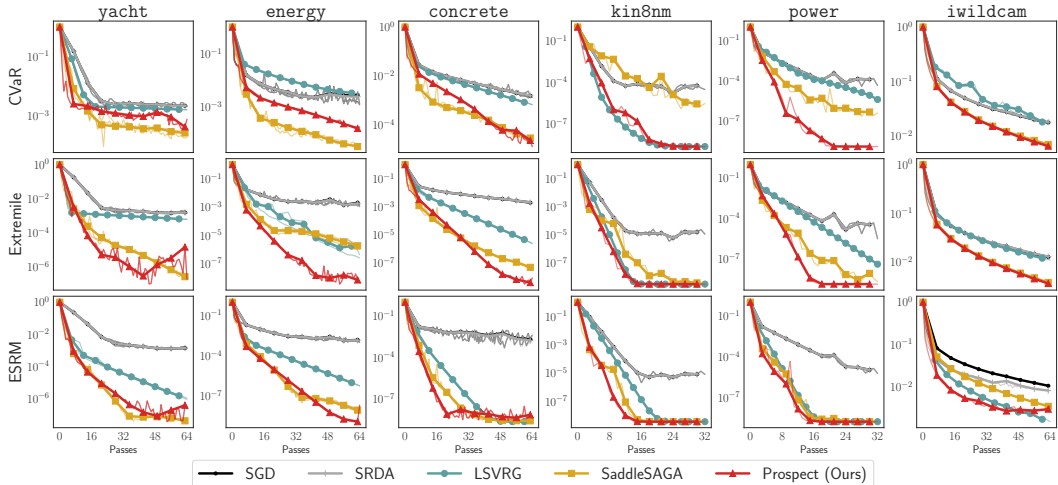

Figure 10: **No shift cost settings.** Each row represents a different spectral risk objective with $\nu = 0$ (instead of $\nu = 1$) while each column represents a different datasets. Suboptimality (9) is measured on the $y$-axis while the $x$-axis measures the total number of gradient evaluations made divided by $n$, i.e. the number of passes through the training set.

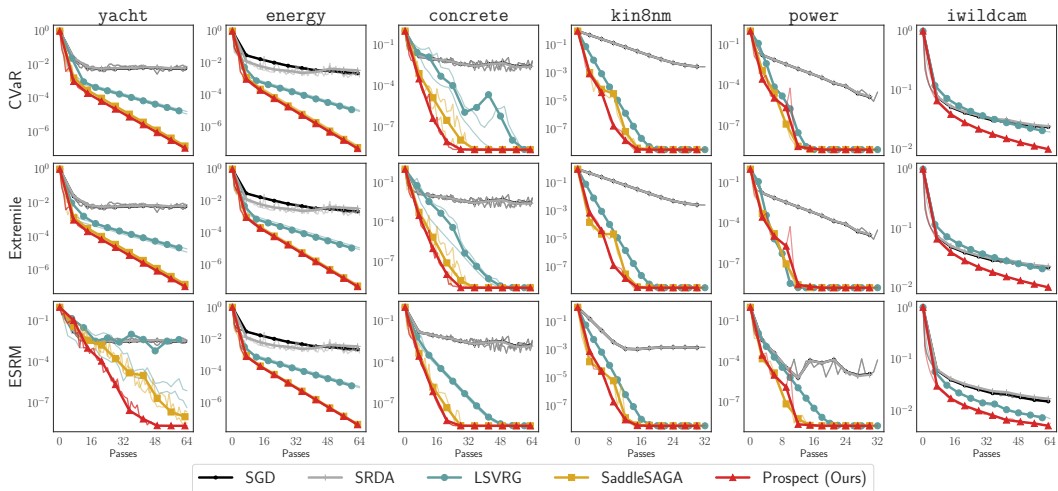

Figure 11: **Reduced $\ell_2$-regularization settings ($\mu = 1/(10n)$.** Each row represents a different spectral risk objective with $\mu = 1/(10n)$ (instead of $\mu = 1/n$) while each column represents a different dataset. Suboptimality (9) is measured on the $y$-axis while the $x$-axis measures the total number of gradient evaluations made divided by $n$, i.e. the number of passes through the training set.

bounded, this function is also $G$-Lipschitz. Thus, if the problem is already well-conditioned, we may observe similar behavior even at negligible regularization ($\mu = 5 \cdot 10^{-7}$ for iwildcam, for example).

**Comparison of Saddle-Point and Moreau Variants.** Finally, observe in Fig. 13 the comparison of SaddleSAGA variants (Appx. F), as well as the Moreau version of Prospect using Moreau envelope-based oracles (Appx. E). There are variants shown.

- **Primal LR = Dual LR:** The original variant of Palaniappan & Bach (2016), in which the primal and dual learning rates are set to be equal and searched as a single hyperparameter.
- **Search Dual LR:** Here, the primal learning rate is fixed as the optimal one for Prospect, and the dual learning rate is searched as a single hyperparameter.

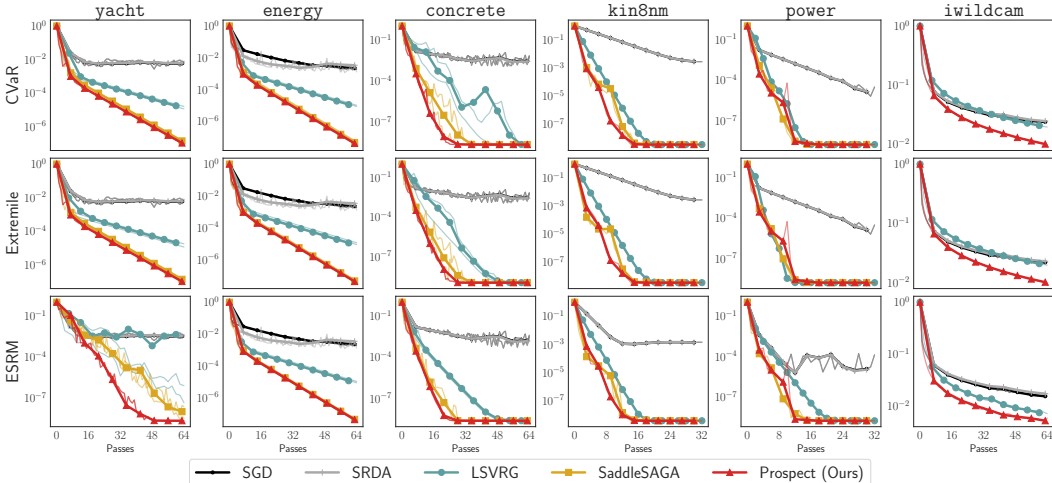

Figure 12: **Low $\ell_2$-regularization settings** ($\mu = 1/(100n)$**.** Each row represents a different spectral risk objective with $\mu = 1/(100n)$ (instead of $\mu = 1/n$) while each column represents a different dataset. Suboptimality (9) is measured on the $y$-axis while the $x$-axis measures the total number of gradient evaluations made divided by $n$, i.e. the number of passes through the training set.

- **Primal-Dual Heuristic:** In this version, used as the "SaddleSAGA" baseline in the main text, the dual learning rate is set to be $10n$ times smaller than the primal learning rate.
- **Prospect-Moreau:** The Moreau-envelope version of Prospect using proximal oracles.

We find that all methods besides the original variant (primal LR = dual LR) perform comparably on `yacht`, `energy`, `concrete`, `kin8nm`, and `power`. Notably, the ProxSAGA method performs similarly to Prospect and the saddle point-based baselines. While using the Moreau envelope results in accelerated rates in the ERM setting Defazio (2016), we find that the convergence rate is the same empirically. This phenomenon is in agreement with Thm. 17, which states that ProxSAGA will achieve the same linear convergence rate as Prospect, but will require a much less stringent condition on the shift cost $\nu$ than in the case of Prospect.

**Measuring Wall Time.** In Fig. 14, we measure the suboptimality for each method as a function of the wall time taken for a training run on the compute environment described in Appx. H. Across tasks, SGD is approximately 2 orders of magnitude faster than Prospect (and SaddleSAGA) but fails to converge due to bias/variance. In low-to-moderate sample size settings, LSVRG achieves a similar suboptimality to Prospect in a 2-3x smaller wall time. However, in the large sample size setting, LSVRG achieves essentially the same suboptimality as SGD compared to for Prospect, indicating that while fast, LSVRG does not demonstrate adequate accuracy. Also, at the large sample size setting, Prospect performs about 3x faster than SaddleSAGA but achieves the same suboptimality. Finally, a wall time cost that is not represented by the attached Figure 2 is the hyperparameter tuning cost. Both LSVRG and SaddleSAGA involve a grid search for tuning two hyperparameters, whereas Prospect has a single hyperparameter that can be tuned with binary search.

**Comparison to General-Purpose Learning Algorithms.** In Fig. 15, we compare the Prospect and baselines against optimization algorithms with widespread usage in generic learning problems, namely stochastic gradient descent (SGD) with momentum (Polyak, 1964; Nesterov, 2018) and Adam (Kingma & Ba, 2015). We use the default hyperparameters such as momentum 0.9 for SGD and $(\beta_1, \beta_2) = (0.9, 0.999)$ for Adam. For both methods, we use a batch size of 64 and tune the learning rate. Because of the bias and variance issues identified in Sec. 2, Adam and SGD with momentum perform comparably to standard SGD and are unable to converge on the objectives considered. Across objectives/datasets, final suboptimality often caps at $10^{-2}$ for both methods.

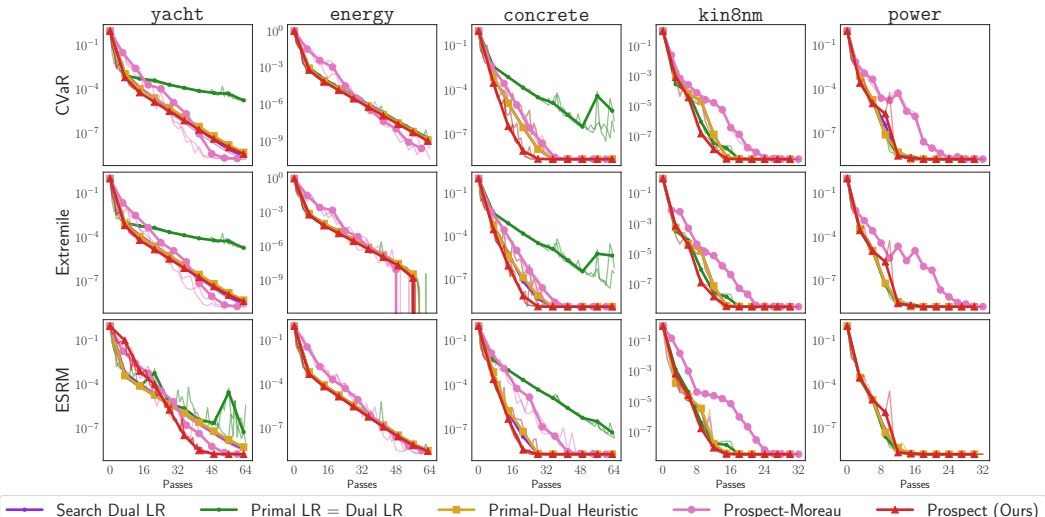

Figure 13: **SaddleSAGA and Prospect-Moreau method comparisons.** Each row represents a different spectral risk objective while each column represents a different dataset. Suboptimality (9) is measured on the $y$-axis while the $x$-axis measures the total number of gradient evaluations made divided by $n$, i.e. the number of passes through the training set.

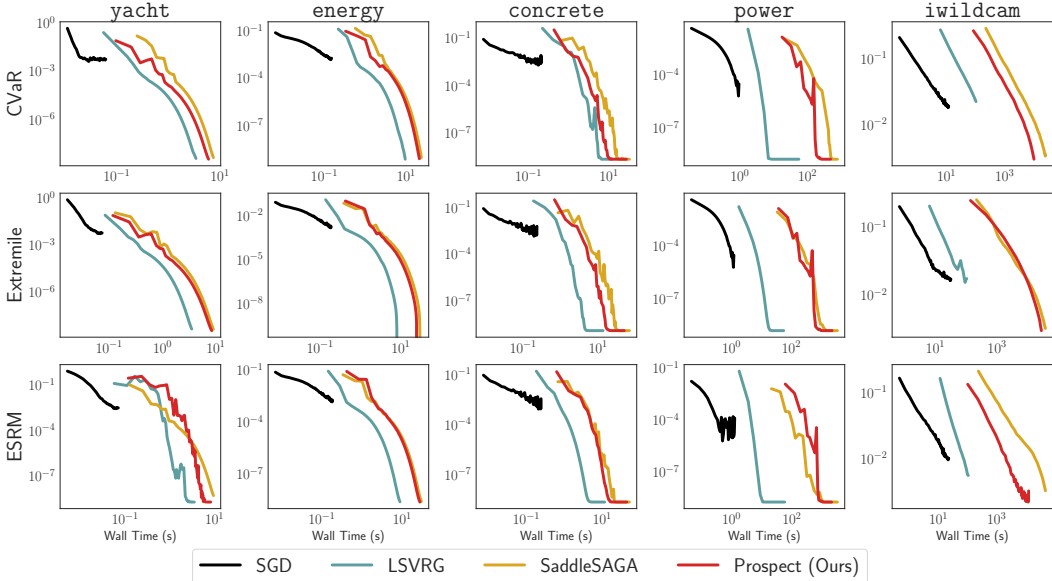

Figure 14: **Wall time comparisons.** Each row represents a different spectral risk objective while each column represents a different dataset. Suboptimality (9) is measured on the $y$-axis while the $x$-axis measures the total wall time in seconds on the environment described in Appx. H.

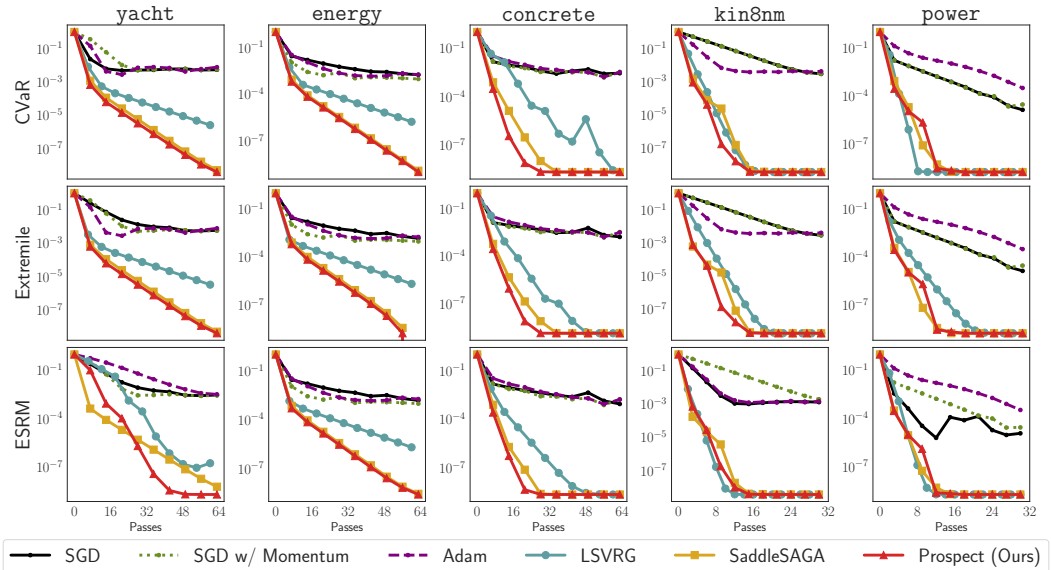

Figure 15: **Comparison to SGD w/ Momentum and Adam.** Each row represents a different spectral risk objective while each column represents a different dataset. Suboptimality (9) is measured on the $y$-axis while the $x$-axis measures the total number of gradient evaluations made divided by $n$, i.e. the number of passes through the training set.

