# OpenReview forum: "Distributionally Robust Optimization with Bias and Variance Reduction"
_ICLR.cc/2024/Conference — ICLR 2024 spotlight_

### Official Review · Reviewer_Ed47 · 2023-10-29

**Soundness:** 3 good
**Presentation:** 3 good
**Contribution:** 3 good
**Rating:** 8
**Confidence:** 3

**Summary:**

The paper provides a practical algorithm to solve DRO problems. The algorithm enjoys a linear convergence rate and it does not require tuning multiple hyperparameters. The author further demonstrated the proposed algorithm can achieve 2-3x speedup compared to other methods vias read-data sets.

**Strengths:**

1. The paper is well-written and the contribution is significant. The algorithm is easy to understand and easy to implement. The authors also provide intuition behind the algorithm via texts and figures.

**Weaknesses:**

1. The paper needs to solve an optimization problem with n variables in each iteration (line 10 in Algorithm 1). It is very computationally expensive if n is large. I wonder whether the algorithm can reduce to another one-step stochastic gradient descent in this step. For example, the two-time scale method in distributionally robust RL [1].

[1] Yang, Wenhao, et al. "Avoiding model estimation in robust Markov decision processes with a generative model." arXiv preprint arXiv:2302.01248 (2023).

2. The paper only provides theoretical results for the case $\mu>0$. I am not sure if it is necessary or if it is only for theoretical convenience. It is known that the DRO objective $\mathcal{R}_\sigma$ could induce smoothness. Some discussions are needed and numerical examples with $\mu=0$ will also help.

**Questions:**

The papers (including the appendix) are long. I am not sure whether it is suitable for a conference review.

I am curious that whether there are some unbiased estimators could be used, e.g., Wang et al (2023).

Wang, Shengbo, et al. "A finite sample complexity bound for distributionally robust Q-learning." International Conference on Artificial Intelligence and Statistics. PMLR, 2023.

---

> ### Author Response · Authors · 2023-11-20
> **Response to Reviewer Ed47**
>
> Thank you for your insightful comments! We address your main concerns below.
>
> **The paper needs to solve an optimization problem with $n$ variables in each iteration (line 10 in Algorithm 1). It is very computationally expensive if $n$ is large. I wonder whether the algorithm can reduce to another one-step stochastic gradient descent in this step. For example, the two-time scale method in distributionally robust RL [Yang (2023)].**
>
> Thank you for identifying this reference. An optimization problem with $n$ scalar variables indeed needs to be solved which would be prohibitive if solved iteratively with gradient-based optimization. However, as we discuss in Section 2 and Appendix C, the exact solution is available by applying the pool adjacent violators algorithm, which in our context requires simply two linear passes through an $n$-length vector of losses ($O(n)$ cost)). In fact, we perform this operation even faster by compiling the computation of the solution. In that sense, this optimization problem is largely negligible. The existence of a fast algorithm for computing the solution of the maximization problem relies on the specific structure of the spectral risk measure uncertainty set, which provides additional motivation for this choice.
>
> **The paper only provides theoretical results for the case $\mu>0$. I am not sure if it is necessary or if it is only for theoretical convenience... Some discussions... and numerical examples with $\mu=0$ will also help.**
>
> Thank you for raising this point. It is true that $\mu>0$ is important theoretically to achieve linear convergence, as this makes the objective $R_\sigma$ strongly convex. While we could put this condition on the losses $\ell_1, …, \ell_n$ themselves, we instead require them to be $G$-Lipschitz continuous so we may relate changes in the loss $\ell_i(w_1)$ and $\ell_i(w_2)$ to changes in inputs $w_1$ and $w_2$. A function cannot be both Lipschitz continuous and strongly convex unless we restrict its domain to a compact set. From a practical point of view, however, the algorithm generally works on settings that violate these assumptions. For example, the regression experiments use squared loss, which is not Lipschitz continuous, so by setting $\mu=0$, we observe the results below on the `yacht` and `energy` datasets (lower is better). In all cases, Prospect performs best or close to best, as in other numerical experiments. Extending theoretically to the non-strongly convex case is an interesting avenue for future work.
>
> *Performance on `yacht` after 64 epochs.*
> | Algorithm | CVaR Suboptimality | Extremile Suboptimality | ESRM Suboptimality |
> | ----------- | ----------- | ----------- | ----------- |
> | SGD                    | 6.311e-3       | 6.379e-3      | 3.490e-3      |
> | LSVRG                | **1.146e-7**  | 6.715e-5      | 6.356e-08       |
> | SaddleSAGA       | 1.707e-7       | 1.954e-7       | 1.824e-08       |
> | Prospect (Ours)   | 1.626e-7       | **1.680e-7**  | **2.124e-09**       |
>
> *Performance on `energy` after 64 epochs.*
> | Algorithm | CVaR Suboptimality | Extremile Suboptimality | ESRM Suboptimality |
> | ----------- | ----------- | ----------- | ----------- |
> | SGD                    | 4.075e-3      | 4.082e-3  | 4.201e-3      |
> | LSVRG                | 1.581e-5      | 1.594e-5  | 1.623e-5       |
> | SaddleSAGA       | 8.685e-8      | **7.714e-8**  | **7.174e-8**       |
> | Prospect (Ours)   | **7.232e-8**      | 7.233e-8  | 7.186e-8       |
>
> **The papers (including the appendix) are long.**
>
> The length of the appendix sections is due in part to optional supplementary material included for the benefit of the reader. This includes the detailed exposition on the implementation of the algorithm (Appx C), experimental details (Appx H), and additional experiments (Appx I). Theoretical appendix sections (such as Appx G)  provide technical references to make the paper self-contained. On the other hand, the essential appendices, Appx D/E are under 20 pages long.
>
> **I am curious that whether there are some unbiased estimators could be used, e.g., Wang et al (2023).**
>
> Thank you for identifying this interesting work. In the setting of Wang et al (2023), the authors optimize their policy by Q-learning iterations to approach a fixed-point of a distributionally robust variant of the Bellman equation. There is not a gradient-based optimization procedure or gradient estimation step as in our problem. That being said, extending our formulation to the reinforcement learning and control setting is an interesting avenue for future work.

---

> > ### Comment · Reviewer_Ed47 · 2023-11-21
> > **I am satified with the the author's response.**
> >
> > I am satified with the the author's response. I only want to point out one remaing thing that O(n) each iteration is not negligible.

---

### Official Review · Reviewer_ss6k · 2023-11-01

**Soundness:** 4 excellent
**Presentation:** 3 good
**Contribution:** 4 excellent
**Rating:** 8
**Confidence:** 4

**Summary:**

In this paper the authors have developed an iterative algorithm to solve optimization problems with spectral risk measures.
This is equivalent to optimization a distributionally robust optimization problem where the ambiguity set is the space of all probability distributions with a term to penalize the distance from the uniform distribution.
Their algorithm consists of 3 key components
1. Gradients steps of the spectral risk measure
2. Bias reduction for the gradient estimates
3. Variance reduction for the gradient estimates

Each of these steps can be done efficiently.
The authors also prove that the algorithm converges to the correct solution linearly
Finally, the authors illustrate the performance of their algorithm through numerical experiments on a variety of tasks.

**Strengths:**

**originality**: I think this work is original. This paper develops a new iterative algorithm for DRO problems with ambiguity sets connected to spectral risk measures.

**quality**: The work is sound and presents justifies the algorithm with both theoretical and experimental results

**clarity**: the work is well presented and the numerical results are clearly explained.

**significance**: I believe the work is significant since it expands the type of DRO problems that can be solved with iterative algorithms and hence the size of DRO problems that can be solved which has always been a key limitation of it.

**Weaknesses:**

I feel the discussion on the connection to DRO is quite limited. The ambiguity sets for equation (2) is the entire space of distributions which is quite large and not very useful. I believe the presence of the penalty term shows that this problem can be equivalent to a tighter ambiguity set (maybe restricted by the divergence metric used in the objective) and it would be good if the authors can discuss this.

f-divergences are quite a broad type of divergence as discussed in the appendix. The 3 types of divergences considered in the experimental section are quite limited. It would be good if authors can also discuss some of the other popular divergences such as KL-divergence etc. and compare the developed algorithm to existing approaches for DRO problems with these divergences.

**Questions:**

1. What is the scale of the problems you can solve. How much time does it take to solve the problem.
2. Will the algorithm work if used along with a projection step to solve constrained optimization problems?

---

> ### Author Response · Authors · 2023-11-20
> **Response to Reviewer ss6k**
>
> Thank you for your thorough comments! We address your main concerns below.
>
> **The ambiguity sets for equation (2) is the entire space of distributions which is quite large and not very useful. I believe the presence of the penalty term shows that this problem can be equivalent to a tighter ambiguity set (maybe restricted by the divergence metric used in the objective) and it would be good if the authors can discuss this.**
>
> Thank you for this point. We added in the revision to Section 1 that the uncertainty set is not necessarily the space of all distributions on $n$ atoms, but a user-defined, problem-specific subset. Increasing the shift cost indeed has a “tightening” effect, but the uncertainty set can be explicitly changed by the choice of $\sigma$. See for example the revised Figure 6 in which we visualize the uncertainty sets for different choices of $\sigma$, which allows for arbitrarily sized uncertainty sets.
>
> **The 3 types of $f$-divergences considered in the experimental section are quite limited. It would be good if authors can also discuss some of the other popular divergences such as KL-divergence etc. and compare the developed algorithm to existing approaches for DRO problems with these divergences.**
>
> The three objective classes considered are the CVaR, extremile, and ESRM, which refer to the choice of uncertainty set. The discussion on $f$-divergences, on the other hand, refers to the choice of penalty and is independent of the uncertainty set. We provide Appendix B the analysis of the distributionally robust objective and in Appendix D the algorithm for *any* $f$-divergence for which $f$ is strongly convex and we have access to the maximizer over the uncertainty set $\mathcal{P}$ with the given shift penalty. From an implementation side, we describe in Appendix C how to implement the algorithm for the same set of $f$-divergences, which include the KL.
>
> **What is the scale of the problems you can solve? How much time does it take to solve the problem?**
>
> To address this point more generally, we have added Figure 13 in Appendix I to measure the wall clock time of Prospect and the baselines. Across tasks, SGD is approximately 2 orders of magnitude faster than Prospect (and SaddleSAGA) but fails to converge due to bias/variance. In low-to-moderate sample size settings, LSVRG achieves a similar suboptimality to Prospect in a 2-3x smaller wall time. However, in the large sample size setting, LSVRG achieves essentially the same suboptimality as SGD compared to for Prospect, indicating that while fast, LSVRG does not demonstrate adequate accuracy. Also, at the large sample size setting, Prospect performs about 3x faster than SaddleSAGA but achieves the same suboptimality. Finally, a wall time cost that is not represented by the attached Figure 2 is the hyperparameter tuning cost. Both LSVRG and SaddleSAGA involve a grid search for tuning two hyperparameters, whereas Prospect has a single hyperparameter that can be tuned with binary search.
>
> **Will the algorithm work if used along with a projection step to solve constrained optimization problems?**
>
> To incorporate constraints, one may consider the Moreau-Prospect variant (described in Appendix E of the revised manuscript) by modifying the loss. Namely, assuming the constraint set to be closed and convex, we may redefine the loss to be the sum of the original loss and the indicator function of the constraint set. We may then employ the proximal operator of such augmented losses defined by the individual loss plus the indicator function of the constraint set. For the squared loss, such a proximal operator can easily be computed in closed form for simple constraints such as an $\ell_2$ ball. For binary logistic loss and multinomial logistic loss, one may also consider approximations of the corresponding proximal operator by means of an additional projection. Alternatively, an additional proximal step could be incorporated into the Prospect algorithm as done in the SAGA algorithm (Defazio, 2014). The theoretical analysis of the corresponding algorithm is an interesting avenue for future work.

---

### Official Review · Reviewer_ucUf · 2023-11-02

**Soundness:** 4 excellent
**Presentation:** 4 excellent
**Contribution:** 4 excellent
**Rating:** 8
**Confidence:** 4

**Summary:**

The authors studied DRO with spectral risk measure and modeled the ambiguity set as $f$-divergence. Strong duality result, together with a stochastic optimization algorithm, was proposed to optimize the formulation. The idea of their proposed algorithm is to reduce the bias and variance of gradient estimators while maintaining small computational costs when performing iteration updates. Convergence analysis revealed that their algorithm converges linearly for "smooth regularized losses". A comprehensive numerical study is performed to show the superior performance of their algorithm.

**Strengths:**

- It seems the strong duality of $f$-divergence DRO when considering spectral risk measure in Proposition 3 is new in the literature. Appendix B shows a range of nice properties regarding the formulation (2).
-  The designed Prospect Algorithm is novel and operates by reducing the bias and variance of gradient estimators while maintaining small computational costs. Nice convergence guarantees are established in Theorem 1 provided that the regularization for $f$-divergence is lower bounded and the regularization value of the model parameter $w$ is positive.
- A comprehensive numerical study is performed to show the superior performance of their algorithm.

**Weaknesses:**

See Questions part.

**Questions:**

- In contrast to the standard DRO that models $\mathcal{P}$ as a probability simplex on $n$ atoms, the authors consider the spectral risk measure such that $\mathcal{P}=\text{CovexHull}(\text{Permutaions of }(\sigma_1,\ldots,\sigma_n))$. I was confused about the motivation for using such a spectral measure. In which scenarios will people focus on this type of measure? The authors should give more justification on this part.
- When deriving the strong duality result in Proposition 3, the authors assume that the conjugate function of $f$, denoted as $f^*$ satisfies $|f^*(y)|<\infty, \forall y\in\mathbb{R}$. However, when studying the standard $f$-divergence DRO with $\mathcal{P}$ being the probability simplex on $n$ atoms (see Section 3.2 in (Shapiro Alex, 2017)), one does not require such an Assumption. Could the authors justify why we need this restrictive assumption? Also, I think the authors missed this important citation.

Ref: Shapiro A. Distributionally robust stochastic programming[J]. SIAM Journal on Optimization, 2017, 27(4): 2258-2275.
- In Proposition 4, the assumption that $f$ is $\alpha_n$-strongly convex may be restrictive. Is it possible to relax this assumption into strictly convex?
- It is difficult to tell why Proposition 5 holds. The authors should add complete proof regarding this proposition. Besides, in Eq. (15) I find the authors implicitly assume the conjugate function $f^*$ is differentiable. I am wondering if the authors assume the differentiability of the function $f$, or the condition $f^*$ is differentiable can be derived based on some conditions?
- In page 28-32, page 35-37, and page 39-40, some equations are highlighted in color. What is the meaning of those highlighted colors?

---

> ### Author Response · Authors · 2023-11-20
> **Response to Reviewer ucUf**
>
> We appreciate your thorough reading of the work, including the mathematical proofs. We address your main concerns below.
>
> **...the authors consider the spectral risk measure such that P = cvxhull(permit)... In which scenarios will people focus on this type of measure?**
>
> DRO problems are defined by the uncertainty set $\mathcal{P}$ of probability distributions. The common choice of $\mathcal{P}=$ probability simplex leads to a “very large” uncertainty set as the resulting objective is the max loss over all the examples. This led to prior work considering $\mathcal{P}$ as a subset of the simplex, e.g. the CVaR.
>
> We consider a general form $\mathcal{P}$ defined by the permutations of some weights — this is much smaller than the entire simplex but includes several important special cases such as the CVaR. Notably, for this general class, known as the family of *spectral risk measures*, maximization over $\mathcal{P}$ and other similar problems can be computed efficiently.
>
> We have added additional background in Section 2 and Appendix B of the revision on spectral risk measures — this gives the motivation from a statistical modeling perspective and computational perspective.
>
> **When deriving the strong duality result in Proposition 3, the authors assume that the conjugate function of $f$, denoted as $f^\star$ satisfies $|f^\star(x)| <+\infty$. However, when studying the standard divergence DRO with $\mathcal{P}$ being the probability simplex on $n$ atoms (see Section 3.2 in (Shapiro Alex, 2017)), one does not require such an Assumption. Could the authors justify why we need this restrictive assumption?... The authors should add complete proof regarding [Proposition 5]. Besides, in Eq. (15)... I am wondering if the authors assume the differentiability of the function $f$?**
>
> The function $f$ in Proposition 4 references the shift penalty, which is in the form of an $f$-divergence, whereas the uncertainty set is in the form of a spectral risk measure and is independent of $f$. The finiteness condition and strong convexity of $f$ were ultimately not necessary for the duality result, and we have updated Proposition 3 to also include the full proof of the previous Proposition 5 in the revision. Now, we only require that $f$ and $f^\star$ are strictly convex, which is satisfied by all the f-divergences considered.
>
> **In Proposition 4, the assumption that $f$ is $\alpha_n$-strongly convex may be restrictive.**
>
> While not necessary for deriving the duality properties, the strong convexity of $f$ will be necessary for quantifying the continuity of $l \mapsto q^l \in \mathcal{P}$ as a function of the loss vector $l \in \mathbb{R}^n$. As a result, our objective can fail to be smooth without $f$ being strongly convex, in which case we cannot expect the linear convergence rate that appears in the main result Theorem 1.
>
> **Also, I think the authors missed this important citation.**
>
> Thank you for identifying this missed reference, which we have now added to the revised Related Works section.
>
> **In page 28-32, page 35-37, and page 39-40, some equations are highlighted in color. What is the meaning of those highlighted colors?**
>
> The colors are there for the convenience of the reader so that the various quantities of interest (or related quantities) are easy to track from result to result. We have clarified this in the revised text. Thank you!

---

> ### Comment · Reviewer_ucUf · 2023-11-20
> **Score after rebuttal**
>
> I am satisfied with the authors' rebuttal so I will keep my score as it is. However, please note the bad box environment at the end of page 33 and fix it.

---

### Official Review · Reviewer_12Ff · 2023-11-06

**Soundness:** 3 good
**Presentation:** 2 fair
**Contribution:** 3 good
**Rating:** 8
**Confidence:** 3

**Summary:**

The authors propose Prospect, a distributionally robust algorithm that requires the tuning of only one hyperparameter. The algorithm's formulation includes a reweighed empirical risk and an f-divergence term to account for the cost requires to shift from a uniform representation of the training data. The authors propose both bias and variance reduction procedures to ensure that both quantities vanish, hence guaranteeing convergence.  Prospect is proved to enjoy a linear convergence for any positive shift cost on regularized convex losses and is empirically competitive with the considered baselines.

**Strengths:**

The paper has multiple strengths in terms of originality, quality and significance. The clarity component is lacking in the ways that I will explain in the weaknesses section.

- The paper has solid contributions compared to prior work.
- The theoretical analysis is sound and well supported.
- The authors also discuss the case where the hypotheses of this work are violated and argue that their algorithm still converges in that case.
- The empirical evaluation considers 3 important problems and covers both classification and regression.
- The proposed algorithm and all baselines are trained to convergence as shown in all figures. The hyperparameter selection seems to also have been done in a fair manner.

**Weaknesses:**

I enumerate below the weaknesses of this work, which to me are important to address but do not undermine the overall quality of this work. I hope the authors will be able to address them during the rebuttal.

- Presentation and clarity: Although the authors clearly attempt to make the paper as clear as possible, some key notions are never introduced. For instance, CVaR was never formally introduced. It is also unclear to me what 0.5-CVaR, 2-extremile, and 1-ESRM really mean mathematically speaking. The entire notion of spectral risk-based needs to be explained much more clearly. I also don't understand Figure 6, and overall the captions of the figures could be more descriptive. The abstract is heavily jargony, and so is a lot of the introduction text. I would appreciate a clearer presentation of this work, its motivation, and key notions at least in the introduction.

- Baselines: I find the baselines considered by this work to be very restrictive. Some of the problems considered in the experiments section violate the hypotheses that lead to Prospect's theoretical guarantees, and therefore it would only be fair to compare it to other algorithms that do not enjoy the same guarantees and do not have only one hyperparameter to tune. In fact, the number of hyperparameters being only one is not as big of an advantage as portrayed in my opinion, since more practical algorithms would have default values for their hyperparameters that would work well in practice. I recommend comparing additionally to at least Adam and SGD with momentum.

**Questions:**

Please refer to the weaknesses section.

---

> ### Author Response · Authors · 2023-11-20
> **Response to Reviewer 12Ff**
>
> Thank you for your thorough and concrete comments! We address your main concerns below.
>
> **Presentation and clarity: ...some key notions are never introduced. For instance, CVaR was never formally introduced... the entire notion of spectral risk-based needs to be explained much more clearly. The abstract is heavily jargony... I would appreciate a clearer presentation of this work, its motivation, and key notions.**
>
> Thank you for pointing this out! We have taken several steps to address this feedback:
>  - We have now updated the abstract, Section 1, and Section 2 to address these concerns. We have updated the abstract and introduction to appeal to a wider audience and moved the DRO examples into the Related Works.
> - We have introduced and motivated spectral risk measures both from a statistical modeling and computational perspective and given the definition of CVaR in the revised Section 2.
> - In Appendix B, we added an extended discussion on spectral risk measures including their historical use in mathematical finance and their counterpart for continuous random variables.
> -  The precise definitions of all spectral risk measures are referenced in Section 2. Finally, we have edited the caption text for Figure 6 (Appendix B) and several experimental figures for increased clarity.
>
> **Baselines: ...I recommend comparing additionally to at least Adam and SGD with momentum.**
>
> Thank you for bringing up this point.
>
> We have added empirical comparisons with  (a) SGD with momentum set to $0.9$ and tuned learning rate and (b) Adam with $(\beta_1, \beta_2) = (0.9, 0.999)$ with tuned learning rate. Please see the results in Figure 14 of the revision.
>
> We observe that these methods behave similarly to vanilla SGD and fail to converge to the optimum. Indeed, this is due to the bias and variance issues of naive minibatch approaches (see discussion in Section 2), which is common across all these baselines.
>
> Thus, while the proposed Prospect can attain convergence up to numerical error with a tuned learning rate,  the SGD variants will not converge no matter how much their hyperparameters are tuned.

---

> > ### Comment · Reviewer_12Ff · 2023-11-22
> > **Thank you for the response; I raised my score**
> >
> > Thank you for addressing my concerns about the clarity of the manuscript and comparing to other baselines. I have now raised my score from 6 to 8.

---

### Author Response · Authors · 2023-11-20
**Highlights of the Responses**

We thank the reviewers for their hard work reviewing our paper and providing insightful and concrete comments! We have revised the manuscript to incorporate these suggestions. Here are some of the highlights.

- In response to comments from Reviewers **12Ff**, **ucUf**, and **ss6k**, we edited Section 1, Section 2, and Appendix B to improve the exposition of spectral risk measures, specifically regarding their motivation in DRO problems from a statistical and computational perspective.
- In response to Reviewer **Ed47** and Reviewer **ss6k**'s comments, we have described key methodological properties of Prospect, such as the computational aspects of solving the inner maximization problem and potential extensions to the non-strongly convex and constrained settings.
- We have added several new experiments based on Reviewer **ss6k** and Reviewer **12Ff**’s comments, including comparisons of suboptimality and wall time as well as comparisons against general-purpose optimizers used in machine learning, such as SGD with momentum and Adam.

Please see the individual responses for more details. **The revisions to the manuscript are highlighted in green text.**

---

### Author Response · Authors · 2023-11-21
**Thank you for your feedback / Addressing additional concerns**

Dear reviewers,

As the discussion period draws to a close, we kindly request that you take a moment to review our responses to your questions and comments.

In response to the comments raised by Reviewers 12Ff and ss6k, we have carefully updated Section 1, Section 2, and Appendix B to improve the motivation and exposition of spectral risk measures. We have also provided additional baselines as recommended by Reviewer 12Ff. We have fully incorporated the comments from Reviewers ucUf and Ed47 and would like to thank them for their recommendations and responses. Please see the attached revision that reflects these updates.

If any further concerns or questions arise that we could address, please do not hesitate to reach out. We appreciate your time and feedback.

Sincerely,
The Authors

---

### Meta-Review · Area_Chair_TYXb · 2023-12-15

**Metareview:**

The authors focus on DRO with spectral risk measures, modeling the ambiguity set through f-divergence. They introduce a strong duality result and a novel stochastic optimization algorithm called Prospect, with minimal hyperparameter tuning. The algorithm's core aim is to reduce the bias and variance of gradient estimators while keeping computational costs low during iteration updates. Convergence analysis shows linear convergence for "smooth regularized losses.”

**Justification For Why Not Higher Score:**

The paper introduces a novel algorithm solving a particular DRO problem; it's interesting but the scope may not be large enough to be an oral paper.

**Justification For Why Not Lower Score:**

I concur with all reviewers about the novelty and the intellectual merit of the paper.

---

### Decision · Program_Chairs · 2024-01-16

Accept (spotlight)